# Dietary amino acids promote glucagon-like hormone release to generate global calcium waves in adipose tissues in *Drosophila*

Muhammad Ahmad [1,8], Shang Wu[2,8], Shengyao Luo[3], Wenjia Shi[4], Xuan Guo [5], Yuansheng Cao [6] ✉, Norbert Perrimon [1,7] ✉ & Li He [2] ✉

Propagation of intercellular calcium waves through tissues has been found to coordinate different multicellular responses. Nevertheless, our understanding of how calcium waves operate remains limited. In this study, we explore the real-time dynamics of intercellular calcium waves in *Drosophila* adipose tissues. We identify Adipokinetic Hormone (AKH), the fly functional homolog of glucagon, as the key factor driving $Ca^{2+}$ activities in adipose tissue. We find that AKH, which is released into the hemolymph from the AKH-producing neurosecretory cells, stimulates calcium waves in the larval fat by a previously unrecognized gap-junction-independent mechanism to promote lipolysis. In the adult fat body, however, gap-junction-dependent intercellular calcium waves are triggered by a presumably uniformly diffused AKH. Additionally, we discover that amino acids activate the AKH-producing neurosecretory cells, leading to increased intracellular $Ca^{2+}$ and AKH secretion. Altogether, we show that dietary amino acids regulate the AKH release from the AKH-producing neurosecretory cells in the brain, which subsequently stimulates gap-junction-independent intercellular calcium waves in adipose tissue, enhancing lipid metabolism.

Adipose tissue is a primary organ of fat storage that coordinates energy metabolism pathways under various nutritional states. Such metabolic processes are under the hormonal control of insulin and glucagon, the two key hormones playing a central role in energy storage and homeostasis regulation. Notably, glucagon promotes lipid breakdown (lipolysis) in the adipose tissue by elevating cytoplasmic $Ca^{2+}$ levels by activation of its GPCR receptor[1]. Intriguingly, these $Ca^{2+}$ activities are spatially organized into intercellular calcium waves (ICWs) across the mammalian liver[2]. However, the mechanism and biological significance of these ICWs in vivo remains poorly understood. Fruit flies have a functional homolog of glucagon, AKH, which plays similar roles in mobilizing adipose lipids upon starvation[3–5]. AKH

triggers $Ca^{2+}$ influx into the isolated insect fat body[6]. Additionally, $Ca^{2+}$ signaling via *Gαq, Gγ1, Plc21C*, and store-operated $Ca^{2+}$ entry pathway has been shown to regulate lipid metabolism in fly adipose tissue[7–9]. However, how AKH regulates fat tissue $Ca^{2+}$ dynamics in vivo and its connection with lipid metabolism remains unexplored. In this study, we demonstrate that AKH triggers significant ICWs in the fly adipose tissue, presenting a unique opportunity to investigate the mechanics, regulation and biological function of ICWs in vivo.

ICWs are complex tissue-level biological events that widely exist in multicellular organisms and impact many processes, e.g., muscle contraction[10,11], gene expression[12,13], cellular proliferation[14], differentiation[15], neuronal firing and excitability[16], and metabolism[17].

[1]Department of Genetics, Harvard Medical School, Boston, MA, USA. [2]The First Affiliated Hospital of USTC, Division of Life Sciences and Medicine, University of Science and Technology of China, Hefei, Anhui, China. [3]Yuanpei College, Peking University, Beijing, China. [4]Department of Applied Physics, Xi'an University of Technology, Xi'an, Shaanxi, China. [5]Life Science Institute, Jinzhou Medical University, Jinzhou, Liaoning, China. [6]Department of Physics, Tsinghua University, Beijing, China. [7]Howard Hughes Medical Institute, Harvard Medical School, Boston, MA, USA. [8]These authors contributed equally: Muhammad Ahmad, Shang Wu. ✉e-mail: yscao@tsinghua.edu.cn; perrimon@genetics.med.harvard.edu; lihe19@ustc.edu.cn

Dysregulated $Ca^{2+}$ signaling results in various diseases, including neurological disorders, cardiovascular diseases and cancer[18,19]. ICWs are generated by an increase in cytoplasmic $Ca^{2+}$ propagating between cells, which coordinates concerted action at the tissue level. In *Drosophila*, ICWs have been found to integrate the upstream inputs including mechanical stimulus and correlated with organ sizes[20–23], involved in egg activation[24], modulate cell fate during oogenesis[25], synchronize inflammatory responses[26], apical constriction in the wounded epithelium[27], and detect wound sites[23,28,29]. In most situations, the propagation of ICWs is governed by two mechanisms: the direct diffusion of the secondary messengers i.e. $Ca^{2+}$ and $IP_3$ through gap junctions that stimulate $Ca^{2+}$ efflux in neighboring cells[30,31]; and the paracrine signaling, where the secretion of extracellular ATP stimulates nearby cells[32]. Despite these insights, the underlying biophysical mechanisms and the biological significance of the emergent pattern and signal propagation of ICWs in different biological systems have largely remained unidentified.

Using the *Drosophila* larval fat body, we analyze the in vivo real-time dynamics of ICWs by live-cell imaging. We show that global ICWs are triggered by AKH in the larval fat body, reminiscent of glucagon-induced $Ca^{2+}$ waves observed in the human liver[1]. Interestingly, we find a global ICW propagation in vivo, which is controlled by a gap junction-independent mechanism. We also identify the dietary factor regulating the AKH-dependent ICWs. Previous studies show that AKH, like the mammalian Glucagon, regulates blood sugar homeostasis and modulates starvation-induced hyperactivity in adults[4,33]. However, compared to insulin, the regulation of AKH remains less understood[34]. It has been recently demonstrated that fly insulin-producing cells (IPCs) sense the dietary amino acid leucine, which triggers the secretion of dILPs[35]. In contrast, whether specific amino acids control AKH secretion remains unknown. Here, we show that AKH secretion is stimulated by specific amino acids, which further promotes fat loss in both larvae and adult flies.

## Results

### ICWs in fly larval fat are induced by a brain-derived factor

The fly fat body, a functional equivalent of the mammalian liver and adipose tissue, plays central roles in metabolic regulation and nutrient sensing[36]. ICWs have been reported in the mammalian liver[37]. Similarly, ICWs in the larval fat body have been mentioned[22]. However, the functional consequences of what triggers such waves have not been investigated in vivo. To address these questions, we expressed the $Ca^{2+}$ indicator GCaMP5G specifically in the larval fat body, and immobilized larvae within a glass channel (Supplementary Fig. 1A). Surprisingly, we noticed global ICWs that emanate in a periodic fashion from the larval head to tail (Fig. 1A, B, Supplementary Movie 1). We further examined the free-behaving larvae and observed prominent global ICWs, indicating that these waves were not a consequence of immobilization (Supplementary Fig. 1B, C). Paralyzing the larva by feeding them neurotoxin Tetrodotoxin (TTX) completely stopped muscle movements, but the global ICWs persisted, suggesting the ICWs are not caused by muscle contractions (Supplementary Fig. 2).

Interestingly, the isolated larval fat bodies displayed no ICWs, except for a few cells that sustained damage during dissection and maintained abnormally high $Ca^{2+}$ (Fig. 1D, Supplementary Fig. 1D), suggesting that the ICWs are triggered by signals from other organs. To identify the organ responsible for this regulation, we co-cultured fat bodies with larval muscle/cuticle, intestine, and brain respectively (Fig. 1C). ICWs were only triggered in fat bodies co-cultured with brains (Fig. 1C–E, Supplementary Movie 2), suggesting that these waves are initiated by a brain-derived factor.

To identify the putative factor, we generated a brain-conditioned medium by incubating Schneider's medium together with dissected larval brains, and applied the conditioned medium to isolated fat bodies. The brain-conditioned medium induced robust ICWs (Fig. 1F, G). To determine the nature of this factor, we filtered the brain-conditioned medium through a 10 K MWCO concentrator that removed molecules larger than 10 kDa. The filtrate evoked robust fat body ICWs, suggesting that the factor has a small molecular weight (Fig. 1G). We then treated the conditioned medium with proteinase K, DNAse I or RNAse A. Only proteinase K significantly reduced $Ca^{2+}$ activities, suggesting that the factor is likely to be a small peptide (Fig. 1H).

### AKH released from AKH-producing cells (APCs) is responsible for inducing ICWs in the fat body

Then we screened 32 major fly neuropeptides on isolated fat bodies[38]. Only AKH evoked significant ICWs in the fat body (Fig. 2A). The fat body responds within several seconds to the addition or removal of AKH, and there was no activity reduction to prolonged AKH addition (up to 4 h), suggesting that the AKH receptor (AkhR) has fast binding and dissociation kinetics with no adaptation (Fig. 2B). The fat body is sensitive to AKH at doses as low as 0.1 ng per mL, which is within the physical range. And both the amplitude and frequency of $Ca^{2+}$ oscillations increased with AKH concentration (Fig. 2C, Supplementary Fig. 3A–D). Finally, knocking down the *AkhR* reduced the $Ca^{2+}$ oscillation frequency, suggesting that the frequency is also regulated by the concentration of the receptor (Supplementary Fig. 3E, F).

The ICWs were blocked immediately after adding the anti-AKH antibody in the medium to sequester AKH (Fig. 2D, E). As AKH is secreted from the APCs in the ring gland, we knocked down *Akh* in the APCs or inhibited AKH release by expressing the neuronal silencing potassium channel Kir2.1[39]. These treatments suppressed the release of AKH resulting in a significant decrease in fat body ICWs in the co-culture experiment (Fig. 2F, G). Finally, the fat body of *AkhR* mutant larvae exhibited neither in vivo ICWs nor an ex vivo response to the AKH peptide (Fig. 2H–K, Supplementary Movie 3, 4).

AkhR is a GPCR that triggers both cAMP and $Ca^{2+}$[36]. AKH increases $Ca^{2+}$ by activating phospholipase C (PLC) and subsequent generation of $IP_3$[40]. Knocking down *Gαq*, which is responsible for the GPCR-dependent PLC activation, blocked the ICWs, suggesting that Gαq is an AKH-downstream effector (Fig. 2H, I). Meanwhile, *Gαq* overexpression triggered ICWs both in isolated fat bodies without AKH and in free-behaving larvae on a 5% sucrose diet (Supplementary Fig.4). Knocking down *SERCA*, a critical ER-localized $Ca^{2+}$ pump, also resulted in a sustained elevation of $Ca^{2+}$ in the fat body (Supplementary Fig. 4A–D). Altering fat body $Ca^{2+}$ via *Gαq-RNAi* or *SERCA-RNAi* led to a significant increase or decrease in TAG levels respectively, supporting that AKH regulates TAG catabolism through $Ca^{2+}$ (Supplementary Fig. 4E). Collectively, these findings demonstrate that *Gαq*-mediated $Ca^{2+}$ activity in the larval fat body is initiated by AKH secreted from APCs to regulate lipid homeostasis.

### Global and local ICWs form through distinct mechanisms

AKH triggers random local $Ca^{2+}$ waves in the cultured fat body but induces directional global organism-level ICWs propagating from the larval head to tail. There seems to be a single pacemaker in the head for the global ICWs. In contrast, local regional ICWs are initiated by multiple independent pacemakers. The mechanism underlying the different behaviors of ICWs in vivo and ex vivo is unknown (Fig. 3A). Previous studies showed that the ICWs are mediated either by gap junctions or extracellular ATP[32,41]. We added ATP to the cultured fat body and found no ICWs (Supplementary Fig. 5), ruling out ATP as a trigger. We then screened all major gap junction proteins by RNAi and found that knockdown of *Inx2* or *Inx3* significantly reduced the fat ICWs ex vivo (Fig. 3B, C, Supplementary Fig. 6A, B). Our observation is consistent with previous studies suggesting that Inx2 is predominantly required for gap junction function in the fly epithelium[23,41,42] and that Inx3 interacts with Inx2 to form a heterohexamer[43]. Both RNAi lines for *Inx2* and *Inx3* have been validated in multiple studies[42,44–46] and showed

similar phenotypes. Since *Inx2* knockdown exhibited the strongest phenotype, we used *Inx2-RNAi* to disrupt gap junctions in subsequent experiments.

*Inx2* knockdown blocked the local propagation of ICWs ex vivo (Fig. 3B, C, Supplementary Fig. 6C–F, Supplementary Movie 5). However, in vivo global ICWs were surprisingly unaffected by *Inx2-RNAi*. Both propagation speed and oscillation period of global ICWs in the *Inx2-RNAi* fat body are similar to the control, while random local ICWs were blocked by the *Inx2-RNAi* (Fig. 3D, E, Supplementary Fig. 7A). Ca²⁺ oscillations in *Inx2-RNAi* flies exhibited an unexpected increase in amplitude and a decrease in frequency (Supplementary Fig. 7B, D). Similarly, *Inx2* knockdown led to an elevation of average Ca²⁺ activity in the fat body of free-behaving larvae (Fig. 3F, G). To rule out the possibility of non-specific effects of *Inx2-RNAi*, we tested the effects of the

gap junction inhibitor carbenoxolone. Consistent with *Inx2-RNAi*, carbenoxolone treatment showed an increase in Ca²⁺ amplitude and a decrease in frequency (Supplementary Fig. 7E–G).

Despite these changes, our data demonstrate that the global ICWs in vivo do not require gap junctions. Meanwhile, both the directional global and the random local ICWs were completely abrogated in *AkhR* mutants, suggesting that both ICWs are triggered by AKH (Fig. 3D, Supplementary Movie 6). Thus, we propose that local ICWs are generated by diffusion of Ca²⁺ from cells stochastically activated by AKH, but that the global ICWs are generated in response to the secretion and diffusion of extracellular AKH from the APCs (Fig. 3A). Supporting this hypothesis, the directional propagation of the ICWs were disrupted into a random pattern after overexpression of AKH in the fat body (Fig. 3D, Supplementary Movie 6). Meanwhile, the global ICWs travel

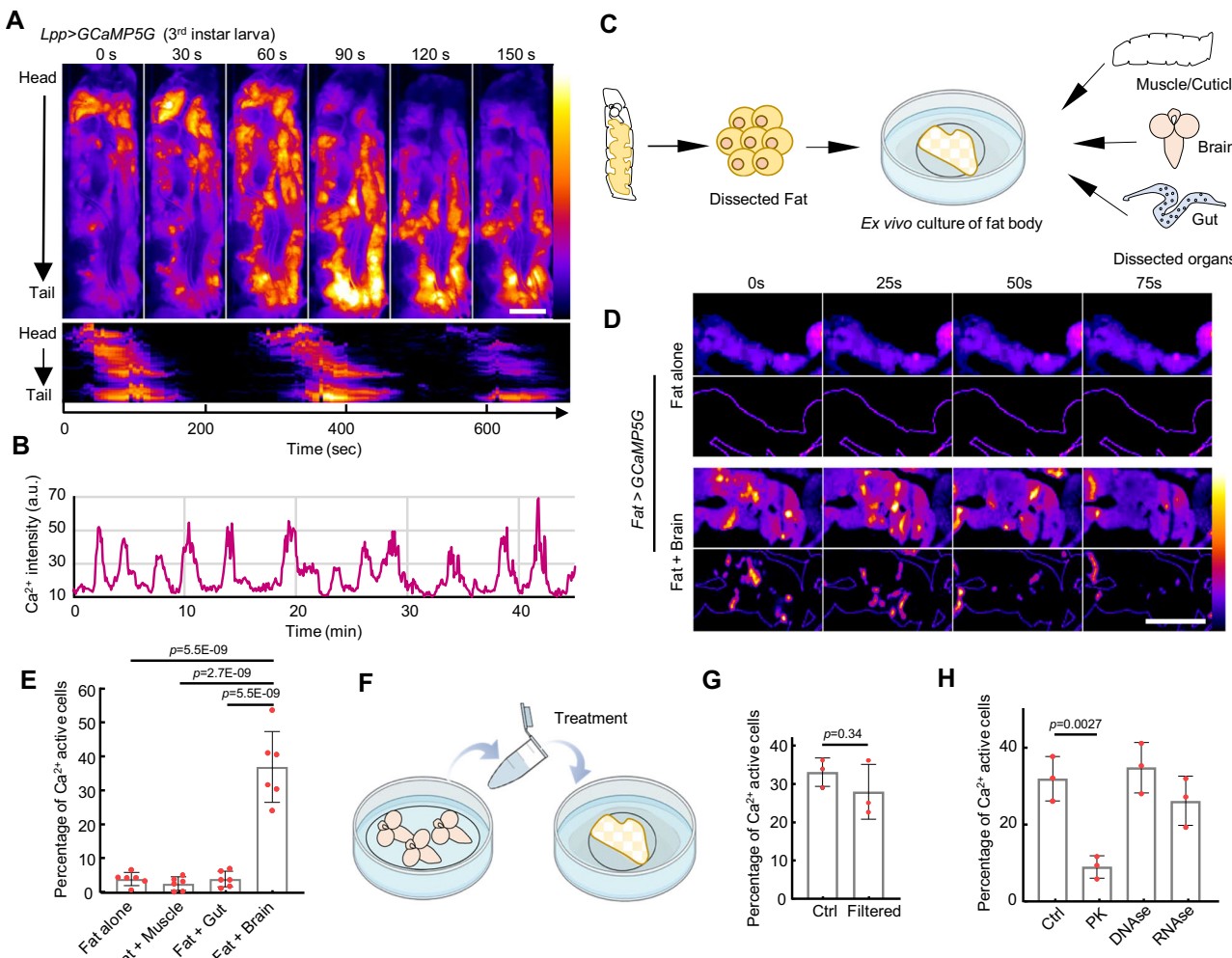

**Fig. 1 | Periodic Ca²⁺ waves are triggered in the fly larval fat body by a brain-derived factor. A** A 3rd instar larva was placed in a narrow glass channel to restrict its mobility. Fat body-specific Ca²⁺ activity was visualized using *Lpp-GAl4 > UAS-GCaMP5G*. A kymograph of Ca²⁺ waves traveling from head to tail is shown. The result was repeated in more than 5 independent experiments. **B** Representative Ca²⁺ activity in the larval fat body within the midsection. Ca²⁺ intensity was plotted in arbitrary units (a.u.). **C** Fat body expressing GCaMP5G was dissected and cultured in Schneider's *Drosophila* medium with or without different larval organs. **D** Fat body cultured alone showed little Ca²⁺ activity, while fat body cultured together with dissected brains showed prominent ICWs. The dynamic Ca²⁺ activities are highlighted in the lower panel by subtracting the constant background signal and showing the organ outline. The result was repeated in 6 independent experiments. **E** Quantification of Ca²⁺ activities in the cultured fat body under different conditions. *N* = 6, 6, 6, 6 fat bodies (left to right). **F** Brain-conditioned medium, prepared

by incubating 20 dissected larval brains of the third instar with 400 µl of Schneider's medium for one hour, was utilized to treat the isolated fat bodies. **G** Brain-conditioned medium before and after filtration triggered significant ICWs in cultured fat bodies. *N* = 3, 3 fat bodies. **H** Ca²⁺ activity triggered by proteinase K-digested brain-conditioned medium was significantly reduced compared to untreated brain-conditioned medium. Brain-conditioned medium was treated with 0.1 mg/mL proteinase K, DNAse I (1 U/mL), or RNAse A (1 µg/mL) for 1 h at 37 °C and then filtered with Pierce 10 K MWCO concentrator. The untreated brain-conditioned medium was processed similarly without the addition of proteinase K. *N* = 3, 3, 3, 3 fat bodies. Ca²⁺ intensity was presented using a linear colour scale (minimum = 0, maximum = 255) (**A**, **D**). Ordinary one-way ANOVA with Dunnett's multiple comparisons was utilized in (**E**, **H**). Two-tail unpaired Student's *t*-test was utilized in (**G**). Data were plotted as mean ± SD. Scale bars, 500 µm (**A**), 500 µm (**D**). Source data are provided as a Source Data file.

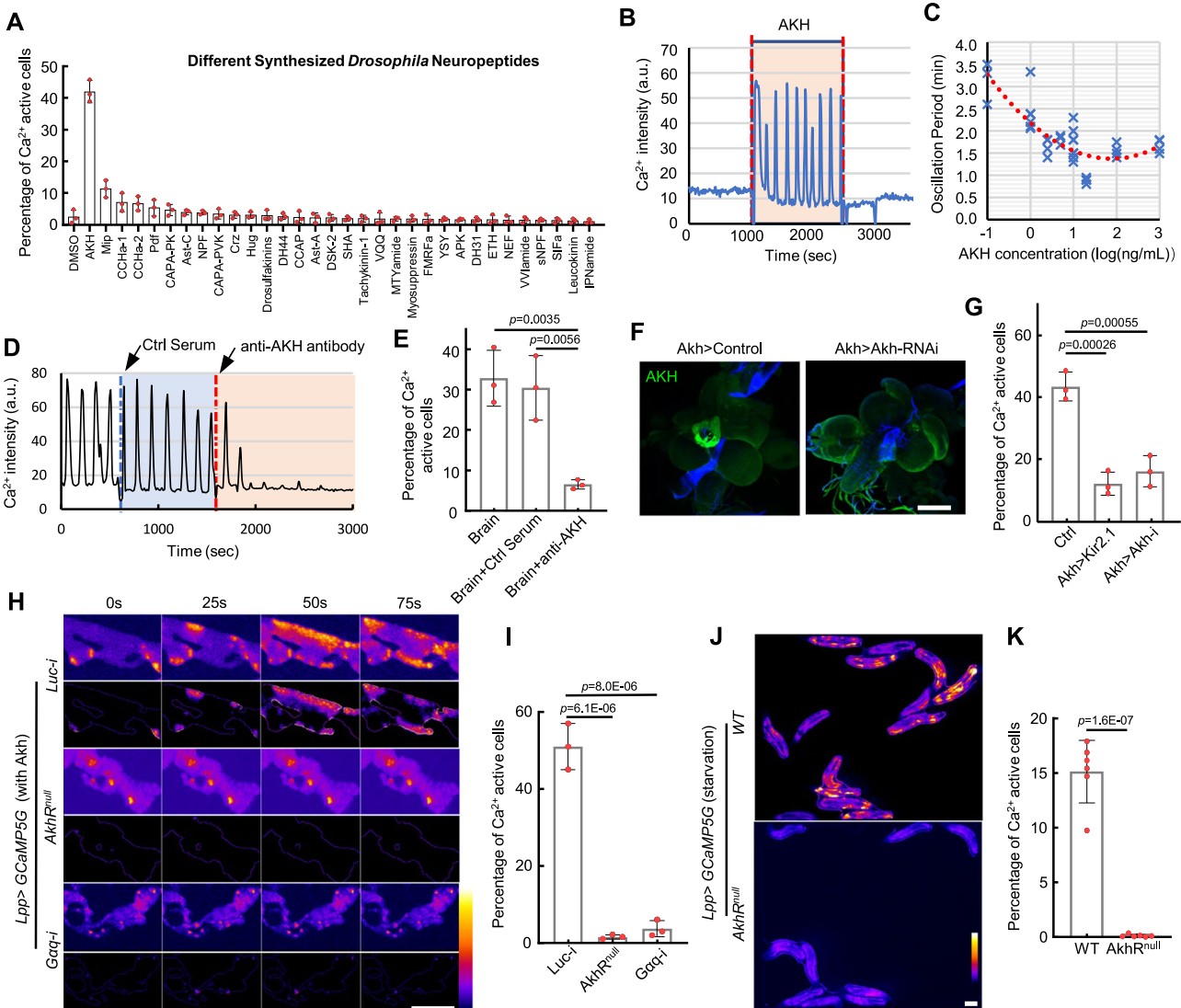

**Fig. 2 | Brain-derived AKH is responsible for the ICWs in the fat body.**
**A** Quantification of Ca²⁺ activities triggered by different synthesized fly neuro-peptides (0.1 ug/mL) in cultured larval fat body expressing GCaMP5G ($N = 3$ fat bodies per group). **B** Ca²⁺ activities were monitored in the cultured fat body. AKH (0.1 ug/mL) was added to the dissected fat body, incubated for 15 min, and then washed away. Representative trace of Ca²⁺ dynamics was collected from a single fat cell. **C** Relationship between the period of Ca²⁺ oscillation and concentration of applied AKH. **D**, **E** Ca²⁺ activity triggered by brain-conditioned medium was blocked by the addition of either a control rabbit serum or rabbit serum with anti-AKH antibody (1:100 dilution). Representative Ca²⁺ activity from a selected single fat body cell were plotted in **D**. $N = 3$, 3 fat bodies (**E**). **F** AKH was efficiently knocked-down in APCs by *Akh>Akh-RNAi*. The result was repeated in 3 independent experiments. **G** Conditional medium derived from brains with APCs silenced by

Kir2.1 or with *Akh* knockdown showed significantly reduced ability to trigger Ca²⁺ activity in the fat body. $N = 3$, 3, 3 fat bodies. **H**, **I** Cultured fat body with *AkhR* mutant or *Gαq* knockdown failed to respond to AKH. The dynamic Ca²⁺ activities are highlighted in the lower panel by subtracting the constant background signal and showing the organ outline. $N = 3$, 3, 3 fat bodies (**I**). **J**, **K** Ca²⁺ waves in free-behaving larvae were significantly reduced in *AkhR* mutant larvae. $N = 6$, 6 inde-pendent biological replicates (**K**). Ca²⁺ intensity was expressed as arbitrary units (a.u.) (**B**, **D**). Ca²⁺ intensity was presented using a linear colour scale (minimum = 0, maximum = 255) (**H**, **J**). Ordinary one-way ANOVA with Dunnett's multiple com-parisons test was used in (**E**, **G**, **I**), and unpaired two-tailed Student's *t*-test was used in (**K**). Data were plotted as mean ± SD. Scale bars, 200 μm (**F**), 500 μm (**H**), 1000 μm (**J**). Source data are provided as a Source Data file.

approximately 10 times faster than the local ICWs measured both in vivo and ex vivo (Fig. 3E). These data further support the notion that global ICWs are triggered by fast traveling extracellular AKH rather than by slow diffusion of messengers through gap junctions.

Previous studies showed that ICWs in fly imaginal discs are blocked after gap junction knockdown[22,23], we wondered if this gap junction-independent ICWs are stage-specific. We analyzed Ca²⁺ activity in the adult fat body and discovered that the cultured adult fat body exhibited gap junction-dependent local ICWs in response to AKH like the larval fat body (Fig. 3H, I, Supplementary Movie 7). However, no global ICWs were observed in the adult fat body in vivo, and the

in vivo ICWs in the adult fat body were significantly reduced with *Inx2-RNAi* (Fig. 3J, K, Supplementary Fig. 6E, F, Supplementary Movie 8), showing that the global ICWs are unique to the larval stage. We then tested if the hemolymph concentration of AKH after starvation was different between larvae and adults. However, the dot-blot assay shows that hemolymph AKH in starved larvae and adults are similar (Sup-plementary Fig. 8A). We also wondered whether the larval and adult fat body respond to AKH with different sensitivity. Treating cultured larval and adult fat bodies with varying concentrations of AKH showed similar levels of AKH sensitivity, within the 1–10 ng per mL range, which elicits ICWs comparable to those observed in vivo (Supplementary

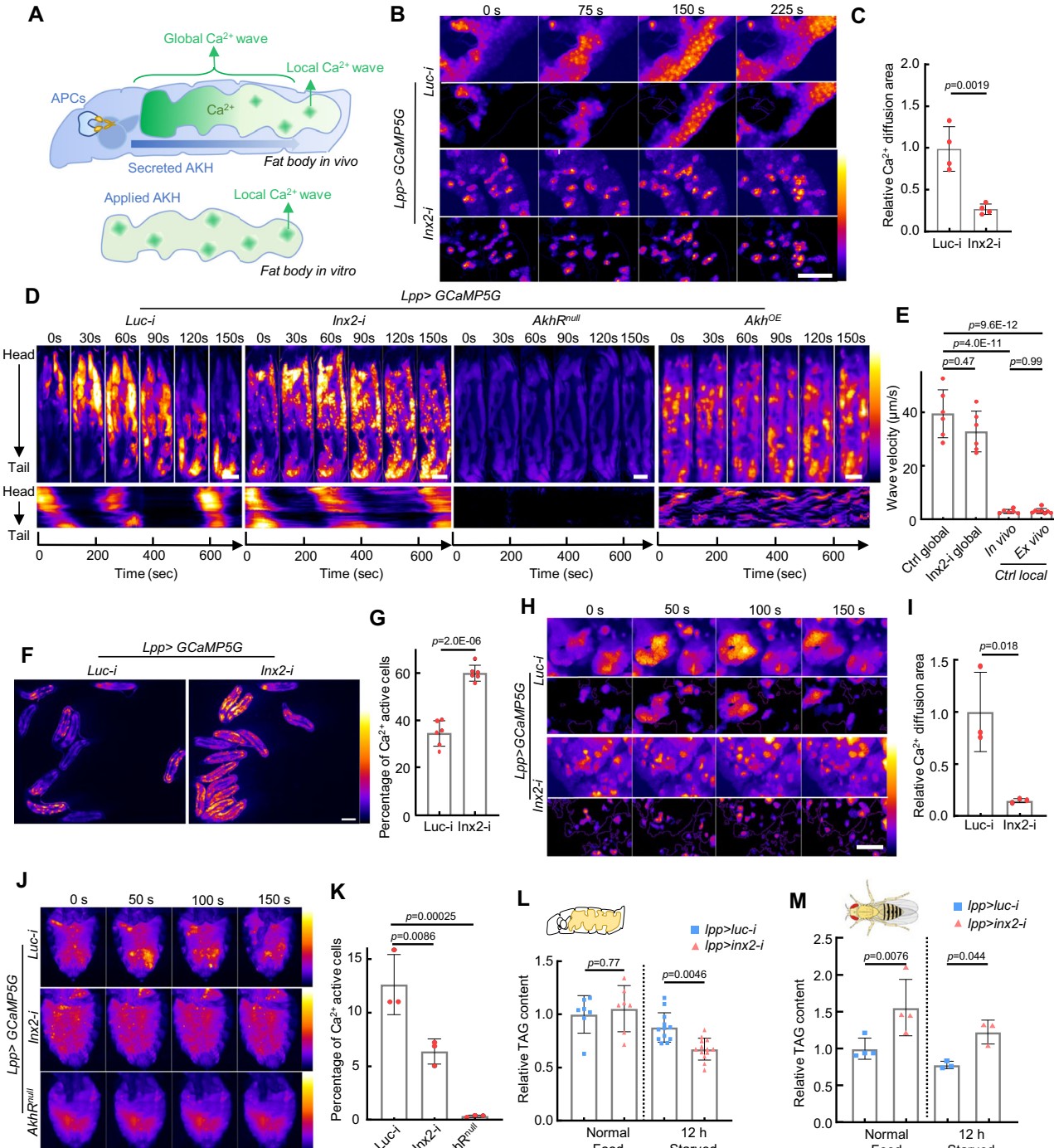

**Fig. 3 | Global and local ICWs are regulated by gap junction differently. A** Both global and local ICWs are present in the larval fat body. Organism-level global ICWs traveling from larval head to tail are only observed in vivo. Conversely, local ICWs, initiated and propagated randomly in the fat body cells, are evident in both in vitro and in vivo experimental settings. **B, C** Disruption of intercellular gap junctions by *Inx2* knockdown significantly scattered the Ca²⁺ activities in ex vivo cultured fat body. The dynamic Ca²⁺ activities were highlighted in the lower panel by removing the constant background signal. *N* = 4, 4 fat bodies (**C**). **D, E** *Inx2* knockdown does not affect the magnitude or period of global Ca²⁺ waves in vivo, while in *AkhR* mutants both global and local Ca²⁺ waves are completely blocked. Meanwhile, overexpressing AKH in the fat body led to a random propagation of Ca²⁺ waves. Larvae were kept under starved conditions to trigger the release of endogenous AKH. **E** Quantification of the traveling velocity of the global and local Ca²⁺ waves. *N* = 6, 6, 6 fat bodies in vivo, 8 fat bodies ex vivo. **F, G** The proportion of cells with Ca²⁺ activities in the fat body of free-behaving WT and *Inx2* RNAi larvae were quantified. *N* = 6, 6 independent biological replicates (**G**). **H, I** *Inx2* knockdown

completely scattered the Ca²⁺ waves in the dissected adult fly fat bodies. The dynamic Ca²⁺ activities were highlighted in the lower panel by removing the constant background signal. *N* = 3, 3 fat bodies (**I**). **J, K** Adult female flies of the indicated genotype were starved for about 30 min and then glued alive on a glass slide with the ventral abdomen imaged. Ca²⁺ activities indicated by fat body-specific expression of *GCaMP5G* were monitored and quantified. *N* = 3, 3, 3 adults (**K**). **L, M** Effects of gap junction disruption in fat bodies during the larval and adult stages on TAG metabolism. *N* = 7, 7, 11, 13 independent biological replicates (**L**). *N* = 4, 4, 3, 3 independent biological replicates (**M**). Ca²⁺ intensity (a.u.) was presented using a linear colour scale (minimum = 0, maximum = 255) (**B, D, F, H, J**). Unpaired two-tailed Student's *t*-test was used in (**C, G, I**). Ordinary two-way ANOVA with Sidak's multiple comparisons was used in (**E, L**). Ordinary one-way ANOVA with Dunnett's multiple comparisons test was used in (**K**). Ordinary two-way ANOVA with uncorrected Fisher's LSD was used in (**M**). Data were plotted as mean ± SD. Scale bars, 200 μm (**B, H**), 500 μm (**D, J**), 1 mm (**F**). Source data are provided as a Source Data file.

Fig. 8B). Collectively, these data suggest that the differential responses of larval and adult fat bodies to AKH may be attributed to differences in the temporal dynamics of AKH ligand.

As the increase of AKH and cytosolic $Ca^{2+}$ have been linked with lipolysis in adipose tissues[8,9,36,47], we assessed the effect of gap junction disruption on TAG catabolism under normal feeding and starvation conditions. TAG levels in *Inx2-RNAi* larvae remained unaffected under standard feeding conditions, as AKH secretion is not induced (Fig. 3L). However, under starvation conditions, TAG levels in gap junction knockdown larvae were significantly lower compared to control larvae (Fig. 3L). This is consistent with our observation that gap junction blockage increases AKH-induced $Ca^{2+}$ activities in larvae. In contrast, adult flies with disrupted gap junctions in the fat bodies displayed a significant increase in TAG accumulation under normal diet and starvation conditions (Fig. 3M), consistent with the reduction of $Ca^{2+}$ in the adult fat body. Thus, the different responses of larval and adult fat bodies to gap junction disruption likely reflect the presence of global ICWs in larvae.

### Global ICWs are triggered by the periodic release of AKH in hemolymph

To test whether global ICWs are triggered by the transport of extracellular AKH released from the head region, we used chloroform to transiently inhibit the larval heartbeat (Fig. 4A, B). Stopping the heartbeat completely prevented the propagation of global ICWs, and only the larval head region showed periodic increases of $Ca^{2+}$ (Fig. 4C, Supplementary Movie 9). Notably, the global ICWs showed a much longer period (~400 s) than the intrinsic $Ca^{2+}$ oscillation period in the fat body (~200 s), supporting that the period of global waves is controlled by a pulsatile diffusion of AKH rather than the intrinsic property of the fat body.

Additionally, the propagation speed of the global ICWs is around $40 \, \mu m \, s^{-1}$, which is over 10 times faster than the speed of gap-junction mediated local ICWs (~$2.5 \, \mu m \, s^{-1}$). This speed also surpasses the free diffusion velocity of a 1 kDa molecule (equivalent to the size of AKH) in water by approximately two orders of magnitude[48] (Fig. 3E), suggesting that AKH is transported by the circulating hemolymph. However, the flow characteristics of the fly hemolymph have not been measured before. Thus, we injected fluorescent polystyrene beads into the 3rd instar larvae to estimate the hemolymph flow speed (Fig. 4D, Supplementary Movie 10). By tracing the beads, we found that the anterograde (head-to-tail) speed of hemolymph is about $250 \, \mu m \, s^{-1}$ and that the retrograde (tail-to-head) flow speed in the heart tube is nearly $10,000 \, \mu m \, s^{-1}$ (Fig. 4E). This high-speed hemolymph circulation is fast enough to facilitate the transport of secreted AKH with a speed beyond free diffusion. It is worth noting that the beads interact with the organs, as some beads become stuck in the tissue. Therefore, the actual speed of the hemolymph could be even higher. Meanwhile, the hemolymph flow in adult flies was primarily retrograde outside the adult heart tube, with an occasional reverse flow direction as previously reported[49] (Supplementary Fig. 9). The APCs are located in the anterior region of the adult thorax[50], we speculate that the change in circulation direction of the adult hemolymph may make the transport of secreted AKH less efficient and thus be responsible for the absence of global ICWs.

Moreover, the circulating AKH model also implies that the release of AKH must have a short half-life compared with the 300 s period of the global ICW, otherwise, AKH should accumulate in the hemolymph and trigger continuously random ICWs as observed in the ex vivo culture. To test this hypothesis, we need to block the release of AKH in APCs and monitor the decrease of $Ca^{2+}$ in the fat body as a read-out of circulating AKH in vivo. I found that anesthetizing the larvae with carbon dioxide ($CO_2$) rapidly suppressed the APC neurons, while the whole brain activity was largely unaffected (Fig. 4F–H, Supplementary Fig. 10A–D). As we expected, the ICWs in the larval fat body rapidly returned to baseline within ~70 s after $CO_2$ administration, implying

that the in vivo functional half-life of AKH is ~35 s (Fig. 4I–L). Meanwhile, the $CO_2$ treatment does not affect the AKH-triggered ICWs in the cultured fat body (Supplementary Fig. 10E, F). Larvae overexpressing AKH in the fat body had significant ICWs remaining after $CO_2$ treatment, suggesting that $CO_2$ does not block the fat body response to AKH (Fig. 4I–L).

### Computational modeling of the global and local ICWs
With the measured dynamic parameters of ICWs and AKH, we applied a receptor-operator calcium model to gain insights into the dynamics of the ICWs in both larvae and adults. The model incorporates three key elements: intracellular $Ca^{2+}$ dynamics, intercellular signaling via gap junctions, and tissue-level AKH transport. The cytoplasmic $Ca^{2+}$ level is governed by a receptor-operator calcium channel model[22,51]. When AKH binds to AkhR, it activates a downstream inositol trisphosphate (IP3), triggering a rapid $Ca^{2+}$ efflux from the endoplasmic reticulum (ER) via positive feedback and SERCA activation pumping $Ca^{2+}$ back into the ER as a slow negative feedback. Intercellular $Ca^{2+}$ signaling through gap junctions is described by diffusion proportional to the cytoplasmic $Ca^{2+}$ concentration difference between neighboring connected cells[22]. Both AKH binding to AkhR and the $Ca^{2+}$ flow through gap junctions trigger the intracellular fast-activation-slow-inhibition $Ca^{2+}$ cycles. The AKH diffusion and transport are described by a reaction-diffusion-advection equation, which accounts for the periodic AKH release from the head region and uniform degradation in the larvae. In adult flies, the global AKH pulses are absent, and the model only considers AKH diffusion with fluctuations. This distinction accounts for potential differences in extracellular AKH dynamics between larvae and adults. Model details and parameters used can be found in Supplementary Methods, and Supplementary Table 3.

The simulations successfully replicated the observed differences in wave characteristics between larvae and adults, and the impact of gap junction knockdown on $Ca^{2+}$ (Fig. 5A–F). Both computational simulations and experimental data support that the directional transport of AKH through the lymphatic circulation is responsible for the global ICWs, while the trigger wave mechanism through gap junctions is responsible for the local ICWs[52] (Fig. 5A–F, Supplementary Movie 11,12). Meanwhile, the modeling also showed similar changes in average $Ca^{2+}$ intensity and peak width in individual fat cells in fat bodies with *Inx2-RNAi* (Fig. 5G, H), supporting that the modeling can recapitulate the dynamics of $Ca^{2+}$ activities.

Interestingly, our modeling implies that the excitable $Ca^{2+}$ cycles are sensitive to both external and internal noises, such as fluctuations in AKH concentration or intracellular signaling molecules. These fluctuations have the potential to cause spontaneous $Ca^{2+}$ signaling randomly, leading to asynchronous firing between individual cells. However, gap junctions help coordinate these excitation processes, resulting in spatially synchronized oscillations. We also quantified the spatial coordination of $Ca^{2+}$ activities by calculating the normalized variance of $Ca^{2+}$ in the wavefront region (Fig. 5I, J). The modeling showed that as global ICWs propagate from head to tail, the normalized variance remains low for WT larvae but increases sharply in larvae with gap junction knockdown. Notably, the experimental data also exhibited a similar increase of variance in the tail region of the *Inx2-RNAi* larvae but not in the WT larvae (Fig. 5I, J). This finding reinforces the relevance of our modeling results (Fig. 5K).

### Regulation of AKH release from APCs by amino acids
The specific and rapid response of the fat body to AKH provides a direct readout for monitoring real-time AKH secretion in free-behaving larvae. As most live-imaging setups used immobilized or anesthetized larvae, which may cause undesirable distress or artefacts, we decided to study metabolic signaling in free-behaving animals. As AKH is released upon starvation[3,5,33,47], early 3rd instar larvae were first starved for 9 h to cleanse their digestive systems, and then fed on 2% sucrose

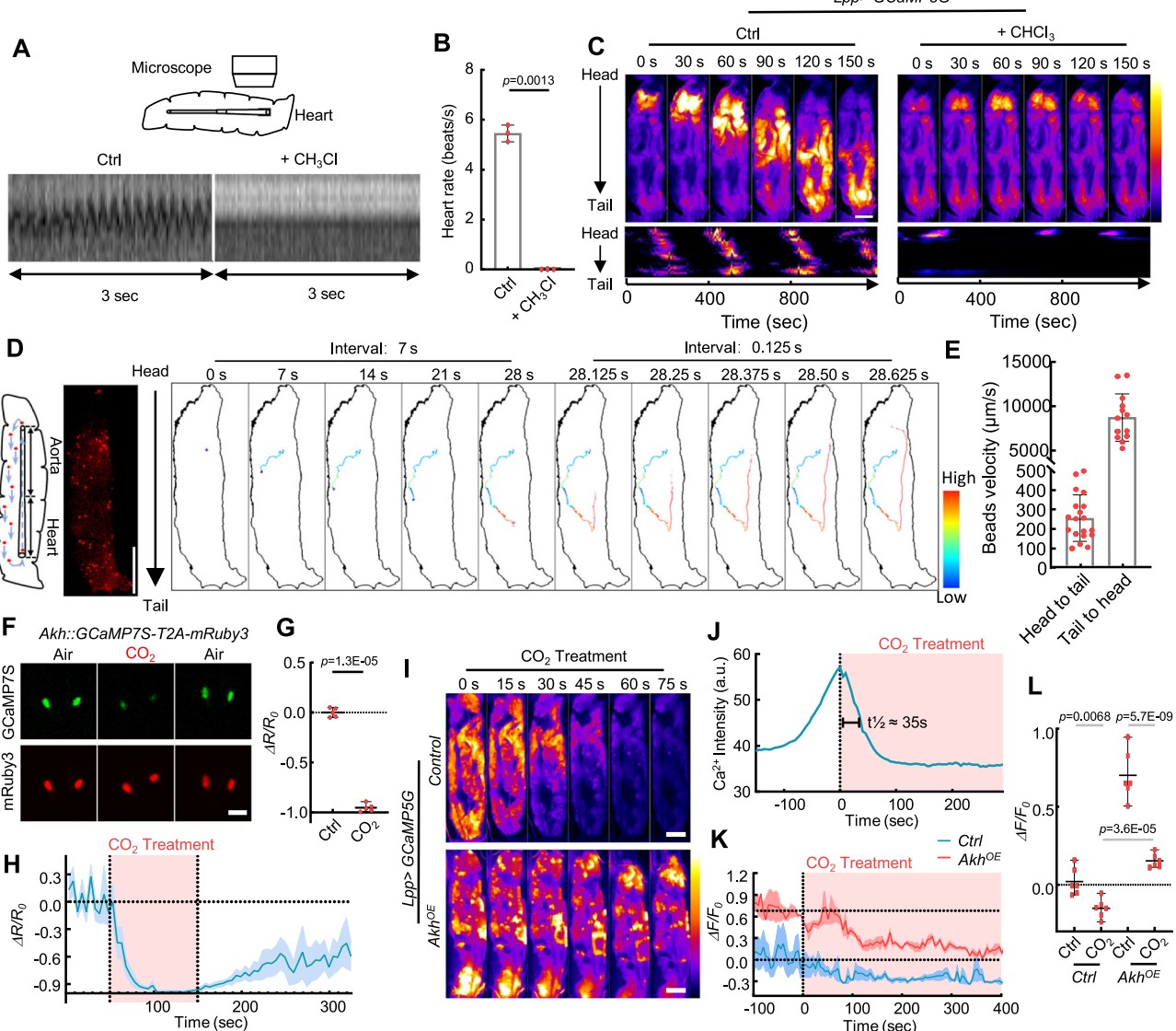

**Fig. 4 | The global ICWs are generated by the periodic release of AKH in the circulating hemolymph. A**, **B** 3rd instar larvae were immobilized on the glass slide and imaged before and after the application of chloroform. Filter paper containing ~10 ul chloroform was placed close to the larvae, which temporarily stopped the heartbeat of the larvae for about 30 min. Data was pooled from 3 independent experiments. $N = 3$, 3 larvae (**B**). **C** $Ca^{2+}$ imaging of the larvae treated with and without chloroform. The result was repeated in 3 independent experiments. **D** Fluorescent beads were injected into the 3rd instar larvae and traced under a microscope. The trajectory of one traced bead was shown with velocity coded in color. The result was repeated in 3 independent experiments. **E** The velocities of beads transported in the anterograde direction (from head to tail) and retrograde direction (from tail to head) were quantified. Data was pooled from 3 independent experiments. $N = 19$, 13 beads. **F**–**H** 3rd instar larvae were immobilized on a glass slide which is placed at the center of a $CO_2$ fly anesthetizing pad. Neuron activity in the APCs was revealed using *Akh::GCaMP7S-T2A-mRuby3*. $Ca^{2+}$ activity before and after $CO_2$ application was quantified. Time courses of average $Ca^{2+}$ signal from multiple larvae were plotted as mean ± S.E.M. (shaded region). Data was pooled from 5 independent experiments. $N = 5$, 5 larvae (**G**). **I**–**L** The $Ca^{2+}$ waves of fat bodies are rapidly ($t\frac{1}{2} \approx 35$ s) and completely blocked by $CO_2$ treatment, and the fat bodies of Akh overexpression still present $Ca^{2+}$ waves in the treatment of $CO_2$. A representative trace of average $Ca^{2+}$ intensity from a $CO_2$ treated larva was plotted in **J**. $Ca^{2+}$ intensity was expressed as arbitrary units (a.u.) (**J**). Time courses of average $Ca^{2+}$ intensity from multiple larvae were plotted as mean ± S.E.M. (shaded region) in (**K**). Data was pooled from 6 independent experiments. $N = 6, 6, 6, 6$ larvae (**L**). $Ca^{2+}$ intensity (a.u.) was presented using a linear colour scale (minimum = 0, maximum = 255) (**C**, **I**). Paired two-tailed Student's $t$-test is utilized in (**B**, **G**), and ordinary two-way ANOVA with uncorrected Fisher's LSD is utilized in (**L**). Data were plotted as mean ± S.E.M (**H**, **K**), or mean ± SD (others). Scale bars, 500 μm (**C**, **I**), 1000 μm (**D**), 50 μm (**F**). Source data are provided as a Source Data file.

for 20 min, allowing the fat body $Ca^{2+}$ activities and AKH to return to a basal level (Fig. 6A). Subsequently, these larvae were transferred onto 2% agarose plate with 2% sucrose, 2% sucrose plus 10% Tryptone (protein-rich), or 2% agarose only (starvation)[53]. Larvae on a starvation diet displayed a significant increase of ICWs after ~10 min of feeding, indicating a swift response to starvation. Importantly, consumption of the protein-rich diet also increased $Ca^{2+}$ activities in the fat body within 10–15 min (Fig. 6B, C, Supplementary Movie 13), suggesting that AKH secretion is also triggered by amino acids. Larvae with *AkhR* mutation

or fat-body specific *AkhR-RNAi* showed a significant reduction of the $Ca^{2+}$ under both starvation and protein-feeding conditions, supporting that both processes depend on AKH (Fig. 6D, E).

The fat body $Ca^{2+}$ increase suggests that AKH release from APCs is triggered by amino acids. To test this, we used *Akh-Gal4 > GCaMP5G-T2A-mRuby3* to visualize $Ca^{2+}$ activity in the APCs with an Extended-Depth-of-Field (EDoF) microscope, which turns slow 3D imaging into a quick 2D acquisition (Supplementary Fig.11A–D). 1st instar larvae were starved for 6 h on 2% agarose, then transferred to new agarose plates

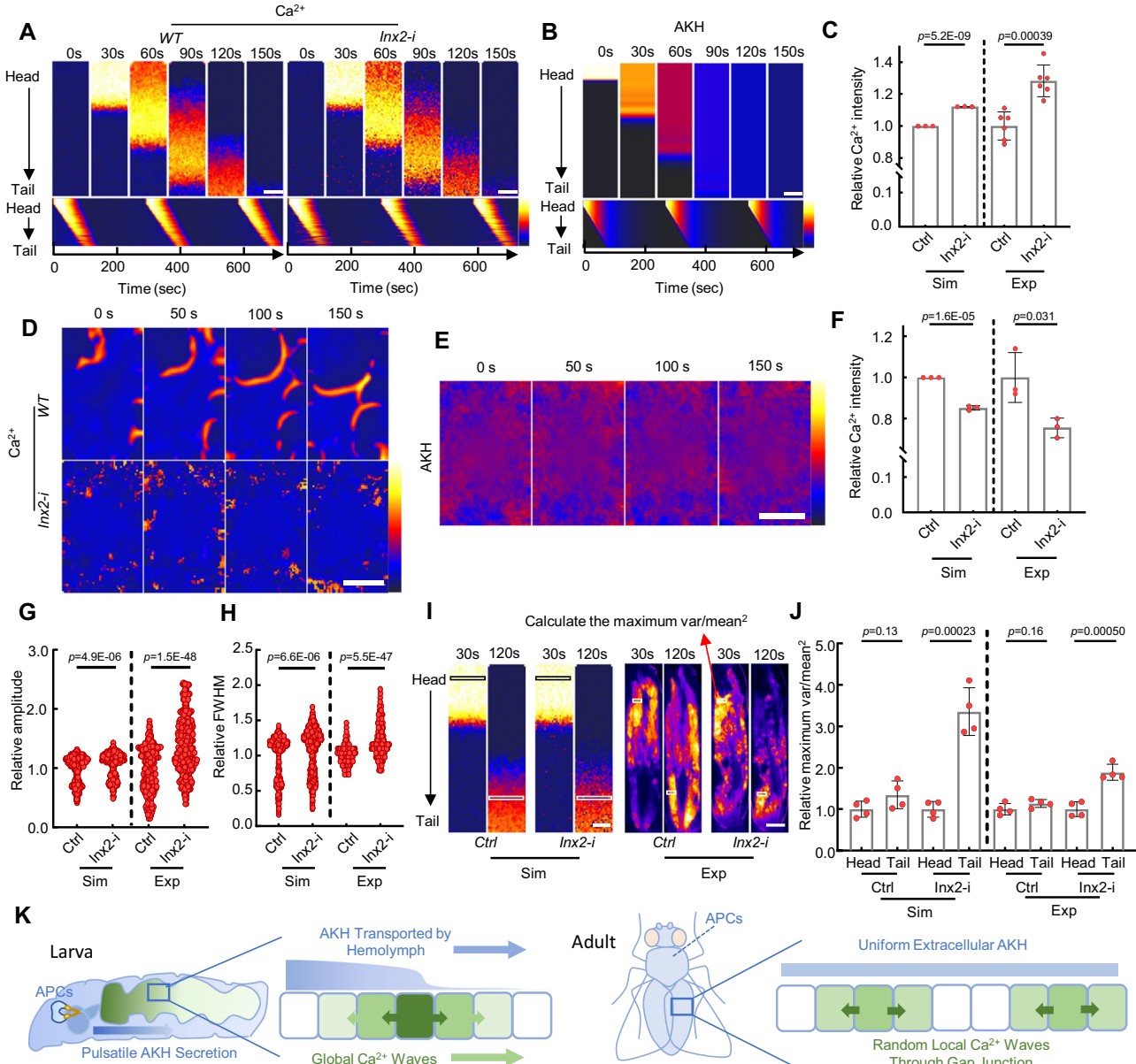

**Fig. 5 | Computational modeling of the ICWs in larval and adult flies.**
**A** Simulated global ICWs in the fat body of WT and *Inx2-RNAi* larvae. The result was repeated in 3 independent simulations. **B** The simulated concentration of AKH after release from the APC at the head of the larva. The result was repeated in 3 independent simulations. **C** Quantification of the relative Ca²⁺ intensity in the entire fat body using simulated (Sim) and experimental (Exp) data. The variance in the simulated data originates from the incorporation of random Ca²⁺ fluctuation into the model. *N* = 3, 3 simulated fat bodies (Sim), *N* = 6, 6 experimental fat bodies (Exp). **D** Simulated ICWs in the fat body of WT and *Inx2-RNAi* adult flies. The result was repeated in 3 independent simulations. **E** The simulated concentration of AKH with stochastic fluctuation in adult hemolymph. The result was repeated in 3 independent simulations. **F** Quantification of the relative Ca²⁺ intensity in the entire fat body using simulated (Sim) and experimental (Exp) data. *N* = 3, 3 simulated fat bodies (Sim), *N* = 3, 3 experimental fat bodies (Exp). **G, H** Quantification of the relative amplitude and FWHM (full width at half maximum) of the Ca²⁺ signal in individual fat cells from WT or *Inx2-RNAi* flies using simulated (Sim) and experimental (Exp) data. Data was pooled from 3 independent simulations and 3

independent experiments. *N* = 511, 511, 511, 511 oscillations (**G**), *N* = 529, 529, 465, 260 oscillations (**H**). **I–J** The correlation of Ca²⁺ activities along the wavefront was quantified by dividing the maximum variance of Ca²⁺ intensity by the square of the mean intensity. The relative maximum variations at the head and tail region were compared in both WT and *Inx2-RNAi* flies. Data was pooled from 4 independent simulations and 4 independent experiments. *N* = 4, 4, 4, 4 simulated fat bodies (Sim), *N* = 4, 4, 4, 4 experimental fat bodies (Exp). **K** A schematic model of the ICWs in larvae and adult flies. The strong exocellular AKH pulse synchronized the global ICWs in a larva, which renders the signal diffusion through the gap junction less important. In contrast, in adult fly, the effective AKH is probably more uniform and triggers gap-junction-dependent random ICWs. Ca²⁺ intensity (arbitrary units) was presented using a linear colour scale (minimum = 0, maximum = 255) (**A, D**), and AKH concentration (μM) was presented using a linear colour scale (minimum = 0, maximum = 100) (**B**) and a linear colour scale (minimum = 0, maximum = 30) (**E**). Unpaired two-tailed Student's *t*-test was used in (**C, F, G, H, J**). Data were plotted as mean ± SD. Scale bars, 500 μm. Source data are provided as a Source Data file.

containing 5% sucrose (sugar), 10% Tryptone (protein-rich), or 2% agarose only (starvation) (Fig. 7A). As expected, Ca²⁺ in the APCs significantly decreased after being fed on 5% sucrose (Fig. 7B–G, Supplementary Movie 14). Moreover, Ca²⁺ in the IPCs was reduced when we

performed simultaneous dual labeling of both the APCs and IPCs (Supplementary Fig. 11E–G). This observation of the counter activities of the IPCs and APCs supports the reliability of this live-imaging system. Next, we found that the 10% tryptone diet triggers a Ca²⁺ increase

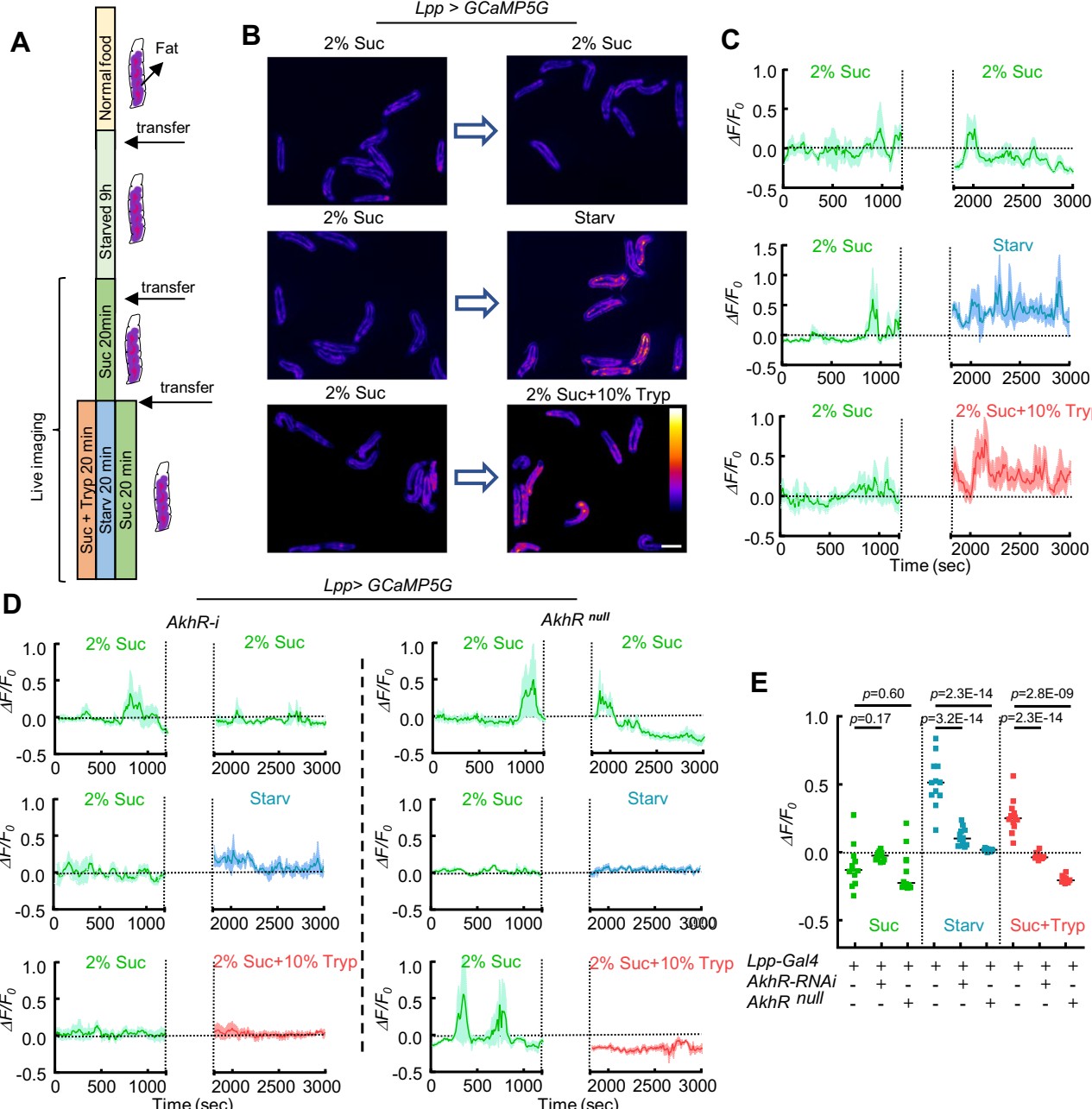

**Fig. 6 | AKH-dependent ICWs are regulated by dietary sugar and protein. A** Early 3rd instar fly larvae were transferred onto 2% agarose for 9 h and then onto 2% agarose + 2% sucrose for 20 min. These conditioned larvae were further transferred onto different test foods containing 2% sucrose, 2% sucrose + 10% Tryptone (protein diet), or no nutrient (starvation diet). **B**, **C** Representative Ca²⁺ images and Ca²⁺ activities of free-behaving larvae transferred between different diets. Ca²⁺ activities were calculated from the average signals obtained from all imaged larvae. The gap between the two diets is the larvae transfer time which is ~10 min. The result was repeated in 3 independent experiments. **D** Representative Ca²⁺ activities in the fat body of free-behaving larvae with whole-body *AkhR* mutant or fat-specific knockdown of *AkhR*. **E** Quantification of Ca²⁺ activities in the fat body of free-behaving fly larvae. $N = 12$ larvae for each experiment. Ca²⁺ intensity (arbitrary units) was presented using a linear colour scale (minimum = 0, maximum = 255) (**B**). Ordinary one-way ANOVA with Dunnett's multiple comparisons was used in (**E**). Data were plotted as mean ± S.E.M (**C**, **D**), or mean ± SD (**E**). Scale bars, 2 mm (**B**). Source data are provided as a Source Data file.

in the APCs after ~10 min of feeding, which is consistent with the quick Ca²⁺ response observed in the fat body (Fig. 7B–G).

Studies in mammals show that only some amino acids trigger glucagon release, with branch-chained amino acids failing to induce such secretion[54]. Thus, we tested which amino acids may activate AKH-mediated Ca²⁺ waves in ex-vivo fat body tissues. As the Schneider's medium contains all amino acids and triggers the release of AKH from dissected larval brains, we prepared a basal *Drosophila* HL6 buffer devoid of amino acids (hereinafter referred to as HL6 (AA-) buffer).

Each amino acid (5 mM) was individually added to the HL6 (AA-) buffer to assess its capacity to induce AKH release (Fig. 8A). Brain-conditioned HL6 (AA-) buffer does not activate the cultured fat body; however, the addition of most polar amino acids triggered AKH secretion and subsequent elevation of fat body Ca²⁺ (Fig. 8B, C, Supplementary Movie 15). Notably, the branch-chained amino acids leucine (Leu) and isoleucine (Ile) failed to trigger AKH release, akin to their effects on the mammalian glucagon system. We further monitored Ca²⁺ activities in APCs in isolated larval brains. APCs responded to

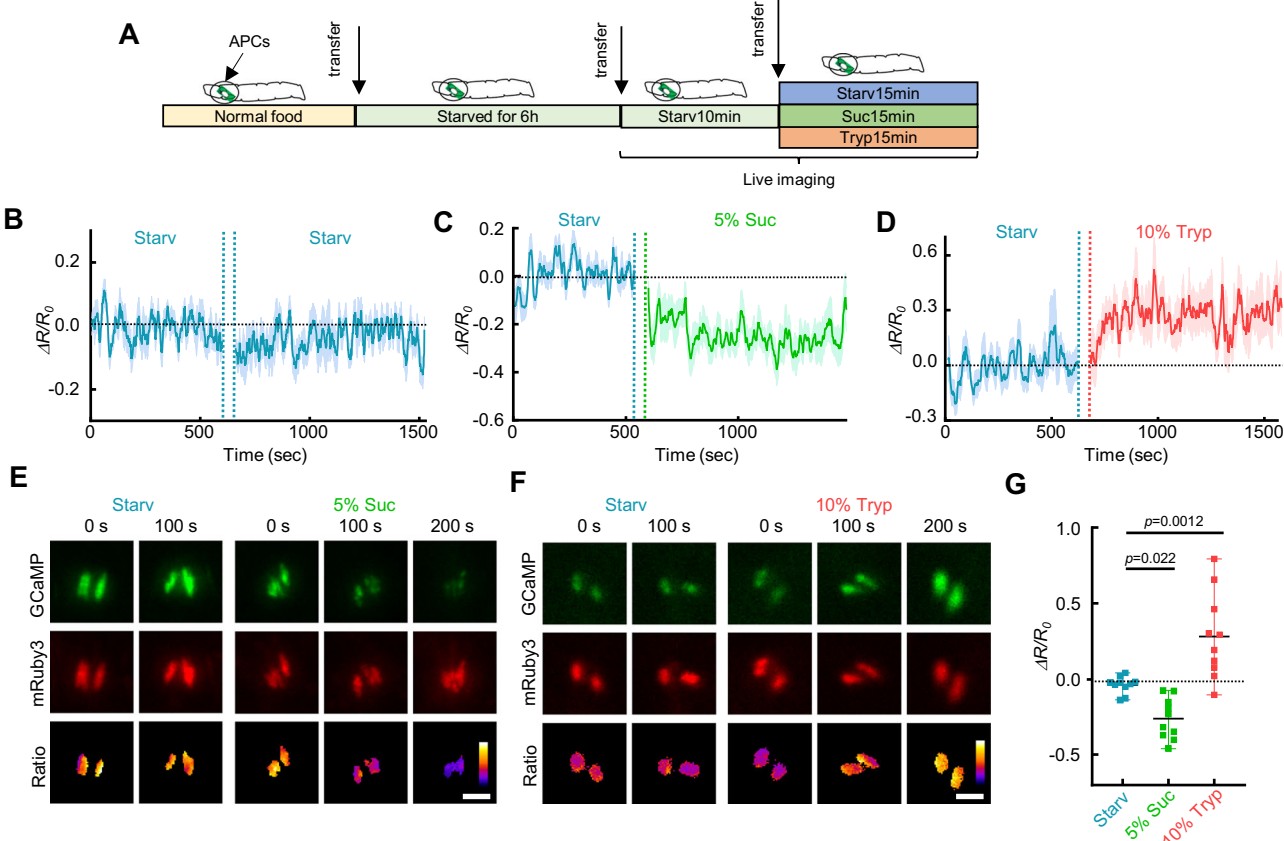

**Fig. 7 | AKH secretion from APCs is regulated by amino acids. A** 1st instar fly larvae were transferred onto 2% agarose for 6 h and then onto different test foods containing 5% sucrose, 10% Tryptone (protein diet), or no nutrient (starvation diet). **B–D** Representative Ca²⁺ activities in the APCs of free-behaving larvae transferred between different diets. Time courses of average Ca²⁺ signal from multiple larvae were plotted as mean ± S.E.M. (shaded region). The gap between the two diets is the larvae transfer time which is ~1 min. **E, F** Representative Ca²⁺ images of the APCs of

free-behaving larvae fed on different diets. **G** Quantification of Ca²⁺ activities in the APCs of free-behaving larvae. $N = 10, 10, 10$ larvae. Each dot represents an independent biological replicate. Ca²⁺ intensity (arbitrary units) was presented using a linear colour scale (minimum = 0, maximum = 255) (**E, F**). Ordinary one-way ANOVA with Dunnett's multiple comparisons was used in (**G**). Data in (**B–D**) were plotted as mean ± S.E.M. Data in were plotted as mean ± S.E.M (**B–D**), or mean ± SD (**G**). Scale bars, 25 μm (**E, F**). Source data are provided as a Source Data file.

threonine (Thr) within two minutes and reached an activation plateau at around six minutes, consistent with the response speed observed in vivo. In contrast, APCs showed little response to Leu (Fig. 8D–G). The release of AKH from APCs was also confirmed by AKH staining of APCs in 3rd instar larvae (Supplementary Fig. 12). Next, we tested 36 h of feeding on 5% sucrose, 5% sucrose plus 40 mM methionine (Met), or 5% sucrose plus 40 mM Leu (Fig. 8H–J). Met feeding significantly reduced the AKH signal in the APCs compared to the sugar control and Leu, suggesting that Met triggers the release of AKH in vivo. Previous studies have shown that increased AKH levels lead to lipolysis in fat body tissues, especially in adult flies[47]. Indeed, 40 mM Met feeding reduced the TAG content in adult flies, and this reduction was entirely abrogated in *AkhR* mutant animals (Fig. 8K). Altogether, our results show that specific dietary amino acids are sensed by APCs to trigger AKH release, which in turn activates ICWs in the fat body.

## Discussion

Canonically, ICWs were considered to spread through tissues via gap-junctions[23,31,41,55]. However, we delineate a gap-junction independent mechanism in which AKH release from the brain actively diffuses through the circulating fly lymph, which in turn results in organ-level ICWs. These gap junction-independent Ca²⁺ waves present a model illustrating how a hormone extracellularly orchestrates collective moving patterns across a tissue. These tissue-level ICWs also suggested that AKH is released from the larval head region periodically. However, Ca²⁺ imaging of the APCs both in vivo and ex vivo showed no significant

pulsatile activities. This discrepancy may be due to the limited temporal resolution of Ca²⁺ imaging in vivo and suboptimal ex vivo cultural conditions. In mammals, glucagon and insulin are released in a pulsatile manner, which is disrupted in patients with type-2 diabetes, contributing to hyperglucagonemia[56,57]. However, the biological significance of this pulsatile hormone release compared to continuous release is unclear. Our study suggests that in the fly larva, a potential pulsatile AKH creates a strong increase of hormone "shock" that collectively activates fat body cells independent of gap junctions.

Meanwhile, the biological significance behind the different ICWs between larvae and adults remains an open question. Interestingly, a recent study revealed that adult flies exhibit intricate interorgan connections (both in terms of organ shape and physiological arrangement)[58–60]. And the adult hemolymph is notably viscous, potentially hindering its flow through the tightly interconnected spaces within the adult's body cavity[58]. In contrast, the AKH-triggered global ICWs in larvae suggest that hemolymph flows efficiently, facilitating the rapid transport of secreted AKH from the head to the tail. Therefore, a potential increase in the intricacy of interorgan connections in adult flies may impede the swift transport of AKH, leading to a more uniform concentration of the hormone in the abdomen. Further investigation into these mechanisms may provide valuable insights into the physiological distinctions between the larval and adult stages.

Mathematical simulations suggest that the inclusion of random fluctuation is essential to recapitulate the Ca²⁺ wave properties in larval fat bodies with *Inx2-RNAi*. The increase of Ca²⁺ after Inx2 knockdown

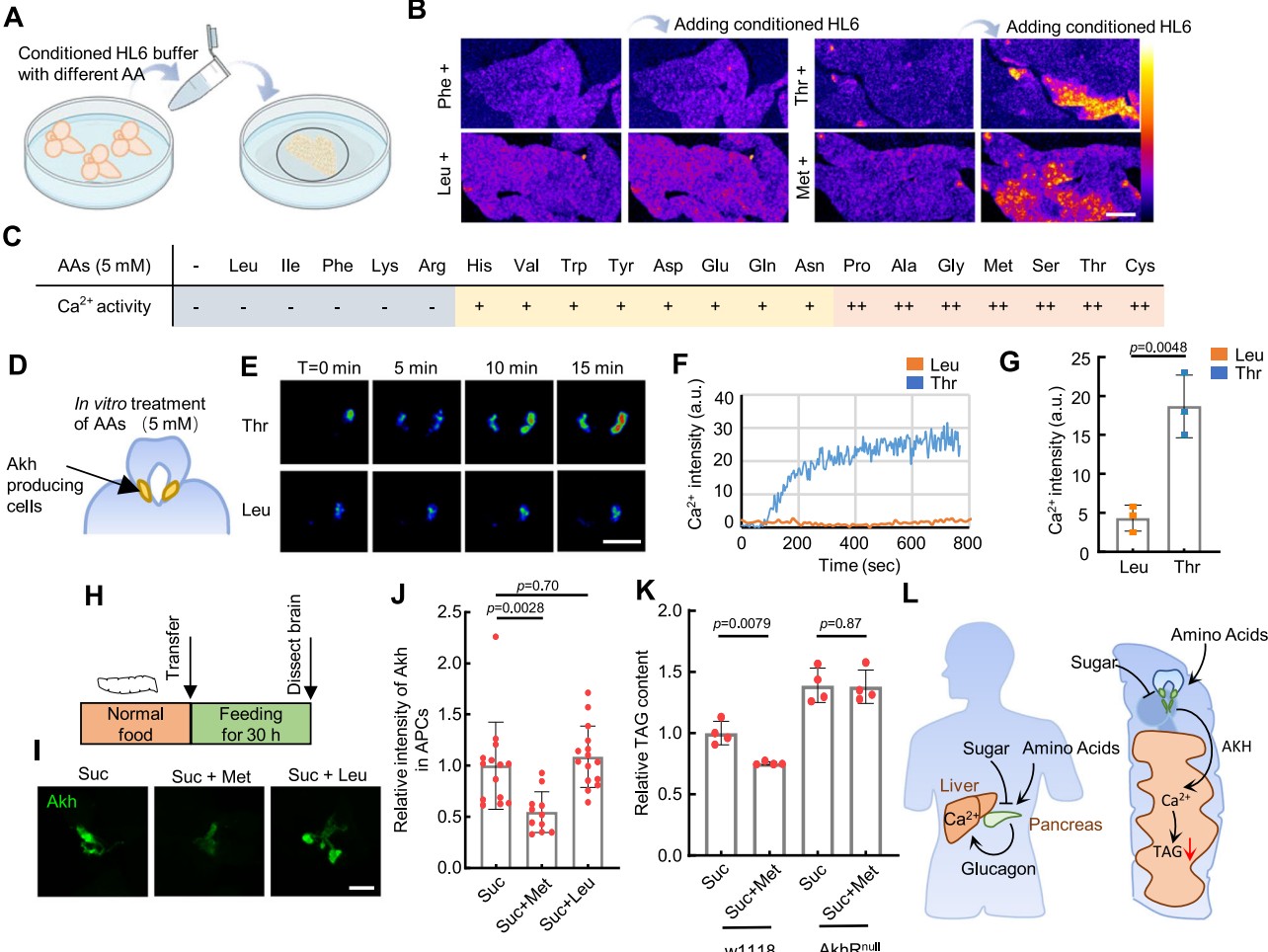

**Fig. 8 | AKH secretion from APCs is regulated by amino acids. A** Brain-conditioned HL6 media with each amino acid (5 mM) were generated and transferred to the dissected 3rd instar larval fat body. **B** Representative Ca²⁺ images of fat bodies before and after the addition of brain-conditioned HL6 media with indicated amino acids. Ca²⁺ intensity (a.u.) was presented using a linear colour scale (minimum = 0, maximum = 255). The result was repeated in 3 independent experiments. **C** Ca²⁺ activities in the bodies treated by brain-conditioned HL6 medium with indicated amino acids. Percentage of fat body cells with positive Ca²⁺ signal was indicated: "++" means more than 10%, "+" means 10–5%, "-" means less than 5%. **D** AKH-producing CC cells are located at the "foot" region of the ring gland associated with the larval brain. Ca²⁺ activity in the CC cells was monitored using *Akh-Gal4 > GCaMP5G*. **E–F** Dissected larval brains together with ring glands were incubated in a HL6 medium with amino acids. Representative Ca²⁺ activities in APCs after the addition of the indicated amino acids (5 mM) are shown. A representative trace of Ca²⁺ dynamics from a single APC cluster was plotted in (**F**). Ca²⁺ intensity

was plotted in arbitrary units (a.u.) **G** Quantification of Ca²⁺ activities in the APCs 15 mins after the addition of the indicated amino acids. N = 3, 3 brains. **H** 3rd instar larvae fed on indicated food for 36 h and then dissected. AKH levels in the APCs were stained with anti-AKH antibody. **I–J** Representative staining of AKH and quantifications of the fluorescent intensity in APCs of fly larvae fed on 2% sucrose, 2% sucrose + 40 mM Met, or 2% sucrose + 40 mM Leu. N = 14, 11, 15 brains (**J**). **K** TAG contents of adult flies fed on indicated foods for 48 h were measured. Diets containing 2% sucrose or 2% sucrose + 40 mM Met were used. N = 4, 4, 4 independent biological replicates. **L** Sugar and amino acid sensing properties of AKH as well as its downstream Ca²⁺ waves are functionally conserved compared with mammalian glucagon. For (**G, J, K**), each dot represents an independent biological replicate. Unpaired two-tailed Student's *t*-test was used in (**G**), ordinary one-way ANOVA with Dunnett's multiple comparisons was used in (**J**), and ordinary two-way ANOVA with uncorrected Fisher's LSD was used in (**K**). Data were plotted as mean ± SD. Scale bars, 50 µm (**B**), 25 µm (**E, I**). Source data are provided as a Source Data file.

can be observed only when the random fluctuation of Ca²⁺ is considered. Within the parameter range that ensures oscillating/wave behavior, an increase in Ca²⁺ fluctuation amplifies the difference between WT and *Inx2*-RNAi in the simulations of the larval fat body. A potential explanation is that the stochastic fluctuations in Ca²⁺ may cause a Ca²⁺ cycle/status that is not synchronized with the external AKH trigger, interfering with the global Ca²⁺ waves. A functional gap junction amplifies this local Ca²⁺ cycle, thus reducing the global ICWs. Decoupling of the neighboring cells with *Inx2* knockdown suppresses local ICWs but promotes global ICWs. The precise reason why random fluctuations serve as a determining factor for wave properties requires further investigation. Yet, our results suggest that the random effect should not be overlooked in the research of biochemical waves.

The APCs do not respond immediately to the applied amino acids but instead exhibit a gradual increase in activity over a 5–10 min period. This suggests that amino acids may not function through a rapid response mechanism such as ligand-gated channels, but via a comparatively slower metabolic process, possibly an increase in cytosolic ATP following amino acid breakdown. Furthermore, the fast and continuous response of the fat body to extracellular AKH without adaptation suggests that AkhR does not undergo activation-induced inactivation. This immediate and persistent response to extracellular AKH could provide flies with an advantage in promptly adapting their metabolic state to environmental changes. Meanwhile, the in vivo half-life of AKH is remarkably short due to an unknown mechanism, probably involving a specific secreted proteinase. Identifying unknown

serum factor may provide insights into the dynamic regulation of hormonal signaling.

Previous studies have identified amino acids sensing mechanisms in different fly organs: amino acid triggers the release of GBP1/2 and Stunted from the fat body to stimulate the *Drosophila* insulin-like peptides (dILPs) secretion and promote larval growth[61,62], essential amino acids promote the release of CNMa from intestine cells to regulate feeding behavior[63], FMRFa secretion from neurons is triggered by amino acid consumption to mobilize lipid stores[64], and IPCs sense amino acids to increase Insulin production[35]. Our study shows that APCs are also activated by amino acids. It will be important to explore how these amino acid-dependent signals interact or integrate to achieve systematic homeostasis. Meanwhile, it is conceivable that AkhR activation has a more extensive role beyond lipid mobilization. Recently, AKH has been found to activate extracellular signal-regulated kinase (ERK), which increases amino acid catabolism and gluconeogenesis in the fat body[65]. It seems plausible that amino acid-induced AKH secretion is a mechanism to process excessive amino acids ingested from the diet. This observation resembles the mammalian Liver-α-Cell axis, whereby an increase in glucagon levels upregulates the expression of amino acid transporters such as *Slc38a4* and *Slc38a5* in the liver, thereby enhancing amino acid uptake resulting in gluconeogenesis and urea production[66]. Moreover, elevated amino acid levels in the bloodstream also contribute to α-cell proliferation, leading to pancreatic α-cell hyperplasia in mice, creating an endocrine feedback[66]. It will be interesting to investigate whether certain amino acid transporters are upregulated by AKH in the fly fat body, and whether the function of APCs is modulated by amino acid consumption (Fig. 8L).

Collectively, our study shows that AKH triggers ICWs in both larval and adult adipose tissues. Furthermore, we discovered that the global and local ICWs in the fat body are generated through different molecular mechanisms: the global ICWs in the larval fat body are generated by the extracellular circulation of AKH secreted from APCs, whereas local ICWs in the adult fat body are formed through intercellular signal propagation mediated by gap junctions. Finally, we found that specific dietary amino acids activate the APCs, leading to increased AKH secretion, which stimulates ICWs in the *Drosophila* fat body to promote lipid metabolism.

## Methods

### Fly husbandry
Flies were raised on standard cornmeal food (2000 ml water, 12.8 g agar, 80 g yeast, 112 g cornmeal, 176 g glucose, 2.5 g methylparaben, 20 ml propionic acid, total 2 l of fly food) at 25 °C with 12 h:12 h light:dark cycles. All fly strains used in this study are listed in Supplementary Table 1. Genotypes for each figure are listed in Supplementary Data 1.

### Molecular cloning and generation of transgenic flies
The UAS-GCaMP5G-T2A-mRuby3 was generated by insertion of the GCaMP5G-T2A-mRuby3 into a fly pVALIUM10-roe vector through Gateway recombination. To generate *Akh-Gal4* (on X) and *Akh::GCaMP7s-T2A-mRuby3* flies, the AKH promoter containing sequence from −1020 to +17 relative to the transcription start site was first cloned into a pCaSpeR-attB vector before a Gateway cassette[4]. Then the Gal4 or GCaMP7s-T2A-mRuby3 sequence was inserted after the AKH promoter via Gateway recombination. The transgenic flies were generated by injecting embryos with the specified landing site (Supplementary Table 1) using PhiC31-mediated site-specific recombination by UniHuaii Corporation.

### Ex vivo GCaMP imaging in APCs
Early 3rd instar larval brains were dissected in a modified basal hemolymph-like solution devoid of any amino acids (modified HL6 (AA-) buffer) (74.2 mM NaCl, 2.0 mM $MgCl_2$, 10.0 mM $NaHCO_3$,

24.8 mM KCl, 0.5 mM $CaCl_2$, 80 mM Trehalose, 5 mM BES, pH 7.2). Dissected brains were immobilized using a slice holder with nylon grids (SH13, Scientific Systems Design Inc.) in the perfusion chamber. The samples in the HL6 (AA-) buffer were recorded for 1 min to generate a baseline. Next, the solutions were changed to HL6 (AA-) buffer + AA (5 mM) with the pH adjusted to 7.2 by gentle perfusion for 10 min. All imaging studies were performed with a Leica M205 FCA high-resolution stereo fluorescence microscopy,

### Ex vivo culture and imaging of larval fat bodies
The fat bodies of 3rd instar larvae were dissected in complete Schneider's medium (Schneider's medium from Sigma, supplemented with 10% FBS and 1% streptomycin and penicillin). A cleanly dissected fat body without any other associated organs was gently rinsed once with the same medium to remove larval hemolymph and then transferred using a transfer pipet to the center of a 35 mm dish with a 20 mm glass bottom (Nest, 801001), along with a drop of Schneider's medium (~20 µl). Next, a 1 × 1 cm² cellulose tea-bag paper (or nylon film with approximately 100 µm pore size) was placed on top of the fat body to spread it. Subsequently, a stainless-steel ring with a diameter of 1 cm was gently placed on top of the film to prevent floating of the tissue. Finally, 200 µl of complete Schneider's medium was added on top of the fat body. The fat body was then imaged using an inverted microscope. For the co-culture experiment, specific organs such as the brain, gut, or muscle/cuticle were dissected, rinsed twice, and transferred into the fat body culture dish. Organs from 8 to 10 larvae were used to activate the fat within 200 µl of culture medium. The $Ca^{2+}$ activity reported by GCaMP5G was examined for at least 30 min after co-culture.

For the neuropeptide treatment experiment, neuropeptide stocks were prepared by dissolving peptide powder (synthesized by Genscript and sequences shown in Supplementary Table 2) in either water or DMSO to a concentration of 1 mg mL⁻¹ depending on solubility. For peptides with disulfide bonds (AstC, CCHa1, and CCHa2), 100 µg mL⁻¹ peptides were treated with 10% DMSO in PBS at room temperature for 6 h. The efficiency of the reaction was assayed by Ellman reagent (DTNB assay kit, Solarbio BC1175), and more than 85% of the peptides were oxidized. During live imaging of the cultured fat body, the samples were first recorded for 5 min (with a time interval of 5 s) to generate a baseline. Subsequently, the medium was replaced by Schneider's medium containing 100 ng mL⁻¹ of different synthesized *Drosophila* neuropeptides, and recording was carried out for 20 min (time interval: 5 s).

Time-lapse recording was performed on a Leica DMi 8 equipped with a Leica DFC9000 sCMOS camera and Leica Application Suite X software. A 1.25x/N.A. 0.04 HC PL Fluotar objective were used. For the higher resolution of the gap junction knockdown experiments, a 5x/N.A. 0.12 Plan objective was employed. GCaMPs was excited by a 475 nm laser and the emission was collected with 510 nm filter. Imaging was conducted in a dark room at 18 °C. We utilized MATLAB to remove background noise and highlight the edges of the sample.

### Ex vivo culture and imaging of adult fat bodies
For adult flies, as most of the fat bodies adhere to the inside of the abdominal cavity, we dissected the dorsal shell together with the fat bodies. The dorsal shell was then adhered to a live cell imaging dish by Vaseline, such that the fat bodies face upwards, and bathed in 200 uL Schneider's medium. Time-lapse recording was performed on a Leica M205 FCA high-resolution stereo fluorescence microscope (Leica) equipped with a DFC7000 GT camera. The remaining steps and parameters are similar to those in the larval experiment.

### In vivo imaging of immobilized larvae and adults
For larvae, we attached two coverslips to a microscope slide with double-sided tape to form a thin slit, then used plasticine to plug both

sides of the slit so that the gap in the slit is the same width as a 3rd instar larva (Supplementary Fig. 1A). Next, a coverslip was added on the top of the chamber to immobilize the larva. For adults, the six legs of an adult female fly were cut off, and the wings adhered to a live cell imaging dish with Vaseline so that the abdomen faced upward. For the carbon dioxide and chloroform treatment, the chamber was covered with a transparent petri dish to prevent gas leakage. Time-lapse recording was performed on a Leica M205 FCA high-resolution stereo fluorescence microscope equipped with a DFC7000 GT camera and Leica Application Suite X software.

## In vivo imaging of fat bodies of free-behaving larvae

We used a 3D printed mold (Wenext) to produce an agarose gel containing different nutrients. The center of the gel has a 11 mm*13 mm*0.66 mm chamber, which can provide a free-behaving arena for more than 10 early 3rd instar larvae. Finally, a coverslip was added on top of the chamber to prevent the larvae from escaping. After being allowed to acclimate for 10 min, larvae were recorded for 20 min (time interval:5 s), then all larvae were transferred to an agarose gel chamber containing another type of food for 20-min recording. To ensure that there was no food residue in and on the body of the 3rd instar larvae, larvae were starved for 9 h before imaging and washed with ddH$_2$O during each transfer process. Time-lapse recording was performed on a Leica M205 FCA high-resolution stereo fluorescence microscope (Leica) equipped with a Leica DFC7000 GT camera. In image processing, we used connected component analysis to approximate each connectome as a larva. Fluorescent signal changes were normalized using the following formula: $\Delta F/F_O = (F(t) - F_O)/F_O$, where $F(t)$ is the fluorescence at time $t$, $F_O$ is the average baseline (before transfer).

## In vivo imaging of APCs of free-behaving larvae

As the thick cuticle of the 3rd instar larvae caused a strong blurring of the signal, we used more transparent 1st instar larvae. To perform high-resolution neuronal imaging of free-behaving 1st instar larvae, we used an extended-depth-of-field microscope with two modules: 1. A dark-field imaging module equipped with a 4X NA 0.2 air objective (Nikon, Japan) and a high-speed near-infrared camera (acA2000-340kmNIR, Basler ace), used to track and record a free-behaving 1st instar larva. 2. A fluorescence imaging module equipped with a 10X NA 0.3 air objective and a sCMOS camera (Zyla 4.2, Andor Inc., UK), with an imaging surface split in two by Optosplit II (Cairn, UK), which allows simultaneous recording of two fluorescent signals (calcium-sensitive GCaMP and calcium-insensitive RFP used as reference). Excitation and emission filters for GCaMP and RFP were 475/510 and 560/590 respectively. Through extended-depth-of-field technology, the effective depth of field was extended by about 5 times, avoiding errors on the Z-axis caused by motion. Images were acquired using Micromanager software. Because the signals from APCs are much dimmer than those of the fat body, we used a starvation diet initially to achieve a reliable visualization of Ca²⁺ activity. Before imaging, the 1st instar larvae were starved for 6 h, and then the larvae were gently picked with a brush into an agarose gel chamber (Φ20 mm*0.15 mm) made with a 3D printed mold (Wenext) for 15-min recording. We used 5% sucrose instead of 2% when imaging the 1st instar larval APCs because 5% sucrose triggers a quicker and stronger AKH suppression. However, 2% sucrose was used for the experiment in 3rd instar larvae, as the AKH secretion is too severely suppressed with 5% sucrose, making the Tryptone diet less effective. All image analyses were conducted using ImageJ and MATLAB. The fluorescent signal changes were normalized using the following formula: $\Delta F/F_O = (F(t) - F_O)/F_O$, where $F(t)$ is the fluorescence at time t, $F_O$ is the average baseline (before transfer).

## Image processing and quantification

Kymograph of Ca²⁺ waves were used to highlight the directive propagation of the wavefront in the fat body of the 3rd instar larva. First, an average density projection was executed on each frame along the Y-axis (representing the direction from the head to the tail of the larva). This process compressed each frame into a solitary line of pixels, spanning from the head to the tail. Subsequently, all these lines from each frame were combined into a kymograph, with time as the X-axis, to illustrate the direction of Ca²⁺ wave propagation.

The temporal dynamics of Ca²⁺ activity within a specific fat cell or region were plotted using ImageJ. Specifically, a region of interest (ROI) containing the cell or region of interest was first selected in a multi-frame live-image stack. Then, the "Plot Z-axis Profile" in the Stack function group was employed to trace the Ca²⁺ level within the ROI over time.

To clearly visualize the calcium (Ca²⁺) dynamics within the fat bodies, background fluorescence that does not vary over time was eliminated by subtracting an averaged image (calculated through the mean projection of the time-lapse sequence) from each frame. Note that fat cells damaged during dissection typically exhibit a consistently high Ca²⁺ signal, which is also removed by this process. The edges of the tissue were delineated with the Canny edge detector in MATLAB.

To quantify the percentage of Ca²⁺ active cells, the whole fat tissue was first selected with a low-intensity threshold (selected by masking the entire fat body) to calculate the total area. Then, Ca²⁺ positive cells were selected with a high threshold (selected by masking the cells with obvious Ca²⁺ signal, usually ~3–5 times higher than the Ca²⁺ negative fat cells). As fat cells are essentially uniformly sized, the ratio between the Ca²⁺ positive area and total fat area was calculated as the percentage of Ca²⁺ active cells.

To calculate the Ca²⁺ diffusion area, we first used threshold function to select all Ca²⁺ positive cells in the fat body. Then the area of individual Ca²⁺ positive regions in the field was calculated by the Analyze Particles function in ImageJ. The average size of Ca²⁺ positive region/diffusion area was calculated from different 20-min time-lapse imaging data.

To calculate the local Ca²⁺ wave velocity, we first manually defined two lines (separated by ~50 μm) parallel to the wavefront. Subsequently, we measured the time required for the wavefront to traverse the distance between these two lines. The wave speed of these local ICWs was determined by dividing the distance by the time.

To assess the Ca²⁺ oscillation correlation, two adjacent cells exhibiting Ca²⁺ oscillations were randomly selected. The sequences of Ca²⁺ intensity changes over time for these two cells were individually obtained. Subsequently, the absolute value of the Pearson correlation coefficient between the two sequences was calculated.

To trace the trajectory of the fluorescent beads, the auto-fluorescent was first removed using the Subtract Background function in ImageJ, followed by tracing the particles with their trajectories calculated using the TrackMate plug-in developed by Ershov, D et al.[67].

## Collection and treatment of conditioned medium

For the preparation of brain-conditioned medium, 20 brains from 3rd instar larvae were initially dissected in PBS and then transferred to 400 μl of Schneider's medium and incubated for 1 h at room temperature. Subsequently, the supernatant of the incubation was collected and referred as the brain-conditioned medium. To treat the conditioned medium, Proteinase K (0.1 mg mL⁻¹, Invitrogen 25530049), DNAse I (1 U mL⁻¹, ThermoFisher EN0521), or RNAse A (1 μg mL⁻¹, ThermoFisher EN0531) was added (final concentration) and incubated at 37 °C for 1 h. Then the added enzymes in the conditional medium were removed with the 10 kDa filter. For filtering the conditioned medium, a 0.5 mL Pierce 10 K MWCO concentrator (Thermo-Fisher 88513) was used. The filtration was conducted on a benchtop centrifuge at 15,000 g for 30 min and approximately 200 μl of the filtered medium was collected for the treatment of the cultured fat body. To generate brain-conditioned medium with specific amino acids, 10 3rd instar larval brains were dissected in modified basal

hemolymph-like solution devoid of any amino acids (modified HL6 (AA-) buffer) (74.2 mM NaCl, 2.0 mM $MgCl_2$, 10.0 mM $NaHCO_3$, 24.8 mM KCl, 0.5 mM $CaCl_2$, 80 mM Trehalose, 5 mM BES, pH 7.2). Dissected brains were washed once and incubate with the HL6 (AA-) medium plus 5 mM specific amino acid for 1 h. Then the supernatant of the incubation was collected for the following experiments.

### AKH secretion assay
To measure AKH retention in APCs, early 3rd instar larvae were picked out from standard food and washed with $ddH_2O$, and after feeding for a period of time under different dietary conditions, the brains were dissected in PBS (1.86 mM $NaH_2PO_4$, 8.41 mM $Na_2HPO_4$, 175 mM NaCl) and fixed in 4% v/v paraformaldehyde (PFA)/PBS for 30 min at 23 °C. After washing in PBST (PBS + 0.05% Triton X-100, BBI) (3 times, 10 min each), the brains were blocked in 5% BSA (Solarbio, A8020) in PBST for 1 h at 25 °C, then incubated with rabbit polyclonal anti-AKH (1:1000; ABclonal, raised against synthesized fly AKH peptide) for 12–20 h at 4 °C. After washing in PBST (3 times, 10 min each), the sample brains were incubated with Alexa Fluor 488 goat anti-rabbit IgG (1:500; Invitrogen) for 1 h at 25 °C and washed again using PBST (3 times, 10 min each). We used an antifade agent to mount the samples. All images were acquired using a Leica M205 FCA high-resolution stereo fluorescence microscope (Leica).

### TAG assay
5 flies from each group were homogenized with 100 μl of isopropyl alcohol (BBI, A600918-0500), centrifuged at 10,000 g for 10 min at 4 °C, and the supernatant was collected. 2 μl of sample solution was mixed with 200 μl of assay reagent (Elabscience, E-BC-K261-M), and incubated at 37 °C for 10 min. We measured the absorbance at 492 nm in a microplate reader (Thermo Scientific Multiskan FC).

### Heart rate assay
The early 3rd instar larvae were attached to a glass slide with light-curing glue, ensuring that their dorsal sides were facing up. Recordings were captured on a Leica M205 FCA high-resolution stereo fluorescence microscope (Leica) equipped with a Leica DFC7000 GT camera. The tracheal movements can readily be seen moving with each heartbeat. To properly understand the dynamic process, it is imperative to measure them in a time-dependent way. We draw a single-pixel line perpendicular to the trachea, and then use the multi-kymograph function of ImageJ to generate a kymograph displaying tracheal movement in the $Y$-axis and time along the $X$-axis. Then the heart rate was calculated by dividing the number of peaks in the kymograph by time. All image analyses were conducted using ImageJ.

### Hemolymph flow assay
To detect the direction of the hemolymph flow, flies anesthetized with carbon dioxide were injected with PBS + 0.1%BSA containing 5 μm diameter fluorescent beads (ex/em: 535/610 nm, Hugebio). Before the injection, the beads were blocked in PBS + 10% yeast extract overnight to prevent adhesion to the tissue in vivo. For larvae, we chose to inject from the tail, and for adults, we chose to inject from the abdomen. After injection, the flies were attached to a glass slide with light-curing glue. Images were acquired with a Leica M205 FCA microscope and Leica Application Suite X software. All image analyses were conducted using ImageJ and MATLAB.

### Dot blot analysis of hemolymph AKH
To extract hemolymph, for larval flies, the body was punctured using microsciccors, while for adult flies, the thorax was gently speared with the peaked stylus. Forty flies, starved for 6 h, were collected and centrifuged at $2000 \times g$ for 15 min at 4 °C in 0.5 mL Eppendorf tube embedding in 1.5 mL Eppendorf tube. 3 μL of hemolymph were diluted in 15 μL of PBS, heated in a 85 °C metal bath for 20 min, and

centrifuged. 5 μL of the supernatant were then spotted onto a 0.2 μM nitrocellulose membrane (GE Healthcare) and left at room temperature until dry. Subsequently, the membrane was boiled in PBS for 3 min, and stained with Ponceau Red as a loading control before blocking with 0.5% BSA in TBST (0.5% Tween-20 in TBS) for 1 h. The membrane was then incubated with purified anti-Akh antibody (1:2000, rabbit anti-AKH from Dr. Song Wei[65]) diluted in TBST containing 2% Polyvinylpyrrolidone (MACKlLIN, P816205) overnight at 4 °C and subsequently with horseradish peroxidase (HRP)-conjugated secondary antibody (1:2000; Invitrogen) diluted in TBST for 1 h at room temperature. The membrane was then visualized using the BeyoECL Moon (Beyotime, P0018FS) and detected by the Tanon 5200 Chemiluminescent Imaging System.

### RT-PCR
Total RNA was extracted using TRIzol Reagent and then quantified with a NanoDrop 2000 spectrophotometer. cDNA was synthesized from 500 ng of total RNA using the HiScript II 1st Strand cDNA Synthesis Kit ( + gDNA wiper) (Vazyme) according to the instructions. Quantitative real-time PCR was performed using AceQ qPCR SYBR Green Master Mix (Low ROX Premixed) (Vazyme) on a QuantStudio 3 system (Thermo Fisher Scientific). The gene expressions were normalized to $\alpha$-Tub84B. The primer sequences are as follows (F, forward; R, reverse): Inx2F 5′-TGATCGCCTTCTCCCTCCTG-3′, Inx2R 5′- TGGG ATCTCGTCCACAATACA -3′; α-Tub84BF 5′- TGTCGCGTGTGAAACAC TTC-3′, α-Tub84BR 5′- AGCAGGCGTTTCCAATCTG-3′.

### Statistics
Sample sizes were determined through preliminary experiments and previous studies to achieve the required statistical power. All assays were repeated more than three times. Quantitative and statistical parameters, including statistical methods, error bars, n numbers, and $p$-values, are indicated in each figure. Two-tailed unpaired Student's $t$-test was performed for comparison between two groups of samples. One-way ANOVA followed by multiple comparison test was performed for comparisons among three or more groups of samples. Two-way ANOVA was performed for the interaction between the treatments. $P$ values of <0.05 were considered statistically significant. Image processing and quantification were performed in ImageJ and MATLAB. Plotting of graphs and statistical analyses were conducted with GraphPad Prism 8.4.2.

### Reporting summary
Further information on research design is available in the Nature Portfolio Reporting Summary linked to this article.

## Data availability
All data generated or analyzed in this study are included in the main text or the Supplementary materials. The raw data generated in this study are provided in the Source Data file in this paper. Source data are provided with this paper.

## Code availability
The computational modeling was conducted using MATLAB. All codes used for modeling of this study are deposited in the GitHub repository (https://github.com/Sy-Luo/HeLab_CalciumWave_Simulation_2024) and the open-access repository Zenodo with DOI: 10.5281/ zenodo.14175603.

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

## Acknowledgements

funding for this work is from the National Natural Science Foundation of China (No. 32070750), the startup funds from USTC to L.H., the American Heart Association (Award ID 24PRE1189954) for M.A., Young Talent Fund of the University Association for Science and Technology in Shaanxi, China (No. 20210506) for W.S. NP is an investigator of Howard Hughes Medical Institute. We thank Dr. Quan Wen for providing the imaging facilities enabling us to monitor the freely behaving larvae, Dr. Wei Song for sharing the anti-AKH antibody for dot-blotting and the Akh mutant flies, Dr. Jun-yuan Ji for sharing the Fb-Gal4 fly stock, and Dr. Aforditi Petsakou, Dr. Ben Ewen-Campen, Dr. Bernard Mathey-Prevot for valuable comments on the manuscript.

## Author contributions

Conceptualization, experiments design by L.H. and N.P.; experiments performance, data collection and analysis by M.A., S.W. and X.G.; modeling and computational simulation by W.S., S.L. and Y.C. All authors have read and agreed to the published version of the manuscript.

## Competing interests

The authors declare no competing interests.
