## [Transparent Peer Review file · Nature Communications]

Dietary Amino Acids Promote Glucagon-like Hormone Release to Generate Global Calcium Waves in Adipose Tissues

Corresponding Author: Professor Li He

Version 0:

Reviewer comments:

Reviewer #1

(Remarks to the Author)

The authors found that intercellular calcium waves (ICWs) occurs in larval fat bodies and identified AKH as the trigger for ICWs. ICWs in larval fat bodies occur in the AP body axis, whereas in ex vivo culture ICWs occur randomly. Random ICWs is mediated by the gap junctional protein *Inx2*, but *Inx2* is not involved in the ICW pattern in vivo. Since overexpression of AKH in fat bodies prevents ICWs along the AP body axis, they thought one possible mechanism is that the concentration of AKH is higher in the head region, which is closer to the ring gland, a source of AKH. Together with computational modeling, they proposed the cyclic ICW pattern along the AP axis is caused by the short half-life of AKH and its cyclic release. Specific amino acids that promote AKH release have also been identified. For example, small polar amino acids were found to raise Ca^{2+} of AKH-producing neurosecretory cells (APCs) and promote AKH release. The present study suggest that AKH release regulated by starvation and amino acids generate ICWs. This study was interesting because they discovered ICWs and approached its propagation mechanism through experiments and simulations. It is assumed that AKH release occurs in a pulsatile manner under normal rearing conditions, but can pulsatile AKH release be shown experimentally?

Specific Comments:

Fig. 1B. The quantification method is unclear. Is the quantification performed on the whole body?

Fig.1C, D. How was the tissue used for coculture prepared? Also, there is no description of how coculture was performed.

Fig.1E. These is no description of the quantification method.

Fig.1G, H. There is no description of how to prepare the conditioned medium and how to treat with DNase and RNase enzymes.

Fig.2A There is no description of how the peptide used was prepared.

Fig.2D It is unclear whether AKH is adequately removed by the addition of the antibody. Experiments using control antibody is also necessary. However, since the requirement of AKH and AKHR has been shown genetically, antibody experiment may not be necessary.

Fig.S3B. Explanation of the starvation experiment is needed.

Fig.3E. How did you quantify "control local"?

Fig.3J-K. AKHR is stimulated by AKH, which promotes TAG catabolism. The involvement of ICWs for TAG catabolism is not clear.

Fig.4F-H. CO₂ treatment reduces APC activity, but the relationship between APC Ca^{2+} dynamics and AKH release has not been directly shown. I am also wondering whether the half-life of AKH can be directly measured?

Figure 5I. The authors attribute the difference in ICW response between larvae and adults to the uniform distribution of extracellular AKHs in adults. However, the argument that this difference is due to the location of APCs in the thorax seems to be weak; it is possible that AKH is abundantly secreted in the adult body fluid, as is the case with the Akh overexpression movie in the fat body in Figure 3D, where ICW occurs in a random direction. This can be test by comparing AKH levels in the blood lymph of larvae and adults by western blot or ELISA.

Fig. 5I. The authors propose a model of pulsatile AKH secretion in larvae. However, the rhythm of AKH secretion itself is unclear. Considering that the velocity of bead movement in Fig. 4E (250 $\mu\text{m/s}$) is much faster than the wave velocity of global ICWs in Fig. 3E (40 $\mu\text{m/s}$), it might be more relevant to assume the desensitization speed of the fat body rather than focusing on AKH secretion.

Why is the Calcium intensity increased in *inx2*-RNAi? Based on Suppl Fig. 7B-D, is there a mechanism that delays AKHR desensitization? Could the suppression of local ICWs lead to a compensatory increase in global ICWs? Since this point is also supported by mathematical models, further discussion based on the mathematical model on which factors are important would be possible.

Fig.S11. What is the status of AKH in normal food?

Minor Points:

1.The velocity measurement experiments using beads in larvae shown in Fig. 4D are excellent. However, compared to the videos of adults, the beads appear to get stuck in the tissue, preventing them from moving. Additionally, the fact that many beads are not moving is a concern. While using smaller beads may be difficult, it would be advisable to mention these limitations as technical constraints.

2.The notation for the fat body Gal4 driver in Figure 1 varies between Fat>, Lpp>, and Fb>. It would be better to standardize the notation.

3.The choice of GCaMP5G raises some questions. Compared to GCaMP6s, the $\Delta F/F$ is lower (Nature. 2013 Jul 18; 499(7458): 295–300). And GCaMP6s was used in the ICWs and SIDICs papers in wing discs. Was this lower sensitivity of 5G advantageous for detection of ICWs in the fat bodies?

4.The differences in response in adult and larval FB are very interesting. If data were available, could they be discussed in terms of differences in localization patterns of AKHR and *inx2* expression? What about differences in fat amount or metabolic demands during development and maturation stages?

5.There is character encoding issue on the y-axis label for Velocity in Suppl Fig. 8B.

6.Is the term "large epithelial tissue" appropriate to describe the fat body in line 518?

7.It would be better to unify the use of the abbreviation for insulin-producing cells (IPCs) throughout the manuscript.

Reviewer #2

(Remarks to the Author)

This is a well-written and thorough manuscript uncovering the role of Ca^{2+} signaling in the *Drosophila* fat body in regulating fat storage mobilization. As mentioned in the Discussion section, this study also provides insights into the Liver-alpha-Cell axis in human biology. In this context, Ca^{2+} signaling is stimulated and acts via hormonal (Adipokinetic hormone, AKH) signaling from specialized cells (APCs). AKH is sensed by the fat body cells expressing the AKH-Receptor (AKHR), a G-Protein Coupled Receptor (GPCR), that activates $\text{G}\alpha_q$ to stimulate Ca^{2+} signaling. The authors investigated the mechanism of transport of AKH, which is circulated in the hemolymph throughout the larval body and is secreted from the APCs, and how the fat body is able to respond to this hormonal signaling with two distinct types of intercellular calcium waves (ICWs). The authors argue that two distinct mechanisms of ICWs dynamics are active in larval fat body: ICWs are spread by either intercellular gap junctions channels providing a slow, local spread, or alternatively, through convective transport due to hemolymph flow that results in extensive calcium waves, initiated through AKHR- $\text{G}\alpha_q$ signaling. They evaluated the differences in calcium signaling between larval and adult stages in the fat body. This aspect of the story mostly confirms and extends previous findings already described in the literature regarding the role of $\text{G}\alpha_q$, PLC Beta and Ca^{2+} signaling in the fat body. A new and significant finding is that this AKH hormonal signaling in larvae, in addition to hunger, can also be triggered by dietary addition of two amino acids, including Methionine and Threonine. These amino acids are sensed by APCs and stimulate AKH release, triggering ICWs and fat mobilization in the fat body. The manuscript is logically organized. The data is robust, and most of the authors' data interpretation and conclusions are reasonable.

However, some of the results are not unexpected. The authors should clearly distinguish what is a confirmation of the already known findings and what are the new findings. The claimed new results are stated in the abstract, citing "We identified Adipokinetic hormone, ... as the key factor driving Ca^{2+} activities in adipose tissue," it should be restated more circumspectly as "we confirmed previous reports with an additional line of evidence." This result is not a completely new finding, and was described previously in the literature. Furthermore, the GPCR AKH-Receptor works via $\text{G}\alpha_q$ also was described previously in the literature (for example, Bumbach "G α_q , G γ 1 and Plc21C Control *Drosophila* Body Fat Storage" <https://doi.org/10.1016/j.jgg.2014.03.005>).

A significant new finding is that authors are claiming that the larval fatbody exhibits two distinct types of ICWs. thors tested a variety of amino acids and found that specifically methionine and threonine are sensed by APCs to trigger AKH release, which in turn stimulates ICWs in the body. This suggests that these two amino acids are able to trigger “starvation”-like responses. It is possible that foods that are rich in these two amino acids are not the preferred/best food sources for the larvae, and if given a choice they would not choose to have this nutritionally “poor” diet. As in another study where one of the coauthors had participated (<https://doi.org/10.1038/s41586-022-04960-2>) about adult fly preference of Leucine in the food. In this study Leucine did not trigger APCs activation, clearly demonstrating that Leu is a preferred dietary amino acid for fly larvae. According to flybase expression data, AKHR is expressed in the larval fat body, but in adults, it is strongly expressed in the head. This suggests that AKH could be sensed in adult heads and could potentially trigger behavioral changes in adults to search for new food sources, for example. The impact of this finding is significant as it has the promising potential for application towards the creation of new medication for obesity, heart disease, and diabetes for humans.

There are a couple of areas for improvement in the present manuscript. A significant finding in the manuscript is that ICWs can happen independently of gap junctions, and therefore are not spread cell-to-cell, but rather reflect an extracellular gradient of hormones. But this finding could be bolstered further. This claim is based on using *Inx2*-RNAi-mediated knockdown, but no verification was made to ensure that the *Inx2* knockdown was 100% knockdown of gap junction activity. Even though pharmacological treatment was used, it was performed in cultured fat bodies and not in vivo, so it can not be used as a substitute for in vivo verification. Also, it is not explicitly mentioned what is the sample size in these experiment. For example the ex vivo experiment in Fig. 3.

Another concern regarding the clarity of presentation of results is that in Supplementary Video 8. The Luc-i condition includes global ICWs (the waves are extensive), although the authors state completely opposite, that no global ICWs were observed in adults, and that global ICWs are stage-specific, only attributed to the larval fat body. The definition of global vs. local ICW is vague in this case. Also, the sample size of this study should be more clearly identified.

The *Inx2*-RNAi line used in the study is made using Valium10 vector, and *Ipp-Gal4>GCamp5G* driver does not seem to have *Dicer2*. ICWs in the *Inx2*-RNAi look very similar to the control, there could be an incomplete penetrance phenotype for this *Inx2*-RNAi line. To strengthen the result, a secondary method such as CRISPR would be needed. Even though a pharmacological gap junction inhibitor was used in cultured fat bodies the results agreed with in vivo *Inx2*-RNAi results. To verify this result, complete inhibition is needed, perhaps with stronger inhibition, such as a second distinct line for *Inx2*-RNAi, or by repeating the in vivo experiment as in Supplementary Video 6 using another driver that has *Dicer2* in its genotype that will help to improve the knockdown or any other way that will provide a concrete evidence that *Inx2*-RNAi mediated knockdown was fully effective. This is an significant concern because the authors claim the novel finding of a new previously undescribed mechanism of ICWs based only on the results with *Inx2*-RNAi knockdown (which may not be 100% knockdown). Therefore, the basis of this novel finding should have more parallel lines of evidence.

It would also be helpful to perform the same experiment as in Video 8 but in cultured adult fat bodies and then update Fig. 3J, K. The adult cuticle is only partially transparent and prevents the observation of the full range of GCaMP5G, a Ca²⁺ sensor, will be out of the way, and the results will be more clearly demonstrating the actual phenomenon of ICWs.

Specific genetic background descriptors are needed in each and every video and image rather than the nondescriptive word “control,” because this introduces a guesswork for the reader. This abbreviation is used sometimes in the text and Videos, and it is not easy to find out what exactly was used as a control each time. The same applies to the driver lines. As an example: Video 6, that has labels “control” (what was a control?), *Inx2*-i (what was the driver?), AKHR-null, and AKH-OE (again, was the driver used?). It creates extra work for the reader to refer to the SI files. Also, it would be more clear if the location of the head is indicated, to clarify the direction of the ICW spread.

Lines 198-199: Please provide a more detailed description of creation of kymographs.

Supplementary Video 2: it is not clear where the brain is on the video. There is no clear larval brain outline.

Lines 42-90, the Introduction section: More than half of the introduction section (lines 64-90) was used to summarize results. Normally, the Introduction section should provide the background necessary for the reader to understand the following “results” section. Furthermore, the Introduction section did not fully cover the scientific literature on the topic of Calcium signaling. It would be helpful to condense the summary of results in the Introduction, and instead add more background information on the Ca²⁺ signaling. For example, there is a Baumbach, Xu, Hehlert, Kuhnlein 2014 paper that reports Gαq-related Calcium signaling in the fat body and its connection to the AKH and AKHR. (<https://doi.org/10.1016/j.jgg.2014.03.005>) and others.

Lines 104-106 where the authors describe the experimental setup: providing a cartoon or a diagram will improve clarity of the experimental set up.

Lines 125-130: the authors should reference Supplementary Figure 1 A.

(Minor issue): The lines 369 and 370 cited source 21 as the location of APCs, but it is not a best resource for describing the location of the APC location in adults, as in Figure 1 of the cited source 21 the location of APCs are shown in the head, which is contrary to what authors are saying. Perhaps an alternate source should be cited regarding the location of the corpora cardiaca DOI 10.1007/s00018-015-2063-3, fig. 1 or fig. 2). The authors are identifying the APCs location correctly, but they need to cite an alternate source.

In general, the introduction section should provide a more contextual literature summary, including ICW in wound detection and healing.

L77: Modify to the plural: computational simulations, unless only one simulation was implemented.

L99: be more specific on this point: "using a holder (what kind) in the perfusion chamber"

L172 : wave direction analysis methods could have been clearer. Consider revising for clarity.

L. 186: Is there a reference that needs to be added? "According to the concept of connected components in image processing..." this is not clearly defined.

L.215: Where is the description of how the transgenic lines were created that were first used in the study

L.271 need citation for Kir2.1 reference

L. 506 : Does insulin stimulate Ca²⁺ in fat body like on other tissues such as the wing disc?

L640 : Making the data available on repository will aid in transparency of results

The title is somewhat misleading: Novel Calcium waves are not a class of calcium waves. It may be a novel finding, but that is also overstated.

Computational codes: The study is only replicable if the codes are deposited to Github. The SI text was not fully transparent in describing and annotating computational code. the Statement "all MATLAB code in the simulations are fully disclosed" cannot be verified.

It was not completely clear how Ca²⁺ traces are measured (E.g. SI Fig 3E). Is the ROI a particular cell or area and does it differ from experiment to experiment?

Fig. S4: Why are there two rows for each condition? The darker second row is not explained. The heat maps of rel. F.I. are also not defined.

Definitions like global or tissue-wide Ca²⁺ waves are not clearly delineated, which makes it difficult to assess the comparisons being made between the larval and the adult experiments. Does a global wave mean that the whole fatbody is activated at the same time or that the wave progresses through the whole body?

Specifically, how can the effect of GJ inhibition leading to increased Ca²⁺ amplitude and decrease in frequency be explained in the computational model?

Reviewer #3

(Remarks to the Author)

I co-reviewed this manuscript with one of the reviewers who provided the listed reports as part of the Nature Communications initiative to facilitate training in peer review and appropriate recognition for co-reviewers.

Reviewer #4

(Remarks to the Author)

Reviewer #5

(Remarks to the Author)

The manuscript by Ahmad et al. describes a novel mechanism of nutrient sensing and metabolic regulation in the fruit fly *Drosophila melanogaster*. The study demonstrates that Adipokinetic hormone (AKH), the fly equivalent of mammalian glucagon, stimulates intracellular Ca²⁺ waves in the larval fat body to promote lipid metabolism. Specific dietary amino acids are shown to activate AKH-producing cells, leading to increased intracellular Ca²⁺ and subsequent AKH secretion. The study also reveals different mechanisms of Ca²⁺ dynamics regulation in larval and adult fat bodies, highlighting the interplay between AKH secretion pulses, extracellular hormone diffusion, and intercellular communication through gap junctions.

While it is disappointing that the precise biological significance of the phenomenon has yet to be fully understood, this manuscript represents a significant advancement in our understanding of nutrient sensing and metabolic regulation. The study utilizes innovative techniques and provides novel insights with potential implications for human health.

Overall, this is a very nice study, and the reviewer has only a few comments on the manuscript for revision.

[Points of concern]

(1) The authors' computational modeling is impressive as it provides a new strategy for *Drosophila* physiology. The reviewer is curious about the relationship between the predicted distribution of AKH in the body and the predicted Ca²⁺ influx in the

fat body in this model. The reviewer would like the authors to show the expected distribution of AKH together with the Ca²⁺ data in Figure 5A, and discuss the relationship between AKH distribution and Ca²⁺. The reviewer thinks that it would be of interest to many *Drosophila* researchers.

(2) Please carefully check Supplemental Table 2 and review what kind of peptides the authors really used. The reviewer realizes several concerns.

a) The peptide sequence of CAPA listed in this table (GDAELRKWAHLLALQQVDL) is entirely different from an actual CAPA mature peptide (GANMGLYAFPRV-amide; see <https://journals.plos.org/plosone/article?id=10.1371/journal.pone.0029897>).

b) While the authors used FQYSRGWTN-amide as Corazonin, the reviewer wonders why the authors used such a shorter peptide than the identified mature Corazonin peptide (QTFQYSRGWTN-amide).

c) While some peptides, such as Ast-C, CCHa1, and CCHa2, have internal disulfide bonds to be active forms, the authors did not state whether they use the proper mature peptides.

If the authors used the "wrong" peptides, they did not correctly examine the effect of some peptides, and the reviewer argues that the authors must carefully revise Supplemental Table 2 and related sentences.

The reviewer also has additional comments as follows.

d) "LK" and "Leukokinin" are exactly the same, and there is no need to include both data in Figure 2A.

e) The reviewer does not see any particular reason why some very well-known peptides, such as NPF and DH44, were not included.

If there are too many inadequacies here, changing the logic flow that focuses on AKH may be necessary.

Version 1:

Reviewer comments:

Reviewer #1

(Remarks to the Author)

The revised version of the manuscript by Ahmad et al. is significantly improved compared to the original submission, as the authors have addressed most of our concerns and the points raised by the other reviewers (although it was difficult to find changes in the text).

However, the following points still need to be addressed by the authors.

Main point:

The experiment in Figure 4 suggests that a beating heart is required for ICW and Figure 5K suggests that there is a pool of AKH in the head. The formation of an AKH pool in the head could be either pulsed or sustained release of AKH from the APC. It is possible that transport from the pool is the main mechanism for ICW. Therefore, the statement that AKH release is pulsed should be tempered.

Minor points:

Fig.1G, H. There is no description of how to prepare the conditioned medium and how to treat with DNase and RNase enzymes.

The authors mention that it was incubated in Complete Schneider's medium for 1 hour, but Complete Schneider's medium contains 10% FBS. Did 10% FBS not affect the proteinase K treatment?

Is the term "large epithelial tissue" appropriate to describe the fat body in line 518?

Response: We have removed "large" in the sentence.

The authors thought that the fat body is epithelial tissue, but fat body is more analogous to liver and adipose tissue in vertebrate. Thus, the *Drosophila* fat body is not classified as epithelial tissue.

(Remarks on code availability)

Reviewer #2

(Remarks to the Author)

Overall, the authors were responsive to the comments on the initial submission. The paper provides interesting findings of non-gap junction-based calcium waves in the fat body and a comprehensive analysis of the mechanisms.

(Minor points): One remaining point is that the authors should clarify if the traces in the figures are from a single representative cell or averaged for the tissue (Fig. 2B and Fig. 2D vs. Fig.6C & D). The shaded regions suggest an averaging of multiple samples, but this was not completely clear in reviewing the text. The shaded intervals should be explicitly defined in captions, or if applicable, state that the trace is representative of a single cell.

Fig. 7B-D mentions representative results, but the shaded regions are not fully described. Is Fig 8F only one trace or an average?

Similarly, there is a question of how analysis is shown in SI Fig. 10C and SI Fig 10.

Additionally, the citations could have been more comprehensive and complete regarding key recent findings of intercellular calcium waves in other *Drosophila* epithelia, such as the wing disc.

(Remarks on code availability)

The readme file was minimal. The code is for Matlab. The code appears to be commented.

Reviewer #3

(Remarks to the Author)

(Remarks on code availability)

Reviewer #4

(Remarks to the Author)

(Remarks on code availability)

Reviewer #5

(Remarks to the Author)

The authors have performed additional experiments and clarified the unclear information in the manuscript. The revised manuscript has been extensively improved. The reviewer does not have further comments on the manuscript.

(Remarks on code availability)

We appreciate the reviewers for their insightful and constructive comments and suggestions on our manuscript. In this revised manuscript, we have addressed all the comments by either performing the suggested experiments or by rewriting the text accordingly. Below, please find our detailed, point-by-point responses (in black) to each of the comments (which are fully copied in blue and italicized).

Reviewer #1: The authors found that intercellular calcium waves (ICWs) occurs in larval fat bodies and identified AKH as the trigger for ICWs. ICWs in larval fat bodies occur in the AP body axis, whereas in ex vivo culture ICWs occur randomly. Random ICWs is mediated by the gap junctional protein Inx2, but Inx2 is not involved in the ICW pattern in vivo. Since overexpression of AKH in fat bodies prevents ICWs along the AP body axis, they thought one possible mechanism is that the concentration of AKH is higher in the head region, which is closer to the ring gland, a source of AKH. Together with computational modeling, they proposed the cyclic ICW pattern along the AP axis is caused by the short half-life of AKH and its cyclic release.

Specific amino acids that promote AKH release have also been identified. For example, small polar amino acids were found to raise Ca^{2+} of AKH-producing neurosecretory cells (APCs) and promote AKH release. The present study suggest that AKH release regulated by starvation and amino acids generate ICWs. This study was interesting because they discovered ICWs and approached its propagation mechanism through experiments and simulations. It is assumed that AKH release occurs in a pulsatile manner under normal rearing conditions, but can pulsatile AKH release be shown experimentally?

Response: We thank the reviewer for recognizing the value of our research. The pulsatile release of AKH is a key inference drawn from our live-imaging experiment. However, directly measuring hormone release in small animals such as fruit flies remains a challenging experiment that is currently beyond our capabilities.

Taking the well-studied insulin as an example, its pulsatile release, with a period of approximately 4-10 minutes, was experimentally identified through the perfusion of the pancreas from large mammals and measuring the Insulin in the effluent every minute (reviewed by Niels Pørksen et al. 2002, doi.org/10.2337/diabetes.51.2007.S245). However, due to the small size of the fruit fly, it is impractical to extract hemolymph from a single fly continuously and determine its AKH concentration. Alternatively, cytosolic Ca^{2+} oscillation has been used as an indicator of Insulin release from isolated islets (reviewed by Niels Pørksen et al. 2002). We performed Ca^{2+} imaging of APCs both *in vivo* and *ex vivo* but have not found a clear periodicity in the Ca^{2+} levels in APCs, which may be due to our current technique limitation: the signal from *in vivo* imaging of APC is still highly noisy which prevents us from extracting fine temporal structure (**Fig. 7**), while the *ex vivo* culture conditions may not optimally preserve the natural excitation dynamics of APCs (**Fig. 8E-F**).

On the other hand, we still consider that the pulsatile release of AKH is the most plausible inference from our observation. Firstly, we have demonstrated that the formation of periodic global intracellular waves (ICWs) is triggered by AKH released

from APCs (**Fig. 2**). Secondly, we observed that the acute inhibition of APCs by application of CO₂ immediately blocked the AKH-triggered Ca²⁺ activity in the fat body (**Fig. 4F-L**), suggesting that the released AKH has a half-life of approximately 35 seconds, which is much shorter than the 3–4-minute interval of global ICWs. Taken together, these results strongly support our hypothesis that AKH is released from APCs in a pulsatile manner.

Specific Comments:

Fig. 1B. The quantification method is unclear. Is the quantification performed on the whole body?

Response: No, it is Ca²⁺ from a selected 100x100 μm square region at the midsection of a larvae. We now clearly state it in the figure legends.

Fig.1C, D. How was the tissue used for coculture prepared? Also, there is no description of how coculture was performed.

Response: In the method section, we have included a schematic illustration in Supplementary Fig. 1D and a detailed description of the experiment in the method section as follows: "The fat bodies of 3rd instar larvae were dissected in complete Schneider's medium (Schneider's medium from Sigma, supplemented with 10% FBS and 1% streptomycin and penicillin). A cleanly dissected fat body without any other associated organs was gently rinsed once with the same medium to remove larval hemolymph and then transferred using a transfer pipet to the center of a 35 mm dish with a 20 mm glass bottom (Nest, 801001), along with a drop of Schneider's medium (~20 μl). Next, a 1x1cm square cellulose tea-bag paper (or nylon film with approximately 100 μm pore size) was placed on top of the fat body to spread it. Subsequently, a stainless-steel ring with a diameter of 1 cm was gently placed on top of the film to prevent floating of the tissue. Finally, 200 μl of complete Schneider's medium was added on top of the fat body. The fat body was then imaged using an inverted microscope. For the co-culture experiment, specific organs such as the brain, gut, or muscle/cuticle were dissected, rinsed twice, and transferred into the fat body culture dish. Organs from 8 to 10 larvae were used to activate the fat within 200 μl of culture medium. The Ca²⁺ activity reported by GCaMP5G was examined for at least 30 minutes after co-culture."

Fig.1E. There is no description of the quantification method.

Response: We have grouped the image quantification descriptions into a new section of Image processing and quantification in the method: "Kymograph of Ca²⁺ waves were used to highlight the directive propagation of the wavefront in the fat body of the 3rd instar larva. Initially, an average density projection was executed on each frame along the Y-axis (representing the direction from the head to the tail of the larva). This process compressed each frame into a solitary line of pixels, spanning from the head to the tail. Subsequently, all these lines from each frame were combined into a kymograph, with time as the X-axis, to illustrate the direction of Ca²⁺ wave propagation.

The temporal dynamics of Ca²⁺ activity within a specific fat cell or region were

plotted using ImageJ. Specifically, a region of interest (ROI) containing the cell or region of interest was first selected in a multi-frame live-image stack. Then, the "Plot Z-axis Profile" in the Stack function group was employed to trace the Ca^{2+} level within the ROI over time.

To clearly visualize the calcium (Ca^{2+}) dynamics within the fat bodies, background fluorescence that does not vary over time was eliminated by subtracting an averaged image (calculated through the mean projection of the time-lapse sequence) from each individual frame. Note that cells damaged during dissection and transfer of tissue typically exhibit a consistently high Ca^{2+} signal, which is also removed by this process. The edges of the tissue were delineated with the Canny edge detector in MATLAB.

To quantify the percentage of Ca^{2+} active cells, the whole fat tissue was first selected with a low-intensity threshold (selected by masking the entire fat body) to calculate the total area. Then, Ca^{2+} positive cells were selected with a high threshold (selected by masking the cells with obvious Ca^{2+} signal, usually ~3-5 times higher than the Ca^{2+} negative fat cells). As fat cells are essentially uniformly sized, the ratio between the Ca^{2+} positive area and total fat area was calculated as the percentage of Ca^{2+} active cells.

To calculate the Ca^{2+} diffusion area, as presented in **Fig. 3C**, we first used the threshold function to select all the Ca^{2+} positive cells in the fat body. Then the area of individual Ca^{2+} positive region in the field was calculated by the Analyze Particles function in the ImageJ. The average size of Ca^{2+} positive region/diffusion area was calculated from different 20-min time-lapse imaging data.

To calculate the local Ca^{2+} wave velocity, as presented in **Fig. 3E**, we first manually defined two lines (separated from each other by ~50 μm) parallel to the wavefront. Subsequently, we measured the time required for the wavefront to traverse the distance between these two lines. The wave speed of these local ICWs was determined by dividing the distance by the time.

To assess the Ca^{2+} oscillation correlation, two adjacent cells exhibiting Ca^{2+} oscillations were randomly selected. The sequences of Ca^{2+} intensity changes over time for these two cells were individually obtained. Subsequently, the absolute value of the Pearson correlation coefficient between the two sequences was calculated."

Fig. 1G, H. There is no description of how to prepare the conditioned medium and how to treat with DNase and RNase enzymes.

Response: We have added the following description in the method section: "For the preparation of brain-conditioned medium, 20 brains from 3rd instar larvae were initially dissected in PBS and then transferred to 400 μl of complete Schneider's medium and incubated for 1 hour at room temperature. Subsequently, the supernatant of the incubation was collected and referred as the brain-conditioned medium. To treat the conditioned medium, proteinase K (0.1 mg/mL), DNase I (1 U/mL), or RNase A (0.1 $\mu\text{g/mL}$) was added and incubated at 37°C for 2 hours. For filtering the conditioned medium, a 0.5 mL Pierce 10K MWCO concentrator (ThermoFisher 88513) was used. The filtration was conducted on a benchtop centrifuge at 15,000g for 30 minutes and approximately 200 μl of the filtered medium was collected for the treatment of the

cultured fat body.”

Fig.2A There is no description of how the peptide used was prepared.

Response: We have added the details about the peptide preparation in the method section: “For the neuropeptide treatment experiment, neuropeptide stocks were prepared by dissolving peptide powder (synthesized by Genscript and sequences shown in **Supplementary Table 2**) in either water or DMSO to a concentration of 1 mg/mL depending on solubility. For peptides with disulfide bonds (AstC, CCHa1, and CCHa2), 100 µg/mL peptides were treated with 10% DMSO in PBS at room temperature for 6 hours. The efficiency of the reaction was assayed by Ellman reagent (DTNB assay kit, Solarbio BC1175), and more than 85% of the peptides were oxidized. During live imaging of the cultured fat body, the samples were first recorded for 5 minutes (with a time interval of 5 seconds) to generate a baseline. Subsequently, the medium was replaced by Schneider's medium containing 100 ng/mL of different synthesized *Drosophila* neuropeptides, and recording was carried out for 20 minutes (time interval: 5 seconds).”

Fig.2D It is unclear whether AKH is adequately removed by the addition of the antibody. Experiments using control antibody is also necessary. However, since the requirement of AKH and AKHR has been shown genetically, antibody experiment may not be necessary.

Response: Judging from the rapid decrease in Ca²⁺ activity, we infer that the antibody effectively removes available AKH. Additionally, we have included a control rabbit serum, as suggested by the reviewer, in **Figures 2D-E**.

Fig.S3B. Explanation of the starvation experiment is needed.

Response: We have added a detailed description of the starvation experiment in the figure legend:” Early 3rd instar digging/feeding larvae with fat body-specific expression of *GCaMP5G-T2A-mRuby* were collected from standard lab food, washed, and transferred onto a 2% agarose (in water) plate for starvation. After 2 h or 8 h of starvation, more than 10 larvae were kept on 2% agarose and imaged for 20 minutes under free-behavior conditions. The period of the ICWs from head-to-tail of the larvae was calculated in ImageJ.”

Fig.3E. How did you quantify “control local”?

Response: We have added a detailed description in the image processing and quantification method section: “To calculate the local Ca²⁺ wave velocity, as presented in **Fig. 3E**, we first manually defined two lines (separated by ~50 µm) parallel to the wavefront. Subsequently, we measured the time required for the wavefront to traverse the distance between these two lines. The wave speed of these local ICWs was determined by dividing the distance by the time.”

Fig.3J-K. AKHR is stimulated by AKH, which promotes TAG catabolism. The involvement of ICWs for TAG catabolism is not clear.

Response: In studies on lipid metabolism in *Drosophila*, Ca^{2+} signaling and Gαq activity have been shown to promote triglyceride (TAG) catabolism and reduce fat storage (Baumbach et al., 2014, DOI: 10.1016/j.cmet.2013.12.004; Baumbach et al., 2014, DOI: 10.1016/j.jgg.2014.03.005; Maus et al., 2017, DOI: 10.1016/j.cmet.2016.12.021).

In our work, based on *in vivo* calcium imaging experiments in both larvae and adult flies, AKH is identified as the primary regulator of Ca^{2+} increases in the fat body, supporting the notion that Ca^{2+} is a downstream effector of AKH that controls TAG catabolism. To further substantiate our claim, we added measurements of TAG levels in adult flies following the modulation of Ca^{2+} levels by knocking down SERCA or Gαq (**Supplementary Fig. 4E**). In addition, AKH is considered a functional homolog of human glucagon which acts in a similar fashion to increase cytosolic Ca^{2+} which subsequently mediates hepatic lipolysis in mammals (Perry et al. 2020, doi.org/10.1038/s41586-020-2074-6).

Fig. 4F-H. CO₂ treatment reduces APC activity, but the relationship between APC Ca²⁺ dynamics and AKH release has not been directly shown. I am also wondering whether the half-life of AKH can be directly measured?

Response: We have demonstrated that silencing APC with Kir2.1 expression significantly reduced AKH release (**Fig. 2G**), supporting that AKH release requires neuronal activation. Additionally, Ca^{2+} increases have been considered by most previous studies as a reliable indicator of neuronal activation. We also show that Ca^{2+} increases in the APCs are triggered by Thr-containing medium, which also elicits AKH release from the APCs (**Fig. 8B-G**). Collectively, these data support the positive correlation between increased Ca^{2+} activity in the APC and AKH release.

Regarding the half-life of AKH, direct *in vivo* measurement remains challenging. In larger animals, the half-life of a hormone in circulation can be determined by continuously collecting blood over a specific period. However, collecting sufficient hemolymph from flies for dot blot or ELISA would take more than 10 minutes and require multiple flies, which makes measuring a half-life of less than several minutes impractical. Furthermore, since AKH release is not synchronized between flies, pooling hemolymph samples would also confound the estimation of the half-life.

Based on our brain-conditioned medium experiment, AKH released into the cell culture medium can continuously activate the fat body for several hours, suggesting that AKH is not inherently unstable. (Note, this data also suggests that the fat body does not desensitize to AKH by itself.) In the case of insulin, which has a half-life of several minutes in the bloodstream, degradation primarily occurs in the liver in mammals. Therefore, we suspect that AKH is degraded by certain organs such as the gut or muscle. We have added other dissected gut or muscle/cuticle, to the brain-conditioned medium to test whether these organs promote the degradation of AKH, but observed no significant effect. It is possible that the other organ(s) is responsible for AKH clearance in flies, or that the current *ex vivo* system is not optimal for the proper functioning of the cultured organs.

Figure 5I. The authors attribute the difference in ICW response between larvae and adults to the uniform distribution of extracellular AKHs in adults. However, the argument that this difference is due to the location of APCs in the thorax seems to be weak; it is possible that AKH is abundantly secreted in the adult body fluid, as is the case with the Akh overexpression movie in the fat body in Figure 3D, where ICW occurs in a random direction. This can be test by comparing AKH levels in the blood lymph of larvae and adults by western blot or ELISA.

Response: We thank the reviewer for the constructive suggestion and have conducted a dot blot analysis to compare AKH levels in the hemolymph of larvae and adults. However, we did not observe any significant differences in AKH levels between larval and adult hemolymph (**Supplementary Fig. 8**).

We then hypothesized that the difference might lie in the sensitivity of the fat body to the AKH ligand. To test this, we dissected fat bodies from both larvae and adults and treated them to different concentrations of AKH peptide. Interestingly, adult fat bodies exhibited a stronger response to higher concentrations of AKH (>10 ng/mL) compared to larval fat bodies, as evidenced by a greater proportion of adult fat cells showing a significant increase in Ca²⁺ (**Supplementary Fig. 8B**). However, within the 1-10 ng/mL range of AKH, which elicits Ca²⁺ increases comparable to those observed in vivo, both adult and larval fat bodies displayed similar levels of sensitivity.

In summary, our data indicate that there is no significant difference in AKH concentration or tissue sensitivity between adult and larval fat bodies. Interestingly, a recently published paper by Laura Blackie et al. (2024 DOI: 10.1038/s41586-024-07463-4) revealed that there are complex interorgan connections (shape and arrangement of organs) in adult flies. Meanwhile, our observation of AKH-triggered global intracellular waves (ICWs) in larvae suggests that hemolymph flows efficiently, facilitating the rapid transport of secreted AKH from the head to the tail. Therefore, a potential increase in interorgan connections in adult flies may impede the swift transport of AKH and promote a more uniform AKH concentration in the abdomen.

Meanwhile, we do not rule out the possibility that potential modulators of AKH signaling may differ between larvae and adults, which could account for the observed differences. Additionally, the APCs in adults may release AKH in a continuous rather than pulsatile manner. We have included these potential explanations in our discussion.

Fig. 5I. The authors propose a model of pulsatile AKH secretion in larvae. However, the rhythm of AKH secretion itself is unclear. Considering that the velocity of bead movement in Fig. 4E (250 $\mu\text{m/s}$) is much faster than the wave velocity of global ICWs in Fig. 3E (40 $\mu\text{m/s}$), it might be more relevant to assume the desensitization speed of the fat body rather than focusing on AKH secretion.

Response: We also suspected that the fat body might desensitize upon continuous AKH exposure. However, treating the cultured fat body with either synthesized AKH peptide or brain-conditioned medium elicited sustained Ca²⁺ oscillations, suggesting that no significant desensitization happens at the physiological level of AKH. Desensitization was observed with extreme AKH concentrations above 1 $\mu\text{g/mL}$ (**Supplementary Fig. 3A**). At this non-physiological AKH concentration, all fat body

cells showed elevated Ca^{2+} levels, which are not observed *in vivo*. In addition, at these abnormally high AKH levels, Ca^{2+} oscillations in the fat body remained detectable. This suggests that AKHR is a GPCR that does not desensitize easily. This may be consistent with the short half-life of AKH *in vivo*. Since the ligand itself is short-lived, the receptor does not need an additional layer of signal shutdown mechanism.

Regarding the difference between hemolymph velocity and AKH propagation, a high blood flow velocity does not imply that a high concentration of AKH will travel at the same speed. Once released from APCs, AKH can be considered to be stored locally within the body cavity of the larval head region. This is supported by the observation that stopping the heartbeat with CH_3Cl treatment traps AKH in the head region, eliciting only a local Ca^{2+} response (**Fig. 4A-C**). The circulating blood flow in the larval body does not carry all the AKH at once but rather gradually dilutes the pool of AKH in the head region, carrying portions of it along with the flow. The rate at which AKH is diluted and transported away from the head region depends on the volume of the head cavity and the flow rate of the circulating hemolymph. This compartment model is widely used in physiologically based pharmacokinetic modeling when studying drug transport in animals.

One way to illustrate the model is to imagine a “lake of AKH”, and the bloodstreams can be imagined as many flowing rivers. Some of these rivers flow through the lake, carrying portions of AKH with them as the AKH pool exchanges material with the hemolymph circulating nearby. And the downstream AKH concentration is the result of the mixing of these rivers. Over time, the AKH in the 'lake' gets diluted and carried away by the rivers but at a much slower speed than the flow speed of an individual river. To illustrate this, we have added a plot of simulated AKH concentration in **Fig. 5B**.

We thank the reviewer for pointing this out. Our previous illustration of the AKH gradient in **Fig. 5K** was incorrect and has been revised.

*Why is the Calcium intensity increased in *inx2*-RNAi? Based on Suppl Fig. 7B-D, is there a mechanism that delays AKHR desensitization? Could the suppression of local ICWs lead to a compensatory increase in global ICWs? Since this point is also supported by mathematical models, further discussion based on the mathematical model on which factors are important would be possible.*

Response: We thank the reviewer for pointing this out. The increase in Ca^{2+} observed after *inx2*-RNAi in larvae is a surprising finding. From our computational simulations, the Ca^{2+} increase appears to originate from internal stochastic fluctuations within the system. When the random fluctuation factor is removed, the Ca^{2+} increase no longer occurs in *inx2*-RNAi. Additionally, the inhibition of gap junctions indeed caused a significant increase in variance as the wave propagates from the head to the tail (Fig. 5I-J), which is consistent with the idea that gap junction knockdown weakens the coupling between neighboring cells.

We currently hypothesize the following mechanism: stochastic fluctuations in Ca^{2+} generate a Ca^{2+} cycle/status that is not in phase with the external AKH ligand, thereby interfering with and partially 'canceling' the effects triggered by AKH. With functional

gap junctions, these fluctuations can propagate between cells and affect more cells. However, with *inx2*-RNAi, the local Ca^{2+} fluctuations can no longer propagate, resulting in a reduced 'canceling effect' on the external AKH signaling, and subsequently a stronger effect by external AKH. That model can also explain why Ca^{2+} decreases after *inx2*-RNAi, as in adults, a uniform external AKH triggers random Ca^{2+} activation which requires gap junctions for propagation.

As the reviewer suggested, the overall effect is that the decoupling of neighboring cells decreases local ICWs but promotes global ICWs. We have added this explanation in the revised discussion section.

Fig.S11. What is the status of AKH in normal food?

Response: We added the comparison of AKH retention in APCs between normal food, 5% sucrose, and starved larvae in **Supplementary Fig. 12D**.

Minor Points:

1. The velocity measurement experiments using beads in larvae shown in Fig. 4D are excellent. However, compared to the videos of adults, the beads appear to get stuck in the tissue, preventing them from moving. Additionally, the fact that many beads are not moving is a concern. While using smaller beads may be difficult, it would be advisable to mention these limitations as technical constraints.

Response: We agree with the reviewer that the speed of the beads may be lower than the actual velocity due to collisions and interactions with other organs. However, even this lower bound of the speed is still much higher than the traveling speed of the global ICWs, suggesting that the circulating hemolymph is sufficiently fast for the transport of secreted AKH. As we mentioned above, the compartment model for material transport within animals only requires that the flow speed between two compartments be greater than the observed transport speed of the carried substance. We have added this limitation of the method to our discussion.

2. The notation for the fat body Gal4 driver in Figure 1 varies between Fat>, Lpp>, and Fb>. It would be better to standardize the notation.

Response: We have corrected the notation. In this study, we used two fat body-specific Gal4 drivers, Fb-Gal4 and Lpp-Gal4, both of which have been employed for fat-specific expression in many previous studies. Our initial experiments utilized Fb-Gal4. However, we later switched to Lpp-Gal4 because it has more available tools built around it. No difference in Ca^{2+} dynamics was observed when using the two Gal4 drivers to express GCaMP5G in the fat body.

3. The choice of GCaMP5G raises some questions. Compared to GCaMP6s, the $\Delta F/F$ is lower (Nature. 2013 Jul 18; 499(7458): 295–300). And GCaMP6s was used in the ICWs and SIDICs papers in wing discs. Was this lower sensitivity of 5G advantageous for detection of ICWs in the fat bodies?

Response: We have tested different GCaMP variants, including GCaMP5G, GCaMP6s, and GCaMP7s, in the fat body and found that GCaMP5G provides the best

signal-to-noise ratio in adipose tissue. Because the newer GCaMPs have a higher affinity for Ca^{2+} : the K_d for GCaMP5G is 0.45 μM , whereas the K_d for GCaMP6s is 0.144 μM (Jacob L. Perry et al., 2015, 10.1016/j.ymeth.2015.09.004). The new GCaMPs (6 and higher) showed a much higher basal signal, resulting in a smaller effective $\Delta F/F$ compared to GCaMP5G. As the reviewer suggested, we found that the lower sensitivity of GCaMP5G better matches the cytosolic dynamics in fly adipose cells.

*4. The differences in response in adult and larval FB are very interesting. If data were available, could they be discussed in terms of differences in localization patterns of AKHR and *inx2* expression? What about differences in fat amount or metabolic demands during development and maturation stages?*

Response: We thank the reviewer for suggesting this interesting direction. We have conducted experiments to explore this possibility. As mentioned above, no significant difference in AKH sensitivity was observed between larval and adult fat during ex vivo experiments (**Supplementary Fig. 8B**). Both adult and larval fat showed a significant reduction in the correlation of Ca^{2+} activity between neighboring cells, suggesting that *inx-2-RNAi* affects both larval and adult fat similarly under ex vivo conditions (**Supplementary Fig. 6C-F**). Thus, our data suggest that the differences between adults and larvae after *inx-2-RNAi* are fat non-autonomous and likely to be caused by factors other than the fat body, as we have discussed above. In terms of fat amount or metabolic demand, we currently lack effective assays to measure and compare these parameters between stages. Nevertheless, as discussed above, both our experimental data and computational simulations support the existence of differences in the dynamics of external AKH.

5. There is character encoding issue on the y-axis label for Velocity in Suppl Fig. 8B.

Response: We have corrected the error.

6. Is the term "large epithelial tissue" appropriate to describe the fat body in line 518?

Response: We have removed "large" in the sentence.

7. It would be better to unify the use of the abbreviation for insulin-producing cells (IPCs) throughout the manuscript.

Response: We have changed the abbreviation as the reviewer suggested.

*Reviewer #2: This is a well-written and thorough manuscript uncovering the role of Ca^{2+} signaling in the *Drosophila* fat body in regulating fat storage mobilization. As mentioned in the Discussion section, this study also provides insights into the Liver-alpha-Cell axis in human biology. In this context, Ca^{2+} signaling is stimulated and acts via hormonal (Adipokinetic hormone, AKH) signaling from specialized cells (APCs). AKH is sensed by the fat body cells expressing the AKH-Receptor (AKHR), a G-Protein Coupled Receptor (GPCR), that activates *Gaq* to stimulate Ca^{2+} signaling. The authors investigated the mechanism of transport of AKH, which is circulated in the*

hemolymph throughout the larval body and is secreted from the APCs, and how the fat body is able to respond to this hormonal signaling with two distinct types of intercellular calcium waves (ICWs). The authors argue that two distinct mechanisms of ICWs dynamics are active in larval fat body: ICWs are spread by either intercellular gap junctions channels providing a slow, local spread, or alternatively, through convective transport due to hemolymph flow that results in extensive calcium waves, initiated through AKHR-Gaq signaling. They evaluated the differences in calcium signaling between larval and adult stages in the fat body. This aspect of the story mostly confirms and extends previous findings already described in the literature regarding the role of Galphaq, PLC Beta and Ca²⁺ signaling in the fat body. A new and significant finding is that this AKH hormonal signaling in larvae, in addition to hunger, can also be triggered by dietary addition of two amino acids, including Methionine and Threonine. These amino acids are sensed by APCs and stimulate AKH release, triggering ICWs and fat mobilization in the fat body. The manuscript is logically organized. The data is robust, and most of the authors' data interpretation and conclusions are reasonable.

However, some of the results are not unexpected. The authors should clearly distinguish what is a confirmation of the already known findings and what are the new findings. The claimed new results are stated in the abstract, citing "We identified Adipokinetic hormone, ... as the key factor driving Ca²⁺ activities in adipose tissue," it should be restated more circumspectly as "we confirmed previous reports with an additional line of evidence." This result is not a completely new finding, and was described previously in the literature. Furthermore, the GPCR AKH-Receptor works via Gaq also was described previously in the literature (for example, Bumbach "Gaq, G γ 1 and Plc21C Control *Drosophila* Body Fat Storage" <https://doi.org/10.1016/j.jgg.2014.03.005>).

Response: We thank the reviewer for the comments and have rephrased the sentences in the manuscript accordingly.

A significant new finding is that authors are claiming that the larval fatbody exhibits two distinct types of ICWs. thors tested a variety of amino acids and found that specifically methionine and threonine are sensed by APCs to trigger AKH release, which in turn stimulates ICWs in the body. This suggests that these two amino acids are able to trigger "starvation"-like responses. It is possible that foods that are rich in these two amino acids are not the preferred/best food sources for the larvae, and if given a choice they would not choose to have this nutritionally "poor" diet. As in another study where one of the coauthors had participated (<https://doi.org/10.1038/s41586-022-04960-2>) about adult fly preference of Leucine in the food. In this study Leucine did not trigger APCs activation, clearly demonstrating that Leu is a preferred dietary amino acid for fly larvae. According to flybase expression data, AKHR is expressed in the larval fat body, but in adults, it is strongly expressed in the head. This suggests that AKH could be sensed in adult heads and could potentially trigger behavioral changes in adults to search for new food sources, for example. The impact of this finding is significant as it has the promising potential for application towards the creation of new medication for

obesity, heart disease, and diabetes for humans.

There are a couple of areas for improvement in the present manuscript. A significant finding in the manuscript is that ICWs can happen independently of gap junctions, and therefore are not spread cell-to-cell, but rather reflect an extracellular gradient of hormones. But this finding could be bolstered further. This claim is based on using Inx2-RNAi-mediated knockdown, but no verification was made to ensure that the Inx2 knockdown was 100% knockdown of gap junction activity. Even though pharmacological treatment was used, it was performed in cultured fat bodies and not in vivo, so it can not be used as a substitute for in vivo verification. Also, it is not explicitly mentioned what is the sample size in these experiment. For example the ex vivo experiment in Fig. 3.

Response: We greatly appreciate the reviewer for the constructive suggestions, which provides a very interesting direction for our future studies. To verify the efficiency of Inx2-RNAi, we first tested the efficiency by RT-PCR and found over 85% reduction in the Inx2 mRNA in larval fat body (**Supplementary Fig. 6B**). We also realized that reduction in mRNA or protein level is not necessarily equivalent to the function reduction. Because the primary function of gap junction is to couple Ca²⁺ activity in neighboring cells (in this study), we compared the Ca²⁺ signal correlation between neighboring cells with and without Inx2 knockdown. In both larval and adult fat, Ca²⁺ activities are highly correlated with a Pearson correlation coefficient close to 0.8-0.9. In contrast, the correlation efficiency decreased to 0.2 after Inx2 knockdown (**Supplementary Fig. 6C-F**), suggesting that Ca²⁺ activity in neighboring cells is significantly uncoupled in Inx2-RNAi fat cells.

As for the sample size, we have at least three different biological replicates in each experiment. As we have displayed actual data points in our bar plots, we did not include the n number in the figure legend, instead, we have provided a source data file in Excel showing all the original data used for all the plots (**Source data 1**).

Another concern regarding the clarity of presentation of results is that in Supplementary Video 8. The Luc-i condition includes global ICWs (the waves are extensive), although the authors state completely opposite, that no global ICWs were observed in adults, and that global ICWs are stage-specific, only attributed to the larval fat body. The definition of global vs. local ICW is vague in this case. Also, the sample size of this study should be more clearly identified.

Response: We thank the reviewer for raising this question and have revised our manuscript to better define the difference between global and local ICWs. Briefly, both local and global ICWs involve multiple fat cells being activated simultaneously. For example, in ex vivo cultured wild-type larval fat, there are extensive cells exhibiting collective Ca²⁺ activities (**Supplementary Video 5**). However, the fundamental difference between global and local ICWs lies in whether the ICWs need to be understood at the organism level. For global ICWs, there is a single pacemaker at the head, the APCs, and the wave travels throughout the entire larval fat body from head to tail. Thus, the whole organism must be considered to understand this phenomenon.

In contrast, ICWs in the adult stage show multiple independent pacemakers, and Ca²⁺ activity spreads regionally. Therefore, studying a 'local' region of the adult fat is sufficient to recapitulate the features of the ICWs.

The Inx2-RNAi line used in the study is made using Valium10 vector, and lpp-Gal4>GCamp5G driver does not seem to have Dicer2. ICWs in the Inx2-RNAi look very similar to the control, there could be an incomplete penetrance phenotype for this Inx2-RNAi line. To strengthen the result, a secondary method such as CRISPR would be needed. Even though a pharmacological gap junction inhibitor was used in cultured fat bodies the results agreed with in vivo Inx2-RNAi results. To verify this result, complete inhibition is needed, perhaps with stronger inhibition, such as a second distinct line for Inx2-RNAi, or by repeating the in vivo experiment as in Supplementary Video 6 using another driver that has Dicer2 in its genotype that will help to improve the knockdown or any other way that will provide a concrete evidence that Inx2-RNAi mediated knockdown was fully effective. This is an significant concern because the authors claim the novel finding of a new previously undescribed mechanism of ICWs based only on the results with Inx2-RNAi knockdown (which may not be 100% knockdown). Therefore, the basis of this novel finding should have more parallel lines of evidence.

Response: We agree with the reviewer that the knock-down efficiency of Inx2-RNAi is important. Therefore, we have performed a functional assay to assess the intercellular coupling efficiency between neighboring cells. The results suggest that the function of gap junctions is significantly inhibited by the Inx2-RNAi (**Supplementary Fig. 6C-F**).

Meanwhile, the propagation speed of the global ICWs is ten times faster than that of classical ICWs mediated by gap junctions and is also much faster than the free diffusion speed of Ca²⁺ in water. This propagation speed cannot be explained by gap junction spreading. Instead, we found that the fast-circulating hemolymph accounts for this rapid global spread of the signal (**Fig. 4**).

It would also be helpful to perform the same experiment as in Video 8 but in cultured adult fat bodies and then update Fig. 3J, K. The adult cuticle is only partially transparent and prevents the observation of the full range of GCaMP5G, a Ca²⁺ sensor, will be out of the way, and the results will be more clearly demonstrating the actual phenomenon of ICWs.

Response: We have performed ex vivo culture of the adult fat body with and without Inx2-RNAi in **Supplementary Fig. 6E-F**. Live-cell imaging of the ex vivo cultured adult fat treated with AKH is shown in **Supplementary Video 7**.

Specific genetic background descriptors are needed in each and every video and image rather than the nondescriptive word "control," because this introduces a guesswork for the reader. This abbreviation is used sometimes in the text and Videos, and it is not easy to find out what exactly was used as a control each time. The same applies to the driver lines. As an example: Video 6, that has labels "control" (what was

a control?), Inx2-i (what was the driver?), AKHR-null, and AKH-OE (again, was the driver used?). It creates extra work for the reader to refer to the SI files. Also, it would be more clear if the location of the head is indicated, to clarify the direction of the ICW spread.

Response: We have revised the labels in the figures and videos as the reviewers suggested.

Lines 198-199: Please provide a more detailed description of creation of kymographs. Supplementary Video 2: it is not clear where the brain is on the video. There is no clear larval brain outline.

Response: We have updated the method section to include details on the generation of kymographs.

As for **Supplementary Video 2**, because the brains were added after the imaging chamber was prepared, they typically floated in the medium and were out of the imaging field (this is now described in detail in the updated method section and illustrated in **Supplementary Fig. 1D**).

Lines 42-90, the Introduction section: More than half of the introduction section (lines 64-90) was used to summarize results. Normally, the Introduction section should provide the background necessary for the reader to understand the following “results” section. Furthermore, the Introduction section did not fully cover the scientific literature on the topic of Calcium signaling. It would be helpful to condense the summary of results in the Introduction, and instead add more background information on the Ca²⁺ signaling. For example, there is a Baumbach, Xu, Hehlert, Kuhnlein 2014 paper that reports Gaq-related Calcium signaling in the fat body and its connection to the AKH and AKHR. (<https://doi.org/10.1016/j.jgg.2014.03.005>) and others.

Response: We have rewritten the introduction section and included the suggested references.

Lines 104-106 where the authors describe the experimental setup: providing a cartoon or a diagram will improve clarity of the experimental set up.

Response: We have added a schematic illustration in the **Supplementary Fig. 1D**.

Lines 125-130: the authors should reference Supplementary Figure 1 A.

Response: We have added the reference.

(Minor issue): The lines 369 and 370 cited source 21 as the location of APCs, but it is not a best resource for describing the location of the APC location in adults, as in Figure 1 of the cited source 21 the location of APCs are shown in the head, which is contrary to what authors are saying. Perhaps an alternate source should be cited regarding the location of the corpora cardiaca DOI 10.1007/s00018-015-2063-3, fig. 1 or fig. 2). The authors are identifying the APCs location correctly, but they need to cite an alternate source.

Response: We thank the reviewer for the suggestion and have revised our manuscript

accordingly.

In general, the introduction section should provide a more contextual literature summary, including ICW in wound detection and healing.

Response: We have revised our introduction to emphasize this aspect.

L77: Modify to the plural: computational simulations, unless only one simulation was implemented.

Response: We have corrected this.

L99: be more specific on this point: “using a holder (what kind) in the perfusion chamber”

Response: We have added a detailed description and reference: “Dissected brains were immobilized using a slice holder with nylon grids (SH13, Scientific Systems Design Inc.) in the perfusion chamber.”

L172 : wave direction analysis methods could have been clearer. Consider revising for clarity.

Response: We have added a new section about image quantification in the updated method:” Kymograph of Ca²⁺ waves were used to highlight the directive propagation of the wavefront in the fat body of the 3rd instar larva. First, an average density projection was executed on each frame along the Y-axis (representing the direction from the head to the tail of the larva). This process compressed each frame into a solitary line of pixels, spanning from the head to the tail. Subsequently, all these lines from each frame were combined into a kymograph, with time as the X-axis, to illustrate the direction of Ca²⁺ wave propagation.”

L. 186: Is there a reference that needs to be added? “According to the concept of connected components in image processing...” this is not clearly defined.

Response: We have revised the sentence. We have added a description of how the processing is done:” To calculate the Ca²⁺ diffusion area, as presented in **Fig. 3C**, we first used threshold function to select all Ca²⁺ positive cells in the fat body. Then the area of individual Ca²⁺ positive regions in the field was calculated by the Analyze Particles function in ImageJ. The average size of Ca²⁺ positive region/diffusion area was calculated from different 20-min time-lapse imaging data.”

L.215: Where is the description of how the transgenic lines were created that were first used in the study

Response: We have added the information on cloning and transgene generation in the method section.

L.271 need citation for Kir2.1 reference

Response: We have added the reference.

L. 506 : Does insulin stimulate Ca²⁺ in fat body like on other tissues such as the wing disc?

Response: We have tested the effect of (100 nM) recombinant human insulin on cultured fat bodies but found no significant increase in Ca²⁺ level.

(For the reviewer, cultured larval fat bodies were treated with 100 nM Insulin or 100 nM AKH)

L640 : Making the data available on repository will aid in transparency of results

Response: We have included source data with original data for all the plots.

The title is somewhat misleading: Novel Calcium waves are not a class of calcium waves. It may be a novel finding, but that is also overstated.

Response: We have changed our title to “Dietary Amino Acids Promote Glucagon-like Hormone Release to Generate Global Calcium Waves in Adipose Tissues”.

Computational codes: The study is only replicable if the codes are deposited to Github. The SI text was not fully transparent in describing and annotating computational code. the Statement “all MATLAB code in the simulations are fully disclosed” cannot be verified.

Response: We have uploaded our codes onto the GitHub with the following linkage: https://github.com/Sy-Luo/HeLab_CalciumWave_Simulation_2024.

It was not completely clear how Ca²⁺ traces are measured (E.g. SI Fig 3E). Is the ROI a particular cell or area and does it differ from experiment to experiment?

Response: We have added an image processing section in the method:” The temporal dynamics of Ca²⁺ activity within a specific fat cell or region were plotted using ImageJ. Specifically, a region of interest (ROI) containing the cell or region of interest was first selected in a multi-frame live-image stack. Then, the “Plot Z-axis Profile” in the Stack function group was employed to trace the Ca²⁺ level within the ROI over time.”

Fig. S4: Why are there two rows for each condition? The darker second row is not explained. The heat maps of rel. F.I. are also not defined.

Response: We have added the description in the new processing procedure section: “To clearly visualize the calcium (Ca²⁺) dynamics within the fat bodies, background fluorescence that does not vary over time was eliminated by subtracting an averaged

image (calculated through the mean projection of the time-lapse sequence) from each frame. Note that fat cells damaged during dissection typically exhibit a consistently high Ca^{2+} signal, which is also removed by this process. The edges of the tissue were delineated with the Canny edge detector in MATLAB.”

Definitions like global or tissue-wide Ca^{2+} waves are not clearly delineated, which makes it difficult to assess the comparisons being made between the larval and the adult experiments. Does a global wave mean that the whole fat body is activated at the same time or that the wave progresses through the whole body?

Specifically, how can the effect of GJ inhibition leading to increased Ca^{2+} amplitude and decrease in frequency be explained in the computational model?

Response: We have discussed the key differences between global and local ICWs above and have revised the manuscript accordingly. Our current hypothesis regarding the observed increase in Ca^{2+} after *inx2* knockdown is discussed in our response to Reviewer 1. Essentially, this can be viewed as a compensatory effect resulting from the inhibition of local ICWs.

Reviewer #3 (Remarks to the Author):

I co-reviewed this manuscript with one of the reviewers who provided the listed reports as part of the Nature Communications initiative to facilitate training in peer review and appropriate recognition for co-reviewers.

Reviewer #4 (Remarks to the Author):

*Reviewer #5: The manuscript by Ahmad et al. describes a novel mechanism of nutrient sensing and metabolic regulation in the fruit fly *Drosophila melanogaster*. The study demonstrates that Adipokinetic hormone (AKH), the fly equivalent of mammalian glucagon, stimulates intracellular Ca^{2+} waves in the larval fat body to promote lipid metabolism. Specific dietary amino acids are shown to activate AKH-producing cells, leading to increased intracellular Ca^{2+} and subsequent AKH secretion. The study also reveals different mechanisms of Ca^{2+} dynamics regulation in larval and adult fat bodies, highlighting the interplay between AKH secretion pulses, extracellular hormone diffusion, and intercellular communication through gap junctions.*

While it is disappointing that the precise biological significance of the phenomenon has yet to be fully understood, this manuscript represents a significant advancement in our understanding of nutrient sensing and metabolic regulation. The study utilizes innovative techniques and provides novel insights with potential implications for human health.

Overall, this is a very nice study, and the reviewer has only a few comments on the manuscript for revision.

(1) The authors' computational modeling is impressive as it provides a new strategy for Drosophila physiology. The reviewer is curious about the relationship between the predicted distribution of AKH in the body and the predicted Ca²⁺ influx in the fat body in this model. The reviewer would like the authors to show the expected distribution of AKH together with the Ca²⁺ data in Figure 5A, and discuss the relationship between AKH distribution and Ca²⁺. The reviewer thinks that it would be of interest to many Drosophila researchers.

Response: We thank the reviewer for the suggestion and have added new figures illustrating the simulated pattern of AKH concentration over time in **Fig. 5B and E**. We assumed that all AKH was released from the head at time 0 and then gradually diluted and transported along the long axis of the larva. The global ICWs move with the front of the AKH diffusion, as the Ca²⁺ in the cells begins to rise when local AKH concentration reaches the activation threshold in the model, while fat cells behind the wavefront gradually cycled into the low Ca²⁺ phase. Our previous illustration of the AKH concentration in Fig. 5K was incorrect and has been revised.

(2) Please carefully check Supplemental Table 2 and review what kind of peptides the authors really used. The reviewer realizes several concerns.

a) The peptide sequence of CAPA listed in this table (GDAELRKWAHLLALQQVDL) is entirely different from an actual CAPA mature peptide (GANMGLYAFPRV-amide; see <https://journals.plos.org/plosone/article?id=10.1371/journal.pone.0029897>).

Response: We thank the reviewer for pointing this out. We have thoroughly checked the peptide used in our study and corrected the errors as requested. For the CAPA peptide, we have synthesized a new CAPA with the correct sequence suggested by the reviewer and redone the fat body test in **Fig. 2A**.

b) While the authors used FQYSRGWTN-amide as Corazonin, the reviewer wonders why the authors used such a shorter peptide than the identified mature Corazonin peptide (QTFQYSRGWTN-amide).

Response: We have synthesized the full Corazonin peptide suggested by the reviewer and redone the fat body test in **Fig 2A**.

c) While some peptides, such as Ast-C, CCHa1, and CCHa2, have internal disulfide bonds to be active forms, the authors did not state whether they use the proper mature peptides. If the authors used the "wrong" peptides, they did not correctly examine the effect of some peptides, and the reviewer argues that the authors must carefully revise Supplemental Table 2 and related sentences.

Response: We thank the reviewer for pointing this out. We have now used DMSO treatment to form the internal disulfide bonds in all three peptides (Ast-C, CCHa-1, and CCHa-2), and have redone the fat body Ca²⁺ activity assay using the oxidized peptides. The efficiency of disulfide bond formation was determined to be higher than 80% using the DTNB Ellman's reagent.

The reviewer also has additional comments as follows.

d) "LK" and "Leukokinin" are exactly the same, and there is no need to include both data in Figure 2A.

Response: We have corrected this error.

e) The reviewer does not see any particular reason why some very well-known peptides, such as NPF and DH44, were not included. If there are too many inadequacies here, changing the logic flow that focuses on AKH may be necessary.

Response: We have added NPF and DH44 as the reviewer suggested. Currently, we have tested 32 peptides out of the approximately 40 known neuropeptides in *Drosophila* that have GPCR receptors (according to the reviews by Dick R. Nässel and Meet Zandawala, 2019, doi: 10.1016/j.pneurobio.2019.02.003). For neuropeptides with multiple sequences, we tried to choose the most studied or abundant ones based on previous literature. We have added this statement in our methods section for clarity.

This study originated from a discovery made six years ago, so some of the most recently studied neuropeptides were not included. We also agree with the reviewer that AKH would be a good candidate after the discovery of fat body Ca^{2+} dynamics. However, we believe that the peptide list, even though not comprehensive from the current perspective, still provides important information on how the fat body responds to most known neuropeptides through Ca^{2+} signaling. Additionally, it includes the actual experiments that led to the identification of AKH in this study.

Updated Point-by-point responses to the reviewers' comments:

Reviewer #1 (Remarks to the Author):

The revised version of the manuscript by Ahmad et al. is significantly improved compared to the original submission, as the authors have addressed most of our concerns and the points raised by the other reviewers (although it was difficult to find changes in the text).

However, the following points still need to be addressed by the authors.

Main point:

The experiment in Figure 4 suggests that a beating heart is required for ICW and Figure 5K suggests that there is a pool of AKH in the head. The formation of an AKH pool in the head could be either pulsed or sustained release of AKH from the APC. It is possible that transport from the pool is the main mechanism for ICW. Therefore, the statement that AKH release is pulsed should be tempered.

Response: We agree with the reviewer that direct evidence of pulsatile release of AKH from APCs is still lacking. Therefore, we removed this claim and proposed that the pulsatile release of AKH is a potential explanation of the periodic global ICWs that require further investigation in the discussion section. (Changes in the main text are highlighted in yellow.)

Minor points:

Fig.1G, H. There is no description of how to prepare the conditioned medium and how to treat with DNase and RNase enzymes.

Response: We have added description in the figure legends: "Brain-conditioned medium, which was prepared by incubating 20 dissected larval brains of the third instar with 400 μ l of Schneider's medium for one hour, was utilized to treat the isolated fat bodies.", and "Brain-conditioned medium was treated with 0.1 mg/mL proteinase K, DNase I (1 U/mL), or RNase A (1 μ g/mL) for 1 hour at 37°C and then filtered with Pierce 10K MWCO concentrator." A more detailed description of the experiment can be found in the method section "*Collection and treatment of conditioned medium*".

The authors mention that it was incubated in Complete Schneider's medium for 1 hour, but Complete Schneider's medium contains 10% FBS. Did 10% FBS not affect the proteinase K treatment?

Response: We used Schneider's medium without serum for the conditional medium experiment. We thank the reviewer for pointing out the error and have removed the word "complete" from the sentence.

Is the term "large epithelial tissue" appropriate to describe the fat body in line 518?

The authors thought that the fat body is epithelial tissue, but fat body is more analogous to liver and adipose tissue in vertebrate. Thus, the Drosophila fat body is not classified as epithelial tissue.

Response: We have removed the phrase “large epithelial tissue” term from the sentence. We have replaced the term “fat epithelium” with “adipose tissue” in the text.

Reviewer #2 (Remarks to the Author):

Overall, the authors were responsive to the comments on the initial submission. The paper provides interesting findings of non-gap junction-based calcium waves in the fat body and a comprehensive analysis of the mechanisms.

(Minor points): One remaining point is that the authors should clarify if the traces in the figures are from a single representative cell or averaged for the tissue (Fig. 2B and Fig. 2D vs. Fig. 6C & D). The shaded regions suggest an averaging of multiple samples, but this was not completely clear in reviewing the text. The shaded intervals should be explicitly defined in captions, or if applicable, state that the trace is representative of a single cell.

Response: Because fat cells at different regions oscillate with different phases, averaging the signal across the entire tissue will lose the temporal dynamics. Therefore, for **Fig. 2B** and **2D**, the temporal Ca^{2+} dynamics of a single fat cell was plotted. We have added the description in the corresponding figure legends: “Representative trace of Ca^{2+} dynamics from a single fat cell was plotted.” As for the **Fig. 6C** and **D**, the signals were from averaged intensity as we are comparing the overall intensity of Ca^{2+} in fat. We have stated in the figure legends that “Time courses of average Ca^{2+} signal from multiple larvae were plotted as mean \pm S.E.M. (shaded region).”

Fig. 7B-D mentions representative results, but the shaded regions are not fully described. Is Fig 8F only one trace or an average?

Response: We have added the required clarifications in the figure legends: in **Fig. 7B-D**, “Time courses of average Ca^{2+} signal from multiple larvae were plotted as mean \pm S.E.M. (shaded region).”; in **Fig. 8F**, “A representative trace of Ca^{2+} dynamics from a single APC cluster was plotted in **(F)**”.

Similarly, there is a question of how analysis is shown in SI Fig. 10C and SI Fig 10.

Response: We have added the clarifications in all figures that contain similar quantifications: in **Fig. 4H**, “Time courses of average Ca^{2+} signal from multiple larvae were plotted as mean \pm S.E.M. (shaded region).”; in **Fig. 4J-K**, “A representative trace of average Ca^{2+} intensity from a CO_2 treated larva was plotted in **(J)**. Time courses of average Ca^{2+} intensity from multiple larvae were plotted as mean \pm S.E.M. (shaded region) in **(K)**.”; in **SI Fig. 3A**, “A representative trace of Ca^{2+} dynamics from a small region containing 1-2 fat cells was plotted.”; in **SI Fig. 3E**, “Representative traces of Ca^{2+} dynamics from a single fat cell was plotted.”; in **SI Fig. 7E**, “A representative trace of Ca^{2+} dynamics from a single fat cell was plotted in **(E)**.”; in **SI Fig. 10C**, “Time courses of average Ca^{2+} signal from multiple larval brains were plotted as mean \pm S.E.M. (shaded region).”; in **SI Fig. 10E**, “Representative Ca^{2+} activity from a single fat cell was plotted in **(E)**.”; in **SI Fig. 11E**, “Time courses of average Ca^{2+} signal from multiple larvae were plotted as mean \pm S.E.M. (shaded region).”

Additionally, the citations could have been more comprehensive and complete regarding key recent findings of intercellular calcium waves in other Drosophila epithelia, such as the wing disc.

Response: We have added more comprehensive discussion and references about the ICWs in flies, including I. S. Han et al. (Mol Biol Cell, 2024), P. A. Brodskiy et al. (Biophys J, 2019), Q. Hu and M. F. Wolfner (PNAS, 2019), C. E. Narciso et al. (Biophys J, 2017), and A. Sahu et al. (Plos Genetics, 2017), Balaji et al. (Sci Reports, 2017), Restrepo et al. (Nat Comm, 2016) (All changes were highlighted in yellow in the main text.)

Reviewer #2 (Remarks on code availability):

The readme file was minimal. The code is for Matlab. The code appears to be commented.

Response: We have updated a new “readme” file with a more detailed description of the programming.

https://github.com/Sy-Luo/HeLab_CalciumWave_Simulation_2024/tree/main

~~~~~

### **Responses to the reviewers' comments.**

We appreciate the reviewers for their insightful and constructive comments and suggestions on our manuscript. In this revised manuscript, we have addressed all the comments by either performing the suggested experiments or by rewriting the text accordingly. Below, please find our detailed, point-by-point responses (in black) to each of the comments (which are fully copied in blue and italicized).

### **Point-by-point responses**

***Reviewer #1:** The authors found that intercellular calcium waves (ICWs) occurs in larval fat bodies and identified AKH as the trigger for ICWs. ICWs in larval fat bodies occur in the AP body axis, whereas in ex vivo culture ICWs occur randomly. Random ICWs is mediated by the gap junctional protein Inx2, but Inx2 is not involved in the ICW pattern in vivo. Since overexpression of AKH in fat bodies prevents ICWs along the AP body axis, they thought one possible mechanism is that the concentration of AKH is higher in the head region, which is closer to the ring gland, a source of AKH. Together with computational modeling, they proposed the cyclic ICW pattern along the AP axis is caused by the short half-life of AKH and its cyclic release.*

*Specific amino acids that promote AKH release have also been identified. For example, small polar amino acids were found to raise Ca2+ of AKH-producing neurosecretory cells (APCs) and promote AKH release. The present study suggest that AKH release regulated by starvation and amino acids generate ICWs. This study was interesting because they discovered ICWs and approached its propagation mechanism through experiments and simulations. It is assumed that AKH release occurs in a pulsatile*

*manner under normal rearing conditions, but can pulsatile AKH release be shown experimentally?*

**Response:** We thank the reviewer for recognizing the value of our research. The pulsatile release of AKH is a key inference drawn from our live-imaging experiment. However, directly measuring hormone release in small animals such as fruit flies remains a challenging experiment that is currently beyond our capabilities.

Taking the well-studied insulin as an example, its pulsatile release, with a period of approximately 4-10 minutes, was experimentally identified through the perfusion of the pancreas from large mammals and measuring the Insulin in the effluent every minute (reviewed by Niels Pørksen et al. 2002, doi.org/10.2337/diabetes.51.2007.S245). However, due to the small size of the fruit fly, it is impractical to extract hemolymph from a single fly continuously and determine its AKH concentration. Alternatively, cytosolic  $Ca^{2+}$  oscillation has been used as an indicator of Insulin release from isolated islets (reviewed by Niels Pørksen et al. 2002). We performed  $Ca^{2+}$  imaging of APCs both *in vivo* and *ex vivo* but have not found a clear periodicity in the  $Ca^{2+}$  levels in APCs, which may be due to our current technique limitation: the signal from *in vivo* imaging of APC is still highly noisy which prevents us from extracting fine temporal structure (**Fig. 7**), while the *ex vivo* culture conditions may not optimally preserve the natural excitation dynamics of APCs (**Fig. 8E-F**).

On the other hand, we still consider that the pulsatile release of AKH is the most plausible inference from our observation. Firstly, we have demonstrated that the formation of periodic global intracellular waves (ICWs) is triggered by AKH released from APCs (**Fig. 2**). Secondly, we observed that the acute inhibition of APCs by application of  $CO_2$  immediately blocked the AKH-triggered  $Ca^{2+}$  activity in the fat body (**Fig. 4F-L**), suggesting that the released AKH has a half-life of approximately 35 seconds, which is much shorter than the 3–4-minute interval of global ICWs. Taken together, these results strongly support our hypothesis that AKH is released from APCs in a pulsatile manner.

*Specific Comments:*

*Fig. 1B. The quantification method is unclear. Is the quantification performed on the whole body?*

**Response:** No, it is  $Ca^{2+}$  from a selected 100x100  $\mu m$  square region at the midsection of a larvae. We now clearly state it in the figure legends.

*Fig.1C, D. How was the tissue used for coculture prepared? Also, there is no description of how coculture was performed.*

**Response:** In the method section, we have included a schematic illustration in Supplementary Fig. 1D and a detailed description of the experiment in the method section as follows: "The fat bodies of 3rd instar larvae were dissected in complete Schneider's medium (Schneider's medium from Sigma, supplemented with 10% FBS and 1% streptomycin and penicillin). A cleanly dissected fat body without any other associated organs was gently rinsed once with the same medium to remove larval hemolymph and then transferred using a transfer pipet to the center of a 35 mm dish

with a 20 mm glass bottom (Nest, 801001), along with a drop of Schneider's medium (~20  $\mu$ l). Next, a 1x1cm square cellulose tea-bag paper (or nylon film with approximately 100  $\mu$ m pore size) was placed on top of the fat body to spread it. Subsequently, a stainless-steel ring with a diameter of 1 cm was gently placed on top of the film to prevent floating of the tissue. Finally, 200  $\mu$ l of complete Schneider's medium was added on top of the fat body. The fat body was then imaged using an inverted microscope. For the co-culture experiment, specific organs such as the brain, gut, or muscle/cuticle were dissected, rinsed twice, and transferred into the fat body culture dish. Organs from 8 to 10 larvae were used to activate the fat within 200  $\mu$ l of culture medium. The  $\text{Ca}^{2+}$  activity reported by GCaMP5G was examined for at least 30 minutes after co-culture."

*Fig. 1E. There is no description of the quantification method.*

**Response:** We have grouped the image quantification descriptions into a new section of Image processing and quantification in the method: "Kymograph of  $\text{Ca}^{2+}$  waves were used to highlight the directive propagation of the wavefront in the fat body of the 3rd instar larva. Initially, an average density projection was executed on each frame along the Y-axis (representing the direction from the head to the tail of the larva). This process compressed each frame into a solitary line of pixels, spanning from the head to the tail. Subsequently, all these lines from each frame were combined into a kymograph, with time as the X-axis, to illustrate the direction of  $\text{Ca}^{2+}$  wave propagation.

The temporal dynamics of  $\text{Ca}^{2+}$  activity within a specific fat cell or region were plotted using ImageJ. Specifically, a region of interest (ROI) containing the cell or region of interest was first selected in a multi-frame live-image stack. Then, the "Plot Z-axis Profile" in the Stack function group was employed to trace the  $\text{Ca}^{2+}$  level within the ROI over time.

To clearly visualize the calcium ( $\text{Ca}^{2+}$ ) dynamics within the fat bodies, background fluorescence that does not vary over time was eliminated by subtracting an averaged image (calculated through the mean projection of the time-lapse sequence) from each individual frame. Note that cells damaged during dissection and transfer of tissue typically exhibit a consistently high  $\text{Ca}^{2+}$  signal, which is also removed by this process. The edges of the tissue were delineated with the Canny edge detector in MATLAB.

To quantify the percentage of  $\text{Ca}^{2+}$  active cells, the whole fat tissue was first selected with a low-intensity threshold (selected by masking the entire fat body) to calculate the total area. Then,  $\text{Ca}^{2+}$  positive cells were selected with a high threshold (selected by masking the cells with obvious  $\text{Ca}^{2+}$  signal, usually ~3-5 times higher than the  $\text{Ca}^{2+}$  negative fat cells). As fat cells are essentially uniformly sized, the ratio between the  $\text{Ca}^{2+}$  positive area and total fat area was calculated as the percentage of  $\text{Ca}^{2+}$  active cells.

To calculate the  $\text{Ca}^{2+}$  diffusion area, as presented in **Fig. 3C**, we first used the threshold function to select all the  $\text{Ca}^{2+}$  positive cells in the fat body. Then the area of individual  $\text{Ca}^{2+}$  positive region in the field was calculated by the Analyze Particles function in the ImageJ. The average size of  $\text{Ca}^{2+}$  positive region/diffusion area was calculated from different 20-min time-lapse imaging data.

To calculate the local Ca2+ wave velocity, as presented in **Fig. 3E**, we first manually defined two lines (separated from each other by ~50 μm) parallel to the wavefront. Subsequently, we measured the time required for the wavefront to traverse the distance between these two lines. The wave speed of these local ICWs was determined by dividing the distance by the time.

To assess the Ca2+ oscillation correlation, two adjacent cells exhibiting Ca2+ oscillations were randomly selected. The sequences of Ca2+ intensity changes over time for these two cells were individually obtained. Subsequently, the absolute value of the Pearson correlation coefficient between the two sequences was calculated.”

*Fig.1G, H. There is no description of how to prepare the conditioned medium and how to treat with DNase and RNase enzymes.*

**Response:** We have added the following description in the method section: “For the preparation of brain-conditioned medium, 20 brains from 3rd instar larvae were initially dissected in PBS and then transferred to 400 μl of complete Schneider’s medium and incubated for 1 hour at room temperature. Subsequently, the supernatant of the incubation was collected and referred as the brain-conditioned medium. To treat the conditioned medium, proteinase K (0.1 mg/mL), DNase I (1 U/mL), or RNase A (0.1 μg/mL) was added and incubated at 37°C for 2 hours. For filtering the conditioned medium, a 0.5 mL Pierce 10K MWCO concentrator (ThermoFisher 88513) was used. The filtration was conducted on a benchtop centrifuge at 15,000g for 30 minutes and approximately 200 μl of the filtered medium was collected for the treatment of the cultured fat body.”

*Fig.2A There is no description of how the peptide used was prepared.*

**Response:** We have added the details about the peptide preparation in the method section: “For the neuropeptide treatment experiment, neuropeptide stocks were prepared by dissolving peptide powder (synthesized by Genscript and sequences shown in **Supplementary Table 2**) in either water or DMSO to a concentration of 1 mg/mL depending on solubility. For peptides with disulfide bonds (AstC, CCHa1, and CCHa2), 100 μg/mL peptides were treated with 10% DMSO in PBS at room temperature for 6 hours. The efficiency of the reaction was assayed by Ellman reagent (DTNB assay kit, Solarbio BC1175), and more than 85% of the peptides were oxidized. During live imaging of the cultured fat body, the samples were first recorded for 5 minutes (with a time interval of 5 seconds) to generate a baseline. Subsequently, the medium was replaced by Schneider’s medium containing 100 ng/mL of different synthesized *Drosophila* neuropeptides, and recording was carried out for 20 minutes (time interval: 5 seconds).”

*Fig.2D It is unclear whether AKH is adequately removed by the addition of the antibody. Experiments using control antibody is also necessary. However, since the requirement of AKH and AKHR has been shown genetically, antibody experiment may not be necessary.*

**Response:** Judging from the rapid decrease in Ca2+ activity, we infer that the antibody

effectively removes available AKH. Additionally, we have included a control rabbit serum, as suggested by the reviewer, in **Figures 2D-E**.

*Fig.S3B. Explanation of the starvation experiment is needed.*

**Response:** We have added a detailed description of the starvation experiment in the figure legend: "Early 3rd instar digging/feeding larvae with fat body-specific expression of *GCaMP5G-T2A-mRuby* were collected from standard lab food, washed, and transferred onto a 2% agarose (in water) plate for starvation. After 2 h or 8 h of starvation, more than 10 larvae were kept on 2% agarose and imaged for 20 minutes under free-behavior conditions. The period of the ICWs from head-to-tail of the larvae was calculated in ImageJ."

*Fig.3E. How did you quantify "control local"?*

**Response:** We have added a detailed description in the image processing and quantification method section: "To calculate the local Ca2+ wave velocity, as presented in **Fig. 3E**, we first manually defined two lines (separated by ~50 μm) parallel to the wavefront. Subsequently, we measured the time required for the wavefront to traverse the distance between these two lines. The wave speed of these local ICWs was determined by dividing the distance by the time."

*Fig.3J-K. AKHR is stimulated by AKH, which promotes TAG catabolism. The involvement of ICWs for TAG catabolism is not clear.*

**Response:** In studies on lipid metabolism in *Drosophila*, Ca2+ signaling and Gαq activity have been shown to promote triglyceride (TAG) catabolism and reduce fat storage (Baumbach et al., 2014, DOI: 10.1016/j.cmet.2013.12.004; Baumbach et al., 2014, DOI: 10.1016/j.jgg.2014.03.005; Maus et al., 2017, DOI: 10.1016/j.cmet.2016.12.021).

In our work, based on *in vivo* calcium imaging experiments in both larvae and adult flies, AKH is identified as the primary regulator of Ca2+ increases in the fat body, supporting the notion that Ca2+ is a downstream effector of AKH that controls TAG catabolism. To further substantiate our claim, we added measurements of TAG levels in adult flies following the modulation of Ca2+ levels by knocking down SERCA or Gαq (**Supplementary Fig. 4E**). In addition, AKH is considered a functional homolog of human glucagon which acts in a similar fashion to increase cytosolic Ca2+ which subsequently mediates hepatic lipolysis in mammals (Perry et al. 2020, doi.org/10.1038/s41586-020-2074-6).

*Fig.4F-H. CO2 treatment reduces APC activity, but the relationship between APC Ca2+ dynamics and AKH release has not been directly shown. I am also wondering whether the half-life of AKH can be directly measured?*

**Response:** We have demonstrated that silencing APC with Kir2.1 expression significantly reduced AKH release (**Fig. 2G**), supporting that AKH release requires neuronal activation. Additionally, Ca2+ increases have been considered by most previous studies as a reliable indicator of neuronal activation. We also show that Ca2+

increases in the APCs are triggered by Thr-containing medium, which also elicits AKH release from the APCs (**Fig. 8B-G**). Collectively, these data support the positive correlation between increased  $\text{Ca}^{2+}$  activity in the APC and AKH release.

Regarding the half-life of AKH, direct *in vivo* measurement remains challenging. In larger animals, the half-life of a hormone in circulation can be determined by continuously collecting blood over a specific period. However, collecting sufficient hemolymph from flies for dot blot or ELISA would take more than 10 minutes and require multiple flies, which makes measuring a half-life of less than several minutes impractical. Furthermore, since AKH release is not synchronized between flies, pooling hemolymph samples would also confound the estimation of the half-life.

Based on our brain-conditioned medium experiment, AKH released into the cell culture medium can continuously activate the fat body for several hours, suggesting that AKH is not inherently unstable. (Note, this data also suggests that the fat body does not desensitize to AKH by itself.) In the case of insulin, which has a half-life of several minutes in the bloodstream, degradation primarily occurs in the liver in mammals. Therefore, we suspect that AKH is degraded by certain organs such as the gut or muscle. We have added other dissected gut or muscle/cuticle, to the brain-conditioned medium to test whether these organs promote the degradation of AKH, but observed no significant effect. It is possible that the other organ(s) is responsible for AKH clearance in flies, or that the current *ex vivo* system is not optimal for the proper functioning of the cultured organs.

*Figure 5I. The authors attribute the difference in ICW response between larvae and adults to the uniform distribution of extracellular AKHs in adults. However, the argument that this difference is due to the location of APCs in the thorax seems to be weak; it is possible that AKH is abundantly secreted in the adult body fluid, as is the case with the Akh overexpression movie in the fat body in Figure 3D, where ICW occurs in a random direction. This can be test by comparing AKH levels in the blood lymph of larvae and adults by western blot or ELISA.*

**Response:** We thank the reviewer for the constructive suggestion and have conducted a dot blot analysis to compare AKH levels in the hemolymph of larvae and adults. However, we did not observe any significant differences in AKH levels between larval and adult hemolymph (**Supplementary Fig. 8**).

We then hypothesized that the difference might lie in the sensitivity of the fat body to the AKH ligand. To test this, we dissected fat bodies from both larvae and adults and treated them to different concentrations of AKH peptide. Interestingly, adult fat bodies exhibited a stronger response to higher concentrations of AKH (>10 ng/mL) compared to larval fat bodies, as evidenced by a greater proportion of adult fat cells showing a significant increase in  $\text{Ca}^{2+}$  (**Supplementary Fig. 8B**). However, within the 1-10 ng/mL range of AKH, which elicits  $\text{Ca}^{2+}$  increases comparable to those observed *in vivo*, both adult and larval fat bodies displayed similar levels of sensitivity.

In summary, our data indicate that there is no significant difference in AKH concentration or tissue sensitivity between adult and larval fat bodies. Interestingly, a recently published paper by Laura Blackie et al. (2024 DOI: 10.1038/s41586-024-

07463-4) revealed that there are complex interorgan connections (shape and arrangement of organs) in adult flies. Meanwhile, our observation of AKH-triggered global intracellular waves (ICWs) in larvae suggests that hemolymph flows efficiently, facilitating the rapid transport of secreted AKH from the head to the tail. Therefore, a potential increase in interorgan connections in adult flies may impede the swift transport of AKH and promote a more uniform AKH concentration in the abdomen.

Meanwhile, we do not rule out the possibility that potential modulators of AKH signaling may differ between larvae and adults, which could account for the observed differences. Additionally, the APCs in adults may release AKH in a continuous rather than pulsatile manner. We have included these potential explanations in our discussion.

*Fig. 5I. The authors propose a model of pulsatile AKH secretion in larvae. However, the rhythm of AKH secretion itself is unclear. Considering that the velocity of bead movement in Fig. 4E (250  $\mu\text{m/s}$ ) is much faster than the wave velocity of global ICWs in Fig. 3E (40  $\mu\text{m/s}$ ), it might be more relevant to assume the desensitization speed of the fat body rather than focusing on AKH secretion.*

**Response:** We also suspected that the fat body might desensitize upon continuous AKH exposure. However, treating the cultured fat body with either synthesized AKH peptide or brain-conditioned medium elicited sustained  $\text{Ca}^{2+}$  oscillations, suggesting that no significant desensitization happens at the physiological level of AKH. Desensitization was observed with extreme AKH concentrations above 1  $\mu\text{g/mL}$  (**Supplementary Fig. 3A**). At this non-physiological AKH concentration, all fat body cells showed elevated  $\text{Ca}^{2+}$  levels, which are not observed *in vivo*. In addition, at these abnormally high AKH levels,  $\text{Ca}^{2+}$  oscillations in the fat body remained detectable. This suggests that AKHR is a GPCR that does not desensitize easily. This may be consistent with the short half-life of AKH *in vivo*. Since the ligand itself is short-lived, the receptor does not need an additional layer of signal shutdown mechanism.

Regarding the difference between hemolymph velocity and AKH propagation, a high blood flow velocity does not imply that a high concentration of AKH will travel at the same speed. Once released from APCs, AKH can be considered to be stored locally within the body cavity of the larval head region. This is supported by the observation that stopping the heartbeat with  $\text{CH}_3\text{Cl}$  treatment traps AKH in the head region, eliciting only a local  $\text{Ca}^{2+}$  response (**Fig. 4A-C**). The circulating blood flow in the larval body does not carry all the AKH at once but rather gradually dilutes the pool of AKH in the head region, carrying portions of it along with the flow. The rate at which AKH is diluted and transported away from the head region depends on the volume of the head cavity and the flow rate of the circulating hemolymph. This compartment model is widely used in physiologically based pharmacokinetic modeling when studying drug transport in animals.

One way to illustrate the model is to imagine a "lake of AKH", and the bloodstreams can be imagined as many flowing rivers. Some of these rivers flow through the lake, carrying portions of AKH with them as the AKH pool exchanges material with the hemolymph circulating nearby. And the downstream AKH concentration is the result of the mixing of these rivers. Over time, the AKH in the 'lake' gets diluted and carried

away by the rivers but at a much slower speed than the flow speed of an individual river. To illustrate this, we have added a plot of simulated AKH concentration in **Fig. 5B**.

We thank the reviewer for pointing this out. Our previous illustration of the AKH gradient in **Fig. 5K** was incorrect and has been revised.

*Why is the Calcium intensity increased in *inx2*-RNAi? Based on Suppl Fig. 7B-D, is there a mechanism that delays AKHR desensitization? Could the suppression of local ICWs lead to a compensatory increase in global ICWs? Since this point is also supported by mathematical models, further discussion based on the mathematical model on which factors are important would be possible.*

**Response:** We thank the reviewer for pointing this out. The increase in  $\text{Ca}^{2+}$  observed after *inx2*-RNAi in larvae is a surprising finding. From our computational simulations, the  $\text{Ca}^{2+}$  increase appears to originate from internal stochastic fluctuations within the system. When the random fluctuation factor is removed, the  $\text{Ca}^{2+}$  increase no longer occurs in *inx2*-RNAi. Additionally, the inhibition of gap junctions indeed caused a significant increase in variance as the wave propagates from the head to the tail (Fig. 5I-J), which is consistent with the idea that gap junction knockdown weakens the coupling between neighboring cells.

We currently hypothesize the following mechanism: stochastic fluctuations in  $\text{Ca}^{2+}$  generate a  $\text{Ca}^{2+}$  cycle/status that is not in phase with the external AKH ligand, thereby interfering with and partially 'canceling' the effects triggered by AKH. With functional gap junctions, these fluctuations can propagate between cells and affect more cells. However, with *inx2*-RNAi, the local  $\text{Ca}^{2+}$  fluctuations can no longer propagate, resulting in a reduced 'canceling effect' on the external AKH signaling, and subsequently a stronger effect by external AKH. That model can also explain why  $\text{Ca}^{2+}$  decreases after *inx2*-RNAi, as in adults, a uniform external AKH triggers random  $\text{Ca}^{2+}$  activation which requires gap junctions for propagation.

As the reviewer suggested, the overall effect is that the decoupling of neighboring cells decreases local ICWs but promotes global ICWs. We have added this explanation in the revised discussion section.

*Fig.S11. What is the status of AKH in normal food?*

**Response:** We added the comparison of AKH retention in APCs between normal food, 5% sucrose, and starved larvae in **Supplementary Fig. 12D**.

Minor Points:

*1. The velocity measurement experiments using beads in larvae shown in Fig. 4D are excellent. However, compared to the videos of adults, the beads appear to get stuck in the tissue, preventing them from moving. Additionally, the fact that many beads are not moving is a concern. While using smaller beads may be difficult, it would be advisable to mention these limitations as technical constraints.*

**Response:** We agree with the reviewer that the speed of the beads may be lower than the actual velocity due to collisions and interactions with other organs. However, even

this lower bound of the speed is still much higher than the traveling speed of the global ICWs, suggesting that the circulating hemolymph is sufficiently fast for the transport of secreted AKH. As we mentioned above, the compartment model for material transport within animals only requires that the flow speed between two compartments be greater than the observed transport speed of the carried substance. We have added this limitation of the method to our discussion.

*2. The notation for the fat body Gal4 driver in Figure 1 varies between Fat>, Lpp>, and Fb>. It would be better to standardize the notation.*

**Response:** We have corrected the notation. In this study, we used two fat body-specific Gal4 drivers, Fb-Gal4 and Lpp-Gal4, both of which have been employed for fat-specific expression in many previous studies. Our initial experiments utilized Fb-Gal4. However, we later switched to Lpp-Gal4 because it has more available tools built around it. No difference in Ca2+ dynamics was observed when using the two Gal4 drivers to express GCaMP5G in the fat body.

*3. The choice of GCaMP5G raises some questions. Compared to GCaMP6s, the  $\Delta F/F$  is lower (Nature. 2013 Jul 18; 499(7458): 295–300). And GCaMP6s was used in the ICWs and SIDICs papers in wing discs. Was this lower sensitivity of 5G advantageous for detection of ICWs in the fat bodies?*

**Response:** We have tested different GCaMP variants, including GCaMP5G, GCaMP6s, and GCaMP7s, in the fat body and found that GCaMP5G provides the best signal-to-noise ratio in adipose tissue. Because the newer GCaMPs have a higher affinity for Ca2+: the Kd for GCaMP5G is 0.45  $\mu$ M, whereas the Kd for GCaMP6s is 0.144  $\mu$ M (Jacob L. Perry et al., 2015, 10.1016/j.ymeth.2015.09.004). The new GCaMPs (6 and higher) showed a much higher basal signal, resulting in a smaller effective  $\Delta F/F$  compared to GCaMP5G. As the reviewer suggested, we found that the lower sensitivity of GCaMP5G better matches the cytosolic dynamics in fly adipose cells.

*4. The differences in response in adult and larval FB are very interesting. If data were available, could they be discussed in terms of differences in localization patterns of AKHR and inx2 expression? What about differences in fat amount or metabolic demands during development and maturation stages?*

**Response:** We thank the reviewer for suggesting this interesting direction. We have conducted experiments to explore this possibility. As mentioned above, no significant difference in AKH sensitivity was observed between larval and adult fat during *ex vivo* experiments (**Supplementary Fig. 8B**). Both adult and larval fat showed a significant reduction in the correlation of Ca2+ activity between neighboring cells, suggesting that *inx-2*-RNAi affects both larval and adult fat similarly under *ex vivo* conditions (**Supplementary Fig. 6C-F**). Thus, our data suggest that the differences between adults and larvae after *inx-2*-RNAi are fat non-autonomous and likely to be caused by factors other than the fat body, as we have discussed above. In terms of fat amount or metabolic demand, we currently lack effective assays to measure and compare these

parameters between stages. Nevertheless, as discussed above, both our experimental data and computational simulations support the existence of differences in the dynamics of external AKH.

*5. There is character encoding issue on the y-axis label for Velocity in Suppl Fig. 8B.*

**Response:** We have corrected the error.

*6. Is the term "large epithelial tissue" appropriate to describe the fat body in line 518?*

**Response:** We have removed "large" in the sentence.

*7. It would be better to unify the use of the abbreviation for insulin-producing cells (IPCs) throughout the manuscript.*

**Response:** We have changed the abbreviation as the reviewer suggested.

*Reviewer #2: This is a well-written and thorough manuscript uncovering the role of Ca2+ signaling in the Drosophila fat body in regulating fat storage mobilization. As mentioned in the Discussion section, this study also provides insights into the Liver-alpha-Cell axis in human biology. In this context, Ca2+ signaling is stimulated and acts via hormonal (Adipokinetic hormone, AKH) signaling from specialized cells (APCs). AKH is sensed by the fat body cells expressing the AKH-Receptor (AKHR), a G-Protein Coupled Receptor (GPCR), that activates Gαq to stimulate Ca2+ signaling. The authors investigated the mechanism of transport of AKH, which is circulated in the hemolymph throughout the larval body and is secreted from the APCs, and how the fat body is able to respond to this hormonal signaling with two distinct types of intercellular calcium waves (ICWs). The authors argue that two distinct mechanisms of ICWs dynamics are active in larval fat body: ICWs are spread by either intercellular gap junctions channels providing a slow, local spread, or alternatively, through convective transport due to hemolymph flow that results in extensive calcium waves, initiated through AKHR-Gαq signaling. They evaluated the differences in calcium signaling between larval and adult stages in the fat body. This aspect of the story mostly confirms and extends previous findings already described in the literature regarding the role of Gαq, PLC Beta and Ca2+ signaling in the fat body. A new and significant finding is that this AKH hormonal signaling in larvae, in addition to hunger, can also be triggered by dietary addition of two amino acids, including Methionine and Threonine. These amino acids are sensed by APCs and stimulate AKH release, triggering ICWs and fat mobilization in the fat body. The manuscript is logically organized. The data is robust, and most of the authors' data interpretation and conclusions are reasonable.*

*However, some of the results are not unexpected. The authors should clearly distinguish what is a confirmation of the already known findings and what are the new findings. The claimed new results are stated in the abstract, citing "We identified Adipokinetic hormone, ... as the key factor driving Ca2+ activities in adipose tissue," it should be restated more circumspectly as "we confirmed previous reports with an additional line of evidence." This result is not a completely new finding, and was*

*described previously in the literature. Furthermore, the GPCR AKH-Receptor works via Gaq also was described previously in the literature (for example, Bumbach “Gaq, G $\gamma$ 1 and Plc21C Control *Drosophila* Body Fat Storage” <https://doi.org/10.1016/j.jgg.2014.03.005>).*

**Response:** We thank the reviewer for the comments and have rephrased the sentences in the manuscript accordingly.

*A significant new finding is that authors are claiming that the larval fatbody exhibits two distinct types of ICWs. thors tested a variety of amino acids and found that specifically methionine and threonine are sensed by APCs to trigger AKH release, which in turn stimulates ICWs in the body. This suggests that these two amino acids are able to trigger “starvation”-like responses. It is possible that foods that are rich in these two amino acids are not the preferred/best food sources for the larvae, and if given a choice they would not choose to have this nutritionally “poor” diet. As in another study where one of the coauthors had participated (<https://doi.org/10.1038/s41586-022-04960-2>) about adult fly preference of Leucine in the food. In this study Leucine did not trigger APCs activation, clearly demonstrating that Leu is a preferred dietary amino acid for fly larvae. According to flybase expression data, AKHR is expressed in the larval fat body, but in adults, it is strongly expressed in the head. This suggests that AKH could be sensed in adult heads and could potentially trigger behavioral changes in adults to search for new food sources, for example. The impact of this finding is significant as it has the promising potential for application towards the creation of new medication for obesity, heart disease, and diabetes for humans.*

*There are a couple of areas for improvement in the present manuscript. A significant finding in the manuscript is that ICWs can happen independently of gap junctions, and therefore are not spread cell-to-cell, but rather reflect an extracellular gradient of hormones But this finding could be bolstered further. This claim is based on using *Inx2*-RNAi-mediated knockdown, but no verification was made to ensure that the *Inx2* knockdown was 100% knockdown of gap junction activity. Even though pharmacological treatment was used, it was performed in cultured fat bodies and not in vivo, so it can not be used as a substitute for in vivo verification. Also, it is not explicitly mentioned what is the sample size in these experiment. For example the ex vivo experiment in Fig. 3.*

**Response:** We greatly appreciate the reviewer for the constructive suggestions, which provides a very interesting direction for our future studies. To verify the efficiency of *Inx2*-RNAi, we first tested the efficiency by RT-PCR and found over 85% reduction in the *Inx2* mRNA in larval fat body (**Supplementary Fig. 6B**). We also realized that reduction in mRNA or protein level is not necessarily equivalent to the function reduction. Because the primary function of gap junction is to couple Ca2+ activity in neighboring cells (in this study), we compared the Ca2+ signal correlation between neighboring cells with and without *Inx2* knockdown. In both larval and adult fat, Ca2+ activities are highly correlated with a Pearson correlation coefficient close to 0.8-0.9. In contrast, the correlation efficiency decreased to 0.2 after *Inx2* knockdown

(**Supplementary Fig. 6C-F**), suggesting that  $\text{Ca}^{2+}$  activity in neighboring cells is significantly uncoupled in *Inx2*-RNAi fat cells.

As for the sample size, we have at least three different biological replicates in each experiment. As we have displayed actual data points in our bar plots, we did not include the n number in the figure legend, instead, we have provided a source data file in Excel showing all the original data used for all the plots (**Source data 1**).

*Another concern regarding the clarity of presentation of results is that in Supplementary Video 8. The *Luc-i* condition includes global ICWs (the waves are extensive), although the authors state completely opposite, that no global ICWs were observed in adults, and that global ICWs are stage-specific, only attributed to the larval fat body. The definition of global vs. local ICW is vague in this case. Also, the sample size of this study should be more clearly identified.*

**Response:** We thank the reviewer for raising this question and have revised our manuscript to better define the difference between global and local ICWs. Briefly, both local and global ICWs involve multiple fat cells being activated simultaneously. For example, in *ex vivo* cultured wild-type larval fat, there are extensive cells exhibiting collective  $\text{Ca}^{2+}$  activities (**Supplementary Video 5**). However, the fundamental difference between global and local ICWs lies in whether the ICWs need to be understood at the organism level. For global ICWs, there is a single pacemaker at the head, the APCs, and the wave travels throughout the entire larval fat body from head to tail. Thus, the whole organism must be considered to understand this phenomenon. In contrast, ICWs in the adult stage show multiple independent pacemakers, and  $\text{Ca}^{2+}$  activity spreads regionally. Therefore, studying a 'local' region of the adult fat is sufficient to recapitulate the features of the ICWs.

*The *Inx2*-RNAi line used in the study is made using Valium10 vector, and *lpp-Gal4>GCamp5G* driver does not seem to have *Dicer2*. ICWs in the *Inx2*-RNAi look very similar to the control, there could be an incomplete penetrance phenotype for this *Inx2*-RNAi line. To strengthen the result, a secondary method such as CRISPR would be needed. Even though a pharmacological gap junction inhibitor was used in cultured fat bodies the results agreed with *in vivo* *Inx2*-RNAi results. To verify this result, complete inhibition is needed, perhaps with stronger inhibition, such as a second distinct line for *Inx2*-RNAi, or by repeating the *in vivo* experiment as in Supplementary Video 6 using another driver that has *Dicer2* in its genotype that will help to improve the knockdown or any other way that will provide a concrete evidence that *Inx2*-RNAi mediated knockdown was fully effective. This is a significant concern because the authors claim the novel finding of a new previously undescribed mechanism of ICWs based only on the results with *Inx2*-RNAi knockdown (which may not be 100% knockdown). Therefore, the basis of this novel finding should have more parallel lines of evidence.*

**Response:** We agree with the reviewer that the knock-down efficiency of *Inx2*-RNAi is important. Therefore, we have performed a functional assay to assess the intercellular coupling efficiency between neighboring cells. The results suggest that the

function of gap junctions is significantly inhibited by the *Inx2*-RNAi (**Supplementary Fig. 6C-F**).

Meanwhile, the propagation speed of the global ICWs is ten times faster than that of classical ICWs mediated by gap junctions and is also much faster than the free diffusion speed of  $\text{Ca}^{2+}$  in water. This propagation speed cannot be explained by gap junction spreading. Instead, we found that the fast-circulating hemolymph accounts for this rapid global spread of the signal (**Fig. 4**).

*It would also be helpful to perform the same experiment as in Video 8 but in cultured adult fat bodies and then update Fig. 3J, K. The adult cuticle is only partially transparent and prevents the observation of the full range of GCaMP5G, a  $\text{Ca}^{2+}$  sensor, will be out of the way, and the results will be more clearly demonstrating the actual phenomenon of ICWs.*

**Response:** We have performed *ex vivo* culture of the adult fat body with and without *Inx2*-RNAi in **Supplementary Fig. 6E-F**. Live-cell imaging of the *ex vivo* cultured adult fat treated with AKH is shown in **Supplementary Video 7**.

*Specific genetic background descriptors are needed in each and every video and image rather than the nondescriptive word “control,” because this introduces a guesswork for the reader. This abbreviation is used sometimes in the text and Videos, and it is not easy to find out what exactly was used as a control each time. The same applies to the driver lines. As an example: Video 6, that has labels “control” (what was a control?), *Inx2-i* (what was the driver?), AKHR-null, and AKH-OE (again, was the driver used?). It creates extra work for the reader to refer to the SI files. Also, it would be more clear if the location of the head is indicated, to clarify the direction of the ICW spread.*

**Response:** We have revised the labels in the figures and videos as the reviewers suggested.

*Lines 198-199: Please provide a more detailed description of creation of kymographs. Supplementary Video 2: it is not clear where the brain is on the video. There is no clear larval brain outline.*

**Response:** We have updated the method section to include details on the generation of kymographs.

As for **Supplementary Video 2**, because the brains were added after the imaging chamber was prepared, they typically floated in the medium and were out of the imaging field (this is now described in detail in the updated method section and illustrated in **Supplementary Fig. 1D**).

*Lines 42-90, the Introduction section: More than half of the introduction section (lines 64-90) was used to summarize results. Normally, the Introduction section should provide the background necessary for the reader to understand the following “results” section. Furthermore, the Introduction section did not fully cover the scientific literature on the topic of Calcium signaling. It would be helpful to condense the summary of*

*results in the Introduction, and instead add more background information on the Ca2+ signaling. For example, there is a Baumbach, Xu, Hehlert, Kuhnlein 2014 paper that reports Gaq-related Calcium signaling in the fat body and its connection to the AKH and AKHR. (<https://doi.org/10.1016/j.jgg.2014.03.005>) and others.*

**Response:** We have rewritten the introduction section and included the suggested references.

*Lines 104-106 where the authors describe the experimental setup: providing a cartoon or a diagram will improve clarity of the experimental set up.*

**Response:** We have added a schematic illustration in the **Supplementary Fig. 1D**.

*Lines 125-130: the authors should reference Supplementary Figure 1 A.*

**Response:** We have added the reference.

*(Minor issue): The lines 369 and 370 cited source 21 as the location of APCs, but it is not a best resource for describing the location of the APC location in adults, as in Figure 1 of the cited source 21 the location of APCs are shown in the head, which is contrary to what authors are saying. Perhaps an alternate source should be cited regarding the location of the corpora cardiaca DOI 10.1007/s00018-015-2063-3, fig. 1 or fig. 2). The authors are identifying the APCs location correctly, but they need to cite an alternate source.*

**Response:** We thank the reviewer for the suggestion and have revised our manuscript accordingly.

*In general, the introduction section should provide a more contextual literature summary, including ICW in wound detection and healing.*

**Response:** We have revised our introduction to emphasize this aspect.

*L77: Modify to the plural: computational simulations, unless only one simulation was implemented.*

**Response:** We have corrected this.

*L99: be more specific on this point: “using a holder (what kind) in the perfusion chamber”*

**Response:** We have added a detailed description and reference: “Dissected brains were immobilized using a slice holder with nylon grids (SH13, Scientific Systems Design Inc.) in the perfusion chamber.”

*L172 : wave direction analysis methods could have been clearer. Consider revising for clarity.*

**Response:** We have added a new section about image quantification in the updated method: “Kymograph of Ca2+ waves were used to highlight the directive propagation of the wavefront in the fat body of the 3rd instar larva. First, an average density projection was executed on each frame along the Y-axis (representing the direction

from the head to the tail of the larva). This process compressed each frame into a solitary line of pixels, spanning from the head to the tail. Subsequently, all these lines from each frame were combined into a kymograph, with time as the X-axis, to illustrate the direction of Ca2+ wave propagation.”

*L. 186: Is there a reference that needs to be added? “According to the concept of connected components in image processing...” this is not clearly defined.*

**Response:** We have revised the sentence. We have added a description of how the processing is done:” To calculate the Ca2+ diffusion area, as presented in **Fig. 3C**, we first used threshold function to select all Ca2+ positive cells in the fat body. Then the area of individual Ca2+ positive regions in the field was calculated by the Analyze Particles function in ImageJ. The average size of Ca2+ positive region/diffusion area was calculated from different 20-min time-lapse imaging data.”

*L.215: Where is the description of how the transgenic lines were created that were first used in the study*

**Response:** We have added the information on cloning and transgene generation in the method section.

*L.271 need citation for Kir2.1 reference*

**Response:** We have added the reference.

*L. 506 : Does insulin stimulate Ca2+ in fat body like on other tissues such as the wing disc?*

**Response:** We have tested the effect of (100 nM) recombinant human insulin on cultured fat bodies but found no significant increase in Ca2+ level.

(For the reviewer, cultured larval fat bodies were treated with 100 nM Insulin or 100 nM AKH)

*L640 : Making the data available on repository will aid in transparency of results*

**Response:** We have included source data with original data for all the plots.

*The title is somewhat misleading: Novel Calcium waves are not a class of calcium waves. It may be a novel finding, but that is also overstated.*

**Response:** We have changed our title to “Dietary Amino Acids Promote Glucagon-like

Hormone Release to Generate Global Calcium Waves in Adipose Tissues”.

*Computational codes: The study is only replicable if the codes are deposited to Github. The SI text was not fully transparent in describing and annotating computational code. the Statement “all MATLAB code in the simulations are fully disclosed” cannot be verified.*

**Response:** We have uploaded our codes onto the GitHub with the following linkage: [https://github.com/Sy-Luo/HeLab\\_CalciumWave\\_Simulation\\_2024](https://github.com/Sy-Luo/HeLab_CalciumWave_Simulation_2024).

*It was not completely clear how Ca2+ traces are measured (E.g. SI Fig 3E). Is the ROI a particular cell or area and does it differ from experiment to experiment?*

**Response:** We have added an image processing section in the method:” The temporal dynamics of Ca2+ activity within a specific fat cell or region were plotted using ImageJ. Specifically, a region of interest (ROI) containing the cell or region of interest was first selected in a multi-frame live-image stack. Then, the "Plot Z-axis Profile" in the Stack function group was employed to trace the Ca2+ level within the ROI over time.”

*Fig. S4: Why are there two rows for each condition? The darker second row is not explained. The heat maps of rel. F.I. are also not defined.*

**Response:** We have added the description in the new processing procedure section: “To clearly visualize the calcium (Ca2+) dynamics within the fat bodies, background fluorescence that does not vary over time was eliminated by subtracting an averaged image (calculated through the mean projection of the time-lapse sequence) from each frame. Note that fat cells damaged during dissection typically exhibit a consistently high Ca2+ signal, which is also removed by this process. The edges of the tissue were delineated with the Canny edge detector in MATLAB.”

*Definitions like global or tissue-wide Ca2+ waves are not clearly delineated, which makes it difficult to assess the comparisons being made between the larval and the adult experiments. Does a global wave mean that the whole fat body is activated at the same time or that the wave progresses through the whole body?*

*Specifically, how can the effect of GJ inhibition leading to increased Ca2+ amplitude and decrease in frequency be explained in the computational model?*

**Response:** We have discussed the key differences between global and local ICWs above and have revised the manuscript accordingly. Our current hypothesis regarding the observed increase in Ca2+ after *inx2* knockdown is discussed in our response to Reviewer 1. Essentially, this can be viewed as a compensatory effect resulting from the inhibition of local ICWs.

*Reviewer #3 (Remarks to the Author):*

*I co-reviewed this manuscript with one of the reviewers who provided the listed reports as part of the Nature Communications initiative to facilitate training in peer review and appropriate recognition for co-reviewers.*

*Reviewer #4 (Remarks to the Author):*

*I co-reviewed this manuscript with one of the reviewers who provided the listed reports. This is part of the Nature Communications initiative to facilitate training in peer review and to provide appropriate recognition for Early Career Researchers who co-review manuscripts.*

*Reviewer #5: The manuscript by Ahmad et al. describes a novel mechanism of nutrient sensing and metabolic regulation in the fruit fly *Drosophila melanogaster*. The study demonstrates that Adipokinetic hormone (AKH), the fly equivalent of mammalian glucagon, stimulates intracellular  $\text{Ca}^{2+}$  waves in the larval fat body to promote lipid metabolism. Specific dietary amino acids are shown to activate AKH-producing cells, leading to increased intracellular  $\text{Ca}^{2+}$  and subsequent AKH secretion. The study also reveals different mechanisms of  $\text{Ca}^{2+}$  dynamics regulation in larval and adult fat bodies, highlighting the interplay between AKH secretion pulses, extracellular hormone diffusion, and intercellular communication through gap junctions.*

*While it is disappointing that the precise biological significance of the phenomenon has yet to be fully understood, this manuscript represents a significant advancement in our understanding of nutrient sensing and metabolic regulation. The study utilizes innovative techniques and provides novel insights with potential implications for human health.*

*Overall, this is a very nice study, and the reviewer has only a few comments on the manuscript for revision.*

*(1) The authors' computational modeling is impressive as it provides a new strategy for *Drosophila* physiology. The reviewer is curious about the relationship between the predicted distribution of AKH in the body and the predicted  $\text{Ca}^{2+}$  influx in the fat body in this model. The reviewer would like the authors to show the expected distribution of AKH together with the  $\text{Ca}^{2+}$  data in Figure 5A, and discuss the relationship between AKH distribution and  $\text{Ca}^{2+}$ . The reviewer thinks that it would be of interest to many *Drosophila* researchers.*

**Response:** We thank the reviewer for the suggestion and have added new figures illustrating the simulated pattern of AKH concentration over time in **Fig. 5B and E**. We assumed that all AKH was released from the head at time 0 and then gradually diluted and transported along the long axis of the larva. The global ICWs move with the front of the AKH diffusion, as the  $\text{Ca}^{2+}$  in the cells begins to rise when local AKH concentration reaches the activation threshold in the model, while fat cells behind the wavefront gradually cycled into the low  $\text{Ca}^{2+}$  phase. Our previous illustration of the AKH concentration in Fig. 5K was incorrect and has been revised.

*(2) Please carefully check Supplemental Table 2 and review what kind of peptides the authors really used. The reviewer realizes several concerns.*

*a) The peptide sequence of CAPA listed in this table (GDAELRKWAHLLALQQVDL) is entirely different from an actual CAPA mature peptide (GANMGLYAFPRV-amide; see <https://journals.plos.org/plosone/article?id=10.1371/journal.pone.0029897>).*

**Response:** We thank the reviewer for pointing this out. We have thoroughly checked the peptide used in our study and corrected the errors as requested. For the CAPA peptide, we have synthesized a new CAPA with the correct sequence suggested by the reviewer and redone the fat body test in **Fig. 2A**.

*b) While the authors used FQYSRGWTN-amide as Corazonin, the reviewer wonders why the authors used such a shorter peptide than the identified mature Corazonin peptide (QTFQYSRGWTN-amide).*

**Response:** We have synthesized the full Corazonin peptide suggested by the reviewer and redone the fat body test in **Fig 2A**.

*c) While some peptides, such as Ast-C, CCHa1, and CCHa2, have internal disulfide bonds to be active forms, the authors did not state whether they use the proper mature peptides. If the authors used the “wrong” peptides, they did not correctly examine the effect of some peptides, and the reviewer argues that the authors must carefully revise Supplemental Table 2 and related sentences.*

**Response:** We thank the reviewer for pointing this out. We have now used DMSO treatment to form the internal disulfide bonds in all three peptides (Ast-C, CCHa-1, and CCHa-2), and have redone the fat body Ca2+ activity assay using the oxidized peptides. The efficiency of disulfide bond formation was determined to be higher than 80% using the DTNB Ellman's reagent.

*The reviewer also has additional comments as follows.*

*d) “LK” and “Leukokinin” are exactly the same, and there is no need to include both data in Figure 2A.*

**Response:** We have corrected this error.

*e) The reviewer does not see any particular reason why some very well-known peptides, such as NPF and DH44, were not included. If there are too many inadequacies here, changing the logic flow that focuses on AKH may be necessary.*

**Response:** We have added NPF and DH44 as the reviewer suggested. Currently, we have tested 32 peptides out of the approximately 40 known neuropeptides in *Drosophila* that have GPCR receptors (according to the reviews by Dick R. Nässel and Meet Zandawala, 2019, doi: 10.1016/j.pneurobio.2019.02.003). For neuropeptides with multiple sequences, we tried to choose the most studied or abundant ones based on previous literature. We have added this statement in our methods section for clarity.

This study originated from a discovery made six years ago, so some of the most recently studied neuropeptides were not included. We also agree with the reviewer that AKH would be a good candidate after the discovery of fat body Ca2+ dynamics. However, we believe that the peptide list, even though not comprehensive from the current perspective, still provides important information on how the fat body responds to most known neuropeptides through Ca2+ signaling. Additionally, it includes the actual experiments that led to the identification of AKH in this study.